# AUTOMATED OPTIMIZATION MODELING VIA A LOCALIZABLE ERROR-DRIVEN PERSPECTIVE

## ABSTRACT

Automated optimization modeling via Large Language Models (LLMs) has emerged as a promising approach to assist complex human decision-making. While post-training has become a pivotal technique to enhance LLMs' capabilities in this domain, its effectiveness is severely constrained by the scarcity and underutilization of high-quality training data. However, through a detailed profiling of error patterns across various problem-response pairs drawn from post-training, we identify two fundamental limitations of existing automated optimization modeling approaches: (L1) the *sparsity* of error-specific problems and (L2) the *sparse rewards* associated with difficult problems. We demonstrate that these limitations can result in suboptimal performance in domain-specific post-training for LLMs. To tackle the above two limitations, we propose a novel error-driven learning framework—namely, automated optimization modeling via a localizable error-driven perspective (MIND)—that customizes the whole model training framework from data synthesis to post-training. MIND is based on our key observation of the unique **localizable** patterns in error propagation of optimization modelings, that is, modeling errors may remain localized to specific semantic segments and do not propagate throughout the entire solution. Thus, in contrast to holistic reasoning tasks such as mathematical proofs, MIND leverages the construction of a focused, high-density training corpus and proposes **D**ynamic Supervised **F**ine-Tuning **P**olicy **O**ptimization (DFPO) to tackle difficult problems through localized refinement. Its appealing features include that (1) it generates targeted, error-aware training problems that achieve superior sample efficiency, and (2) it ensures a coherent and structured learning progression for stable and effective reinforcement learning on difficult problems. Experiments on six benchmarks demonstrate that MIND *consistently* outperforms all the state-of-the-art automated optimization modeling approaches. Furthermore, we open-source a new training dataset, MIND-Train, and a new benchmark, MIND-Bench, for the automated optimization modeling research community.

## 1 INTRODUCTION

Advances in computational power and algorithmic techniques have made optimization a fundamental tool across engineering (Antoniou & Lu, 2007), economics (Intriligator, 2002), logistics (Bartolacci et al., 2012), manufacturing (Rao, 2010), and artificial intelligence (Kingma & Ba, 2014), enabling more intelligent and data-driven decision-making. Optimization seeks values for decision variables that maximize or minimize an objective function while satisfying a set of constraints. Optimization modeling formalizes complex real-world problems into mathematical representations by defining variables, objectives, and constraints, allowing state-of-the-art solvers such as Gurobi (Gurobi Optimization, LLC, 2024), PySCIPOpt (Berthold et al., 2024), and CPLEX (IBM Corporation, 2024) to efficiently compute solutions. Recently, the emergence of Large Language Models (LLMs) has opened a new avenue for automated optimization modeling, enabling the translation of natural language problem descriptions directly into mathematical formulations and executable solver code. Although automated optimization modeling cannot guarantee complete accuracy, their ability to rapidly generate candidate formulations to support human experts in optimization modeling is nonetheless of substantial practical value.

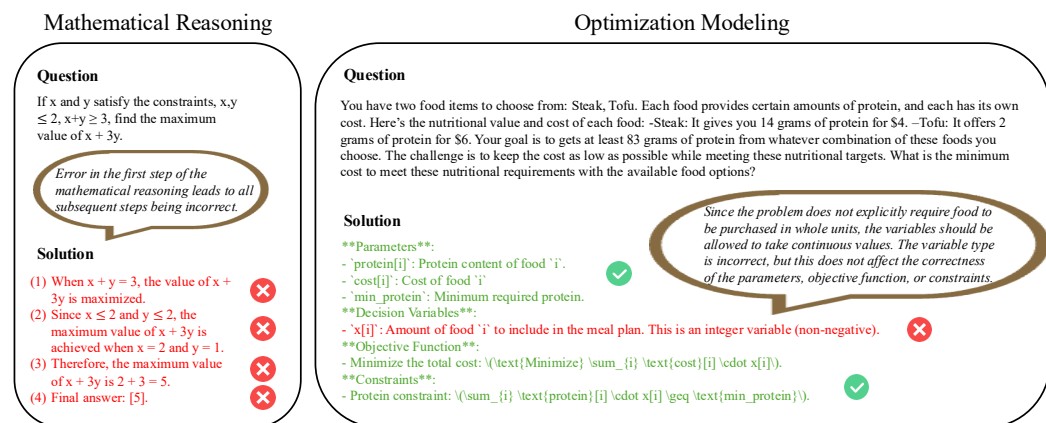

Figure 1: Illustration of the difference between mathematical reasoning and optimization modeling.

Recently, many general post-training techniques have been successfully adapted to improve the performance of automated optimization modeling. A range of studies, such as ORLM (Huang et al., 2025), ReSocratic (Yang et al., 2024), Step-Opt (Wu et al., 2025) and OptMATH (Lu et al., 2025), adopt the paradigm of first synthesizing new data and subsequently fine-tuning models on the generated data. Another line of research, including LLMOPT (Ethayarajh et al., 2024) and SIRL (Chen et al., 2023), explores the adaptation of reinforcement learning methods to this domain. For instance, SIRL introduces partial KL regularization and leverages solver feedback as a reward signal to update the model. A distinct line of methods focuses on test-time scaling (TTS), which effectively enhances model performance at inference without modifying the underlying parameters. Within this line, Chain-of-Experts (Xiao et al., 2023) and OptiMUS (AhmadiTeshnizi et al., 2023) explore multi-agent systems, whereas Autoformulator (Astorga et al., 2024) leverages Monte-Carlo Tree Search. However, progress in this field remains constrained by two major challenges: (1) High cost of generating high-quality data. Existing methods rely heavily on seed data and demonstrate limited generalization beyond the scope of that data. (2) Sparse reward signals. Representative approaches, such as SIRL, primarily use the correctness of the final outcome as the reward signal, which tends to be sparse, particularly for difficult problems. However, our insight reveals that LLMs typically make errors only within a limited subset of optimization modeling formulations—such as those involving variables, constraints, or objectives—rather than across all components (as illustrated in Fig 1). This observation suggests that the formulation of these factors exhibits relative independence, which in turn motivates us to exploit this characteristic in both the data generation and training stages.

In this work, we propose a novel error-driven learning framework—namely, auto**m**ated opt**i**mization modeli**n**g via a localizable error-**d**riven perspective (MIND) to address the aforementioned challenges. Specifically, MIND is a two-stage framework: (1) Motivated by our key observation of the unique localizable patterns in error propagation of optimization modeling, we propose an error-driven reverse data synthesis pipeline to construct a focused, high-density training corpus, MIND-Train, which captures common error patterns to support the post-training pipeline; (2) To mitigate the sparse reward problem arising from the limited capacity of the base model on difficult problems, we introduce a novel **D**ynamic Supervised **F**ine-tuning **P**olicy **O**ptimization method (DFPO) that dynamically corrects wrong responses while generating corrected responses that remain close to the distribution of the base model's responses during the training stage. By leveraging this slight distributional discrepancy, we integrate the supervised fine-tuning (SFT) and reinforcement learning (RL) in a novel, stable, and effective manner for automated optimization modeling.

Our contributions are summarized as follows: (1) *Conceptually*, through extensive empirical analysis, we observe a low error ratio in automated optimization modeling, highlighting a key difference from general mathematical problems. (2) *Methodologically*, we propose a novel error-driven learning framework to customize the entire model training framework from data synthesis to post-training to address two challenges in automated optimization modeling: the sparsity of error-specific problems and the scarcity of learning signals on difficult problems. (3) *Experimentally*, we evaluate MIND on six benchmarks, demonstrating that it outperforms state-of-the-art automatiza-

tion modeling methods. (4) *From a data perspective*, we open-source a new training dataset, MIND-Train, and a new benchmark, MIND-Bench, for the automated optimization research community.

## 2 RELATED WORK

**Domain-specific Data Synthesis and Augmentation** Recently, data generation methods have followed two main directions: data augmentation, which enhances existing samples through transformations (including data labeling (Khan et al., 2023), data reformation (Dunlap et al., 2023), and co-annotation (Li et al., 2023a)), and data synthesis, which creates entirely new samples either from scratch or using generative models. With the advancements of LLMs (Brown et al., 2020), data synthesis has made significant strides in both the quality and efficiency of synthetic data generation. General model distillation (Chen et al., 2023; Eldan & Li, 2023; Li et al., 2023b), domain model distillation (Lewkowycz et al., 2022; Luo et al., 2023), and model self-improvement (Maini et al., 2024; Wang et al., 2022; Zelikman et al., 2022) have emerged as mainstream data synthesis methods. Benefiting from verifiable outputs, data synthesis methods in mathematics, such as those in (Zelikman et al., 2022; Luo et al., 2023), generate diverse questions, answers, and more rationale corpora, which are preserved after verification. Similar to general mathematics, optimization modeling can also be verified using an optimizer solver. There are three common data synthesis and augmentation methods in this domain. ORLM (Huang et al., 2025) applies data augmentation to transform existing automated modeling instances and utilizes forward data synthesis to rephrase questions, subsequently employing LLMs to generate corresponding mathematical formulations. Step-Opt (Wu et al., 2025) employs iterative problem generation, evolving both complexity and scope, to systematically and effectively augment existing datasets. In contrast, Resocratic (Yang et al., 2024) proposes a reverse data synthesis approach that rephrases formulations and then leverages LLMs to generate the corresponding questions. Combining these methods, OptMATH (Lu et al., 2025) introduces bidirectional data synthesis, which first rephrases mathematical formulations, then uses LLMs to generate questions, and finally applies LLMs again to produce mathematical formulations. The two sets of mathematical formulations are then compared to ensure data quality. Although these data synthesis and augmentation methods have successfully applied general data synthesis and augmentation techniques to the automated modeling domain, they overlook the unique characteristics of automated optimization modeling data. This gap motivates the development of MIND.

**Domain-specific Post-Training** The predominant post-training techniques can be broadly categorized into fine-tuning (Ouyang et al., 2022; Lester et al., 2021; Luong et al., 2024), alignment (Kaufmann et al., 2024; Bai et al., 2022; Rafailov et al., 2023), and reasoning (Gou et al., 2023; Jaech et al., 2024; Guo et al., 2025). By leveraging the verifiable answer characteristics in mathematics (Hu et al., 2025) and code generation (Luo et al., 2025), Reinforcement Learning with Verifiable Rewards (RLVR) has made significant progress in addressing these complex reasoning problems. The success of the representative RLVR method Group Relative Policy Optimization (GRPO) (Shao et al., 2024) has inspired increasing research on improving RLVR methods through techniques such as normalization, clipping, data filtering, and loss aggregation. Compared to Proximal Policy Optimization (PPO) (Schulman et al., 2017), GRPO (Shao et al., 2024) computes response-level advantages for prompts within a group, replacing the value function used in PPO to improve training efficiency. Based on GRPO, Decoupled Clip and Dynamic Sampling Policy Optimization (DAPO) (Yu et al., 2025) introduce four curated tricks: it decouples the upper and lower clipping ranges to encourage exploration and prevent entropy collapse, dynamically filters out samples where all responses are correct or incorrect to improve training efficiency and stability, aggregates losses at the token level to better handle long responses, and applies special reward shaping to control overlong or truncated responses. To address the training instability and inefficiency of the RLVR method, Guided Hybrid Policy Optimization (GHPO) (Liu et al., 2025) explores the use of hints extracted from the ground-truth solution during the reinforcement learning process. Unlike these approaches, Value-model-based Augmented Proximal Policy Optimization (VAPO) (Yue et al., 2025) uses a value-model-based RLVR method and adds a negative log-likelihood loss for correctly sampled outcomes. Within the vertical domain of automated optimization modeling, ORLM (Huang et al., 2025), Step-Opt (Wu et al., 2025), Resocratic (Yang et al., 2024), and OptMATH (Lu et al., 2025) investigate supervised fine-tuning (SFT), LLMOPT (Jiang et al., 2024) explores Kahneman–Tversky Optimization, and SIRL (Chen et al., 2025) examines RLVR. Although these approaches apply general post-training techniques to automated optimization modeling, they overlook its unique characteristics.

Building on the progress of these methods, we emphasize that RLVR can effectively bridge the gap between general-purpose LLMs and the specific requirements of automated optimization modeling from an error-driven perspective.

# 3 PRELIMINARIES

## 3.1 AUTOMATED OPTIMIZATION MODELING

In general, optimization modeling entails a complex chain-of-thought (Wei et al., 2022), including problem analysis, extraction of key information to build a rationale, formulation of a mathematical model with variables, objective functions, and constraints, followed by translation into executable code. An automated optimization modeling instance is defined as a tuple $(q, o, a)$, where $q$ denotes the natural language description of the question, $o$ represents the corresponding reasoning path consisting of the rationale $\mathcal{Z}$, mathematical formulation $\mathcal{MF}$, and executable code $\mathcal{C}$, and $a$ is the resulting objective value. Thus, the corresponding training instance is expressed as $(q, a^*)$, where $a^*$ denotes the ground-truth objective of $q$. The problem of automated optimization modeling is to transform $q$ into $o$, such that an optimization solver can execute the code $\mathcal{C}$ contained in $o$ to compute the objective value $a$. The goal is to find a reasoning path $o$ that yields an objective value $a$ matching the ground-truth objective $a^*$, thereby corresponding to the correct optimization modeling. We formulate the automated optimization modeling problem as follows:

$$\max_{\theta} \mathbb{E}_{(q,a^*)\sim\mathcal{D}, o\sim\pi_{\theta}(\cdot|q), a\sim\text{BS}(o)} \left[ R(a, a^*) \right], \tag{1}$$

where $\mathcal{D}$, $\theta$ and BS denote the training dataset, the parameters of the target policy $\pi_{\theta}$ and the backbone solver, respectively. Given a question $q$, the policy $\pi_{\theta}$ produces a reasoning path $o$. The backbone solver, such as PySCIPOpt, takes the reasoning path $o$ as input, extracts the corresponding executable code $\mathcal{C}$, and outputs the objective value $a$. Finally, $a$ is compared with the ground truth $a^*$ to compute the reward $R$.

## 3.2 PRELIMINARY RESULTS

The automated optimization modeling task involves generating mathematical formulations that typically consist of <VARIABLES, CONSTRAINTS, OBJECTIVES>. To investigate how and where errors occur, we conducted preliminary experiments using the base model Qwen-2.5-7B-Instruct (Yang et al., 2025) on the ORLM training dataset (Huang et al., 2025), which contains questions paired with their correct mathematical formulations. For each question, we compare the generated code against the ground-truth mathematical formulation using an LLM-as-a-judge approach to identify errors in the variables, constraints, and objectives. We define the error ratio $\mathcal{E}$ of each instance as $\frac{N_{err\_var}+N_{err\_con}+N_{err\_obj}}{N_{var}+N_{con}+N_{obj}}$, where $N(\cdot)$ is the

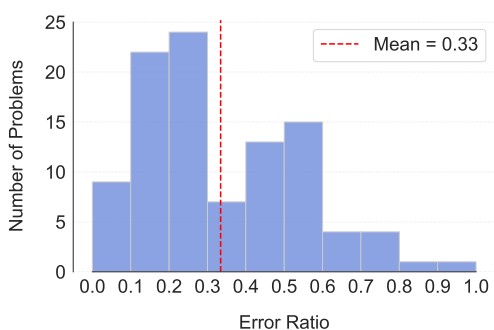

Figure 2: Distribution of error ratio across 100 incorrect generation results for Qwen2.5-7B-Instruct.

number of the corresponding component. As shown in Figure 2, when errors occur, LLMs tend to introduce only a small fraction of errors rather than producing entirely incorrect formulations in most cases. The low average error ratio of 0.33 indicates that the variables, constraints, and objectives are relatively independent, thus limiting the error propagation. Additionally, we observed that certain types of errors are more likely to occur in specific components of the formulation. For instance, when modeling variables, LLMs often struggle to determine the appropriate data type (e.g., integer or continuous). As shown in Figure 1, we illustrate the difference in error propagation between a general mathematical reasoning question and an optimization modeling question. This observation motivates us to systematically collect the most frequent error types from existing datasets and then synthesize new data that explicitly incorporates these common error patterns.

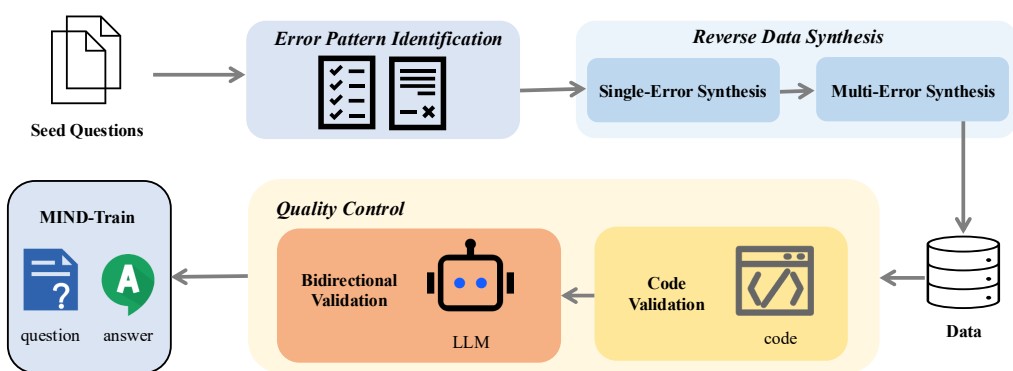

Figure 3: Overview of our proposed data synthesis pipeline.

## 4 METHODOLOGY

### 4.1 MIND: ERROR-DRIVEN REVERSE DATA SYNTHESIS PIPELINE

Motivated by our observations, we propose an error-driven reverse data synthesis pipeline, as illustrated in Figure 3. Our data generation process differs from prior work (Huang et al., 2024; Yang et al., 2024) in two key aspects: (1) we skip the costly collection of high-quality seed data by directly leveraging existing optimization modeling datasets as seeds; and (2) we deliberately target common error patterns that LLMs are prone to, thereby producing synthesized data that is inherently more challenging and better suited for robust model training. Our synthesis pipeline consists of three stages, including error pattern identification, reverse data synthesis, and quality control.

**Error Pattern Identification** Since our pipeline requires LLMs to make errors on the problems, we sample seed data from existing optimization modeling training datasets, namely OR-Instruct-Data-3K (Huang et al., 2025) and OptMATH-Train (Lu et al., 2025). We then apply our base model to perform the reasoning process on this seed data and extract error patterns by comparing the generated code with the corresponding ground-truth formulations. The error pattern identification and extraction are accomplished by powerful LLMs such as DeepSeek-R1 (Guo et al., 2025).

**Reverse Data Synthesis** After identifying the error patterns, we evolve the original questions into new ones by systematically incorporating these patterns. Since each question may contain multiple error types, we design two complementary strategies: single-error reverse data synthesis, where the LLM is instructed to focus on a single error pattern and generate a new problem that deliberately embeds a trap at that specific point (See example in Figure 4); and multi-error reverse data synthesis, which seeks to construct more challenging problems containing multiple potential error-prone points (See example in Figure 7). Notably, LLMs are instructed to output not only the new problem but also its corresponding modeling solution.

**Quality Control** To ensure the quality of the generated data, we implement a two-stage quality control process: (1) *Code validation*: We employ the target solver to verify the executability of the generated code and retain only those instances that can be successfully executed and solved to yield a reasonable solution (e.g., non-zero optimal value). (2) *Bidirectional validation*: Since both the problem and its solution are evolved from the original question, we further employ another powerful LLM to directly solve the newly synthesized problem and compare the obtained optimal value against the ground-truth value in the synthesized dataset. Only the instances that pass this bidirectional validation are retained.

We highlight that our reverse data synthesis method can leverage error patterns from different training datasets or industry scenario problems to generate diverse and challenging data. This approach significantly reduces the reliance on costly expert annotations for seed data, thereby improving both scalability and practicality.

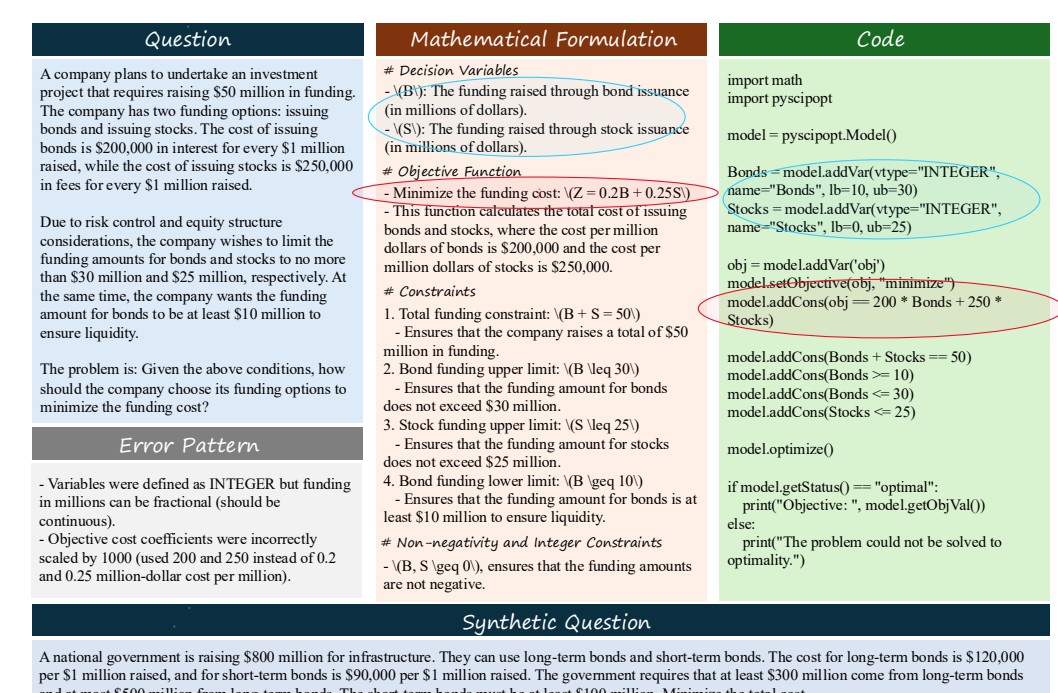

Figure 4: Example on single-error reverse data synthesis.

## 4.2 DYNAMIC SUPERVISED FINE-TUNING POLICY OPTIMIZATION

Existing approaches such as DAPO (Yu et al., 2025), SIRL (Chen et al., 2025) and GHPO (Liu et al., 2025) have sought to address the sparse reward problem on difficult samples through techniques like dynamic sampling, curated reward design, and adaptive prompt guidance. However, we argue that these methods still suffer from critical limitations, including insufficient guidance and distribution shifting on the policy model. To mitigate these challenges, we propose a novel framework termed **D**ynamic Supervised **F**ine-Tuning **P**olicy **O**ptimization (DFPO).

**Reward Design**   We define modeling fidelity as the extent to which a mathematical formulation accurately represents the optimization problem it is intended to model. It is measured as the distance between the predicted formulation and the correct formulation, denoted by $\mathcal{E}$ (see Section 3.2 for details). Objective accuracy represents the distance between the formulation's objective value and the ground-truth objective value. We hypothesize that, in general, higher fidelity in the mathematical formulation of an optimization problem is associated with more accurate objective values. Let $\mathcal{MF}_\theta$ denote the predicted mathematical formulation based on parameters $\theta$, and let $\mathcal{MF}^*$ denote the one correct mathematical formulation. We introduce a modeling error measure, $\mathcal{E}(\mathcal{MF}_\theta, \mathcal{MF}^*)$, which captures discrepancies in variables, constraints, and the objective function. A larger $\mathcal{E}$ indicates greater deviation from the correct mathematical formulation. Our working hypothesis is that optimization modeling error and objective deviation are positively correlated in expectation. Formally, for two predicted problems $\mathcal{MF}_\theta^{(1)}$ and $\mathcal{MF}_\theta^{(2)}$, we generally expect:

$$\text{if } \mathcal{E}\left(\mathcal{MF}_\theta^{(1)}, \mathcal{MF}^*\right) < \mathcal{E}\left(\mathcal{MF}_\theta^{(2)}, \mathcal{MF}^*\right),$$
$$\text{then } \mathbb{E}\left[\left|\text{Obj}\left(\mathcal{MF}_\theta^{(1)}\right) - \text{Obj}(\mathcal{MF}^*)\right|\right] \leq \mathbb{E}\left[\left|\text{Obj}\left(\mathcal{MF}_\theta^{(2)}\right) - \text{Obj}(\mathcal{MF}^*)\right|\right], \quad (2)$$

where $\text{Obj}(\mathcal{MF})$ denotes the objective value of the mathematical formulation $\mathcal{MF}$. This assumption underpins our reward design: by rewarding the agent based on the degree of modeling errors, we enable it to perceive the extent of such errors, thereby guiding the solutions to be structurally closer to the correct mathematical formulation.

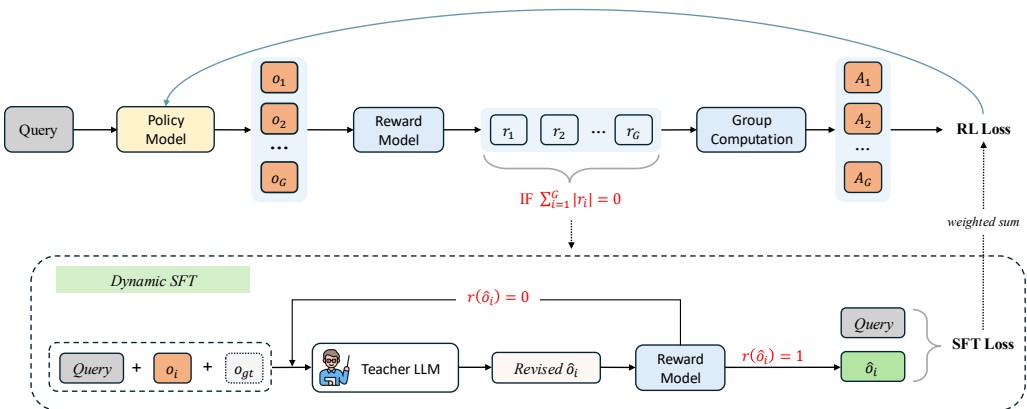

Figure 5: Overview of our proposed post-training method.

Therefore, we present our reward function as follows:

$$R = \alpha \cdot R_{fid} + (1 - \alpha) \cdot R_{acc},$$

where $\alpha = 0.2$, the modeling fidelity reward is defined as $R_{\text{fid}} = 1 - \frac{|obj_{\text{MIND}} - obj_{\text{GT}}|}{max(|obj_{\text{MIND}}|, |obj_{\text{GT}}|)}$, and the accuracy reward as

$$R_{\text{acc}} = \begin{cases} 1, & \text{if the answer is right,} \\ 0, & \text{otherwise.} \end{cases}$$

In this way, we mitigate the sparse reward problem by introducing a fidelity score, which provides partial credit when the generated mathematical formulation is close to, but not exactly identical to, the ground truth—a situation that accounts for the majority of cases.

**Dynamic Supervised Fine-Tuning Policy Optimization** Standard GRPO and DAPO algorithms suffer from the sparse reward problem when dealing with difficult tasks, as they either perform inefficient explorations or discard unsuccessful samples. A straightforward remedy is to replace an incorrect rollout with the ground-truth solution or to provide partial solutions as hints when all rollouts fail, thereby alleviating the sparse reward issue. However, we contend that this approach still faces notable limitations: (1) not all ground-truth labels include the intermediate reasoning process, which is essential for fostering reasoning capabilities; and (2) the ground-truth solutions do not always align with the output distribution of the current policy model, making it difficult for the model to directly imitate the labeled behavior. We refer to this challenge as *distributional shifting*. To overcome these limitations, we introduce Dynamic Supervised Fine-Tuning Policy Optimization (DFPO). Unlike existing methods, as illustrated in Figure 5, our approach leverages a stronger teacher LLM (e.g., DeepSeek-V3 (Liu et al., 2024)) to refine the base model's incorrect responses, thereby ensuring that the corrected outputs remain closely aligned with the response distribution of the base model (see examples in Appendix C.6). To enhance the reliability of this correction process, we provide the teacher LLM with access to the ground-truth solution. Once the teacher LLM generates a corrected response that is both accurate and distributionally consistent with the original incorrect rollout, this response is incorporated into the training process by computing the SFT loss. Finally, both the standard RL loss and the SFT loss are jointly utilized to guide the optimization of the policy model.

$$\mathcal{L}_{\text{RL}}(\theta) = -\mathbb{E}_{(q,a^*)\sim\mathcal{D},\{o_i\}_{i=1}^G\sim\pi_{\theta_{\text{old}}}(\cdot|q)} \left[ \frac{1}{\sum_{i=1}^G |o_i|} \sum_{i=1}^G \sum_{t=1}^{|o_i|} \min\left( r_{i,t}(\theta)\hat{A}_{i,t}, \right.\right.$$

$$\left.\left. \text{clip}\left(r_{i,t}(\theta), 1 - \varepsilon_{\text{low}}, 1 + \varepsilon_{\text{high}}\right)\hat{A}_{i,t}\right)\right] \quad (3)$$

$$\text{s.t.} \ 0 < |\{o_i|\text{is\_equivalent}(a^*, \text{BS}(o_i))\}| < \gamma \times G,$$

$$\mathcal{L}_{\text{NLL}}(\theta) = -\mathbb{E}_{(q,a^*)\sim\mathcal{D}, \{o_i\}_{i=1}^G \sim \pi_{\theta_{\text{old}}}(\cdot|q), \hat{o}_i \sim \pi_{\text{teacher}}(\cdot|q,\{o_i\}_{i=1}^G, o_{\text{gt}})} \left[ \sum_{t=1}^{|\hat{o}_i|} \log \pi_\theta(a_t \mid s_t) \right] \quad (4)$$

$$\text{s.t. } |\{o_i | \text{is\_equivalent}(a^*, \text{BS}(o_i))\}| = 0.$$

$$\mathcal{L}_{\text{DFPO}}(\theta) = \mathcal{L}_{\text{RL}}(\theta) + \beta \cdot \sqrt{\frac{n_{\text{SFT}}}{n_{\text{RL}}}} \cdot \mathcal{L}_{\text{NLL}}(\theta), \quad (5)$$

where $n_{\text{SFT}}$ and $n_{\text{RL}}$ denote the numbers of SFT and RL responses in each training batch.

## 5 EXPERIMENTS

We conduct extensive experiments to study the effectiveness of our proposed MIND on automated optimization modeling. We aim to study the following research questions (RQ):

**RQ1** Can MIND improve the base model's performance in automated optimization modeling?

**RQ2** How does MIND compare with state-of-the-art automated optimization methods?

**RQ3** How effective is the proposed error-driven reverse data synthesis pipeline?

**RQ4** How effective is the proposed error-driven DFPO post-training method?

**RQ5** Can MIND generalize to out-of-distribution automated optimization modeling scenarios?

### 5.1 EXPERIMENTAL SETUP

Following existing work (Huang et al., 2025; Chen et al., 2025), Qwen-2.5-7B-Instruct is employed as our base model. To further align with recent advances, we also adopt Qwen3-8B, a newly released and widely adopted open-source model, as an additional base model. We construct the MIND-Train dataset by synthesizing data from the seed datasets OR-Instruct-Data-3K (Huang et al., 2025) and OptMATH-Train (Lu et al., 2025). We note that Qwen2.5-7B-Instruct is specifically used for the error pattern identification stage in the data synthesis pipeline. A detailed summary of MIND-Train is provided in Appendix A.3. Finally, we sample 10,000 instances for training.

**Benchmarks & Baselines** We conduct comprehensive evaluations on NL4Opt (Ramamonjison et al., 2023), IndustryOR (Huang et al., 2025), MAMO (Huang et al., 2024) (EasyLP and ComplexLP), OptMATH-Bench (Lu et al., 2025), and OptiBench (Yang et al., 2024). Further details on the benchmarks can be found in Appendix A.1. We compare our method against GPT-4 (Achiam et al., 2023), OpenAI o3 (Jaech et al., 2024), Deepseek-V3 (Liu et al., 2024), Deepseek-R1 (Guo et al., 2025), Qwen2.5-7B-Instruct (Yang et al., 2025), Qwen3-8B (Yang et al., 2025), Autoformulator (Astorga et al., 2024), Chain-of-Experts (Xiao et al., 2023), Step-Opt (Wu et al., 2025), OptiMUS (AhmadiTeshnizi et al., 2023), ORLM (Huang et al., 2025), LLMOPT (Jiang et al., 2024), OptMATH (Lu et al., 2025), and SIRL (Chen et al., 2025).

**Evaluation and Metrics** Following previous work, we evaluate all methods with pass@1 accuracy in a zero-shot setting. A solution is deemed correct if the relative error between the objective value produced by the LLM and the ground-truth objective value is less than $10^{-6}$.

### 5.2 MAIN RESULTS

***RQ1: MIND consistently improves automated modeling performance.*** As shown in Table 1, MIND-Qwen2.5-7B enhances the base model's automated modeling performance by approximately 14.3% across six benchmarks. On relatively simpler benchmarks such as NL4Opt and EasyLP, MIND yields moderate improvements over already strong baseline scores. In contrast, on more challenging benchmarks such as IndustryOR, ComplexLP, and OptMATH, the performance gains are significant, with an average improvement of 24.1%. Moreover, we observe that on OptiBench, which primarily

consists of tabular data, MIND-Qwen2.5-7B achieves only marginal improvement, likely due to the limited representation of similar problem types in the training dataset. Furthermore, MIND-Qwen3-8B enhances its base model's performance by approximately 31.0%, providing additional evidence that MIND is effective across different base model architectures.

Table 1: Performance comparison of models on benchmarks (pass@1↑). Methods marked with * indicate that their results are taken from the original or reproduced papers.

| Category | Methods | NL4Opt | IndustryOR | EasyLP | ComplexLP | OptMATH | OptiBench | Macro AVG |
|---|---|---|---|---|---|---|---|---|
| **Proprietary** | GPT-4* | 89.0% | 33.0% | 87.3% | 49.3% | 16.6% | 68.6% | 57.4% |
| | OpenAI o3* | 69.4% | 44.0% | 77.1% | 51.2% | 44.0% | 58.6% | 57.4% |
| **Open-Source** | Deepseek-V3* | 95.9% | 37.0% | 88.3% | 50.2% | 44.0% | 71.6% | 64.5% |
| | Deepseek-R1* | 82.4% | 45.0% | 87.2% | 67.9% | 40.4% | 66.4% | 61.9% |
| | Qwen2.5-7B-Instruct | 89.0% | 24.0% | 89.4% | 31.5% | 3.0% | 53.2% | 48.4% |
| | Qwen3-8B | 72.2% | 14.0% | 76.8% | 17.2% | 7.2% | 36.5% | 37.3% |
| **TTS-based** | Autoformulator* | 92.6% | 48.0% | - | 62.3% | - | - | - |
| | Chain-of-Experts* | 64.2% | - | - | 40.2% | - | - | - |
| | OptiMUS* | 78.8% | 31.0% | 77.0% | 43.6% | 20.2% | 45.8% | 49.4% |
| **Fine-Tuning** | ORLM-Llama3-8B* | 85.7% | 24.0% | 82.3% | 37.4% | 2.6% | 51.1% | 47.2% |
| | Step-Opt-Llama3-8B* | 84.5% | 36.4% | 85.3% | 61.6% | - | - | - |
| | LLMOPT-Qwen2.5-14B* | 80.3% | 29.0% | 89.5% | 44.1% | 12.5% | 53.8% | 51.1% |
| | OptMATH-Qwen2.5-7B* | 94.7% | 20.0% | 86.5% | 51.2% | 24.4% | 57.9% | 55.8% |
| | OptMATH-Qwen2.5-32B* | 95.9% | 31.0% | 89.9% | 54.1% | 34.7% | 66.1% | 62.0% |
| **RLVR** | SIRL-Qwen2.5-7B* | 96.3% | 33.0% | 91.7% | 51.7% | 30.5% | 58.0% | 60.2% |
| | SIRL-Qwen2.5-32B* | 98.0% | 42.0% | 94.6% | 61.1% | 45.8% | 67.4% | 68.2% |
| **Ours** | MIND-Qwen2.5-7B | 96.7% | 34.0% | 92.2% | 60.1% | 36.7% | 56.7% | 62.7% |
| | MIND-Qwen3-8B | 95.1% | 42.0% | 92.7% | 76.8% | 41.0% | 62.0% | 68.3% |

***RQ2: MIND outperforms state-of-the-art automated modeling methods.*** We compare MIND-Qwen2.5-7B and MIND-Qwen3-8B against a range of representative approaches, including proprietary models, agent-based frameworks, and training-based methods. As reported in Table 1, MIND-Qwen2.5-7B achieves superior average performance compared with all baseline models of comparable parameter size. In particular, relative to prior training-based approaches, MIND-Qwen2.5-7B demonstrates remarkable improvements on more challenging benchmarks such as IndustryOR, ComplexLP, and OptMATH. These results highlight the effectiveness of our reverse data synthesis pipeline and our proposed DFPO method. Furthermore, we observe that MIND-Qwen3-8B achieves competitive performance across all baselines, including larger models such as Deepseek-V3, GPT-4, OptMATH-Qwen2.5-32B, and SIRL-Qwen2.5-32B.

## 5.3 ABLATION STUDY

**Data Synthesis Framework (*RQ3*)** To assess the effectiveness of our data synthesis approach, we employ DAPO (Yu et al., 2025) to train Qwen-2.5-7B-Instruct from scratch on two datasets: OR-Instruct-Data-3K (3,000 instances) and MIND-3K (3,000 instances), where MIND-3K is generated from OR-Instruct-Data-3K using our proposed reverse data synthesis technique. As illustrated in Figure6, the model trained on MIND-3K achieves consistently higher accuracy gains as training progresses, indicating that our error-driven reverse data synthesis method yields superior sample efficiency. Furthermore, Table 2 reports a detailed performance comparison after seven training epochs across six benchmarks, showing that the model trained on MIND-3K outperforms its counterpart trained on the majority of benchmarks. (See the details of the ablation study on single-error and multi-error strategies in Appendix C.4).

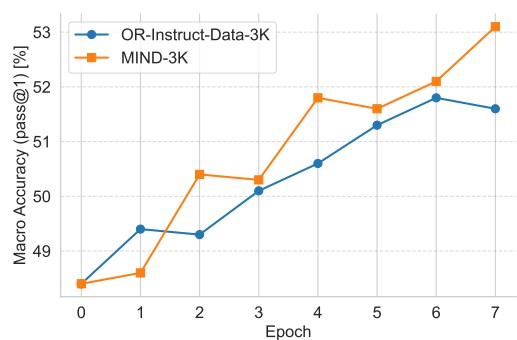

Figure 6: Ablation study of data synthesis methods across six benchmarks.

Table 2: Ablation results of the data synthesis pipeline on Qwen2.5-7B-Instruct (pass@1↑).

| Data | NL4OPT | IndustryOR | EasyLP | ComplexLP | OptMATH | OptiBench | Macro AVG |
|------|--------|------------|--------|-----------|---------|-----------|-----------|
| OR-Instruct-Data-3K | 93.9% | 25.0% | 90.7% | 35.0% | 10.8% | 54.4% | 51.6% |
| MIND-3K | 94.3% | 30.0% | 90.8% | 39.9% | 7.8% | 55.5% | 53.1% |

**Post-Training Framework (*RQ4*)**   To verify the effectiveness of DFPO, we compare it with DAPO, SFT, and SFT+GRPO under our reward design.  Both methods are trained on the same dataset of 10,000 instances (Table 1) and use the same chain-of-thought prompt. As shown in Table 3, DFPO outperforms DAPO by about 1.9% in macro-average accuracy across six benchmarks, with a notable gain of 10.2% on OptMATH. This demonstrates that DFPO provides more effective learning signals for difficult problems.  We highlight that while DAPO receives sufficient learning signals from easy problems through reinforcement learning, it receives limited signals from difficult problems. In contrast, DFPO leverages dynamic SFT techniques to capture additional learning signals from difficult problems, as evidenced by its improvement over DAPO in OptMATH. Furthermore, we observe that applying SFT alone on a relatively small training dataset (10,000 instances) does not yield significant performance gains. However, when used as a warm start for GRPO, the model achieves notable improvement, though it still lags behind DFPO on challenging benchmarks.

Table 3: Ablation results for post-training method on Qwen2.5-7B-Instruct (pass@1↑).

| Methods | NL4OPT | IndustryOR | EasyLP | ComplexLP | OptMATH | OptiBench | Macro AVG |
|---------|--------|------------|--------|-----------|---------|-----------|-----------|
| DFPO | 96.7% | 34.0% | 92.2% | 60.1% | 36.7% | 56.7% | 62.7% |
| DAPO | 96.7% | 33.0% | 92.5% | 58.6% | 26.5% | 57.5% | 60.8% |
| SFT | 92.2% | 31.0% | 85.4% | 37.4% | 9.6% | 55.9% | 51.9% |
| SFT+GRPO | 93.9% | 34.0% | 90.2% | 54.7% | 28.3% | 57.0% | 59.7% |

## 5.4 GENERALIZATION STUDY (*RQ5*)

In this paper, we introduce MIND-Bench, a benchmark that comprises 69 carefully curated operations research problems drawn from industry scenarios and textbooks (see Appendix A.4 for details). As shown in Table 4, MIND-Qwen2.5-7B demonstrates superior generalization on MIND-Bench compared with the state-of-the-art post-training model SIRL-Qwen2.5-7B, although it still lags behind the 671B-parameter foundation models Deepseek-V3 and Deepseek-R1. Furthermore, MIND-Qwen3-8B is competitive with Deepseek-V3, Deepseek-R1, and SIRL-Qwen2.5-32B, all of which have larger parameter sizes.

Table 4: Performance comparison of our proposed MIND and baselines on MIND-BENCH.

| Deepseek-V3 | Deepseek-R1 | Qwen2.5-7B-Instruct | Qwen3-8B |
|-------------|-------------|---------------------|----------|
| 66.7% | 75.4% | 29.0% | 27.5% |
| **SIRL-Qwen2.5-7B** | **SIRL-Qwen2.5-32B** | **MIND-Qwen2.5-7B** | **MIND-Qwen3-8B** |
| 46.4% | 65.2% | 50.7% | 68.1% |

## 6 CONCLUSION

In this paper, we empirically show that modeling errors are often localized within specific semantic segments. Motivated by this finding, we propose a novel error-driven learning framework, which customizes the whole model training framework from data synthesis to post-training. Our study highlights two key insights: (1) Data synthesis: Domain-specific LLM performance depends heavily on the diversity, quality, and quantity of training data. (2) Post-training: Due to the complexity of automated optimization modeling tasks, LLMs often struggle to receive sufficient learning signals through reinforcement learning alone on difficult problems. Together, these insights advance the understanding and development of LLMs for automated optimization modeling.

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

APPENDIX

## A DATASET

### A.1 BENCHMARK DATASET

**NL4Opt (Ramamonjison et al., 2023)** NL4OPT contains 245 high-quality questions, validated by (Lu et al., 2025). It includes only linear programming (LP) problems across various contexts. As the first curated dataset derived from the NL4OPT Competition, NL4OPT is considered an easy benchmark, featuring simple constraints and scenarios.

**MAMO (Huang et al., 2024)** MAMO consist of 642 high-quality questions in EasyLP and 203 high-quality questions in ComplexLP, as revised by (Chen et al., 2025). It focuses on linear programming (LP) and mixed-integer linear programming (MILP) problems. Compared with other benchmarks, MAMO primarily emphasizes LLM modeling skills on MILP, which constitutes the majority of real-world optimization problems.

**IndustryOR (Huang et al., 2025)** IndustryOR contains 100 questions collected from real-world optimization scenarios across various sectors, as verified by (Chen et al., 2025). It includes integer programming (IP), linear programming (LP), mixed-integer linear programming (MILP), and nonlinear programming (NLP), and other specialized formulations. Unlike other benchmarks, IndustryOR specifically targets industrial applications, capturing the complexity of real-world optimization problems.

**OptiBench (Yang et al., 2024)**    OptiBench contains 605 questions collected from textbooks (Bertsimas & Tsitsiklis, 1997; Conforti et al., 2014; Wolsey, 2020). It includes integer programming (IP), linear programming (LP), mixed-integer linear programming (MILP), and nonlinear programming (NLP). Compared with other benchmarks, OptiBench features extensive tabular data, enabling the evaluation of LLMs' ability to understand and reason with tables.

**OptMATH-Bench (Lu et al., 2025)**    OptMATH-Bench contains 166 carefully curated questions constructed by human experts. It includes integer programming (IP), linear programming (LP), mixed-integer linear programming (MILP), nonlinear programming (NLP), and second-order cone programming (SOCP). Compared with other benchmarks, OptMATH-Bench features longer natural language contexts and more complex constraints, enabling the evaluation of LLMs' long-context optimization modeling capacity.

OptMATH (Lu et al., 2025) and SIRL (Chen et al., 2025) highlight that portions of the problem statements in benchmarks contain ambiguities, making it difficult for both LLMs and human experts to determine whether a variable should be treated as integer or continuous, depending on the practical context. Following their approach, we also adopt a rule-based substitution method. We consider a case as passed if the optimal solution, whether derived under the integer or continuous assumption, matches the ground truth, i.e., the objective absolute difference between the LLM-generated mathematical formulation and the ground-truth formulation is less than $10^{-6}$.

## A.2    SEED DATASET

**OR-Instruct-Data-3K**    OR-Instruct-Data-3K, released by ORLM, contains 3,000 training instances (a subset of the full 30,000 ORLM training examples), each including the `question`, `mathematical formulation`, and `code`.

**OptMATH-Train**    OptMATH-Train, released by OptMATH, contains 200,000 training instances, each including the `question`, `mathematical formulation`, and `code`.

## A.3    MIND-TRAIN DATASET

**MIND-Train Statistics**    As shown in Table 5, we provide statistical information for MIND-Train, summarizing the question examples across three stages of the reverse data synthesis pipeline. We present a multi-error reverse data synthesis example in Figure 7, complementing the single-error reverse data synthesis example (see Figure 4). We note that error pattern 1 comes from Figure 7, while error pattern 2 comes from Figure 4.

Table 5: MIND-Train dataset construction summary. The single-error strategy uses DeepSeek-R1-0528, while the multi-error strategy uses DeepSeek-V3.1-Think.

| Synthesis Category | Seed Data | Initial Count | Code | | Bidirectional | | Passed Rate |
|---|---|---|---|---|---|---|---|
| | | | Count | Rate | Count | Rate | |
| Single-Error | ORLM | 5033 | 5016 | 99.66% | 2007 | 40.01% | 39.88% |
| Multi-Error | ORLM | 2977 | 2910 | 97.75% | 1795 | 61.68% | 60.30% |
| Single-Error | OptMATH | 9676 | 5950 | 61.49% | 2961 | 49.76% | 30.60% |
| Multi-Error | OptMATH | 2850 | 2102 | 73.75% | 1494 | 71.07% | 52.42% |
| Multi-Error | ALL | 2473 | 1843 | 74.52% | 1406 | 76.29% | 56.85% |
| Total | - | 23009 | 17821 | 77.45% | 9663 | 54.22% | 42.00% |

**Word Cloud Analysis**    As shown in Figure 8, the word cloud highlights diverse automatic optimization modeling topics (e.g. hospital, transportation, machine, warehouse, surgery, facility, energy, product).

**Gerund Pairs Analysis**    As shown in Figure 9, we use en_core_web_sm (AI, 2023) to extract gerund pairs. The top 50 frequent gerund pairs represent typical optimization modeling patterns.

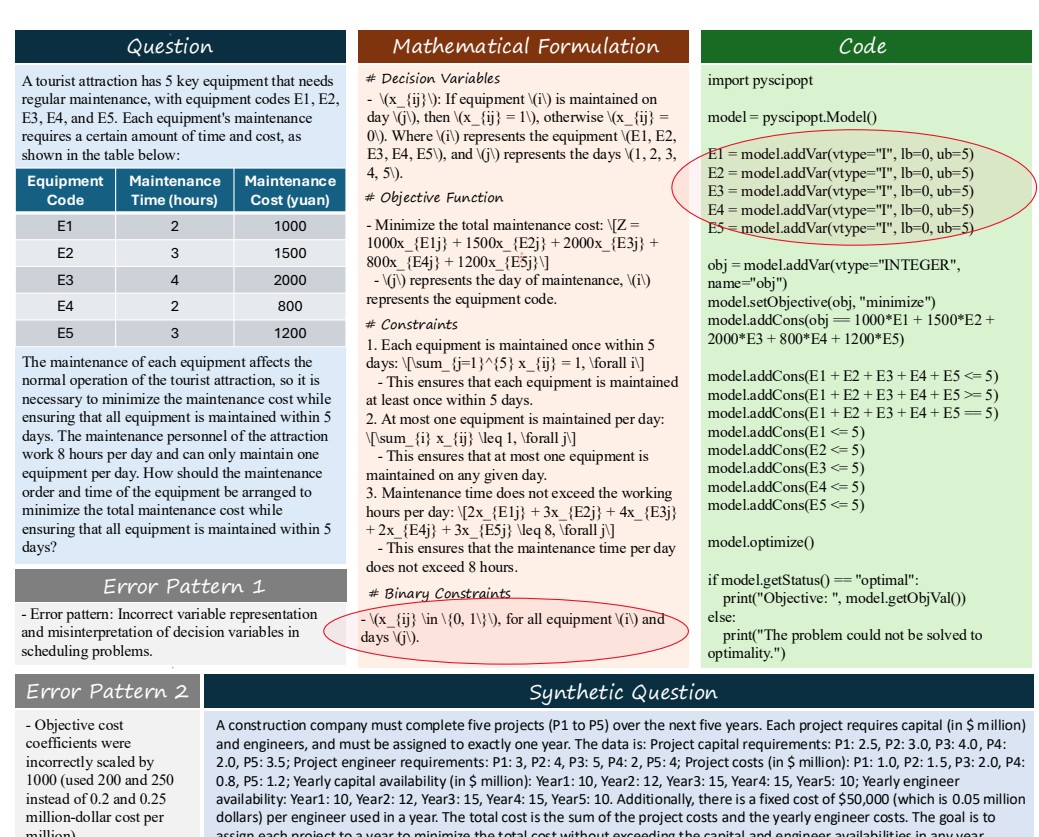

Figure 7: Example on multi-error reverse data synthesis.

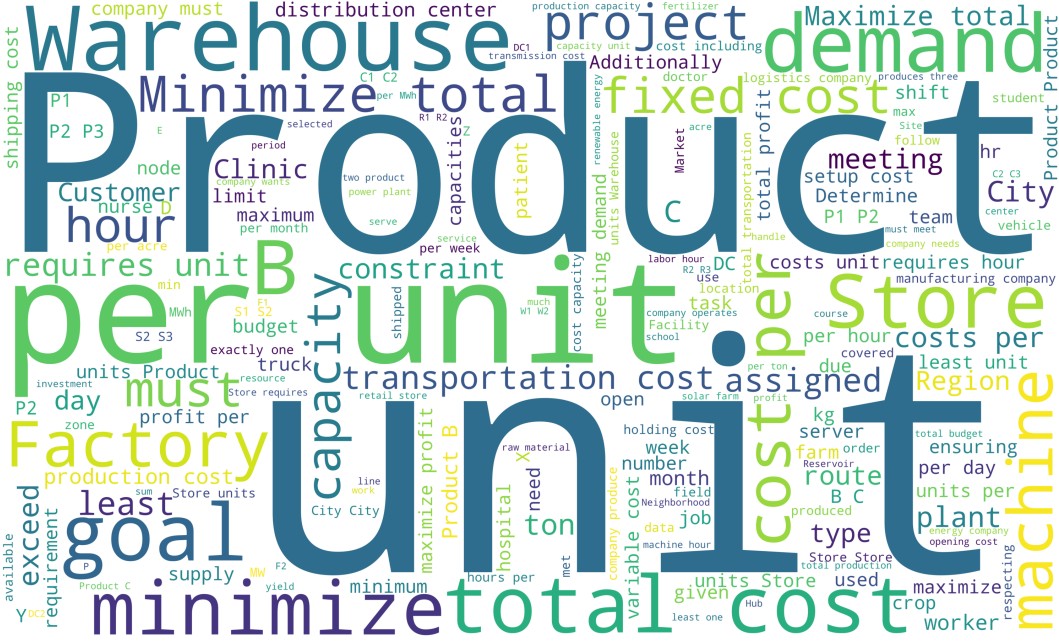

Figure 8: The statistical word cloud of MIND-Train.

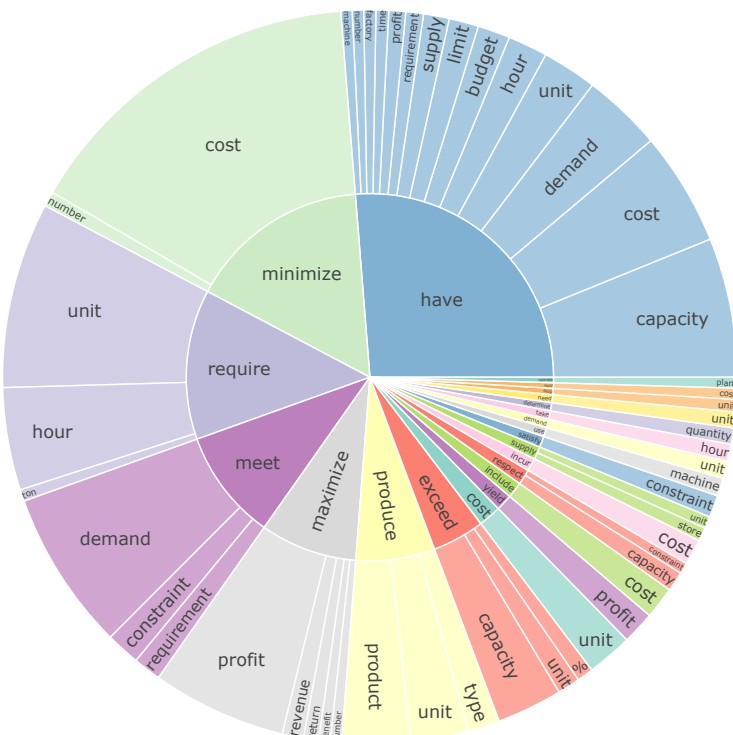

Figure 9: Top 50 gerund pairs of questions in MIND-Train.

**Length Distribution Analysis**    As shown in Figure 10, we examine the word length distributions of the prompts and responses in the training dataset (10,000 instances). The prompts exhibit an average length of 392 words, with most within the 200–600 word range. In comparison, responses are substantially longer, averaging 790 words, with the majority falling between 500 and 1,200 words.

To increase the diversity of the training dataset, we sample 5,000 instances from MIND-Train, 1,000 instances from OR-Instruct-Data-3K, and 4,000 instances from OptMATH-Train. In total, we use 10,000 instances to train the Qwen2.5-7B-Instruct.

## A.4    MIND-BENCH DATASET

To evaluate the generalization ability of LLMs, we carefully curated 69 questions derived from textbooks or industry scenarios (See details in Figure 11). These questions originate from out-of-distribution data sources that differ from those of other public benchmarks and training datasets. Examples of the questions are shown in Figure 12. For questions in MIND-Bench, there is no ambiguity regarding variable types, and we do not use a rule-based substitution method.

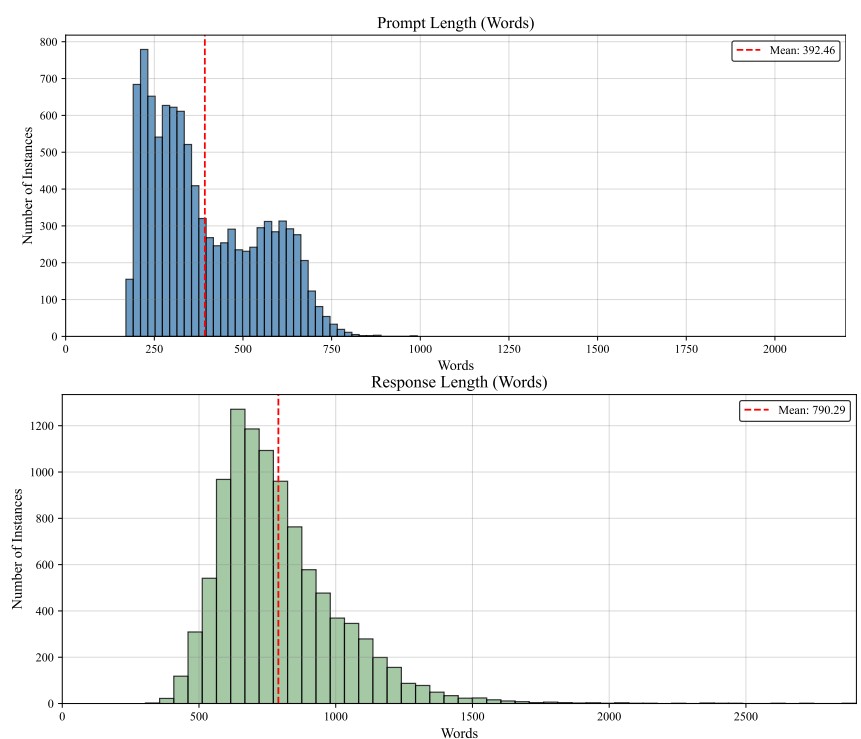

Figure 10: Length distribution of the training dataset for MIND-Qwen2.5-7B.

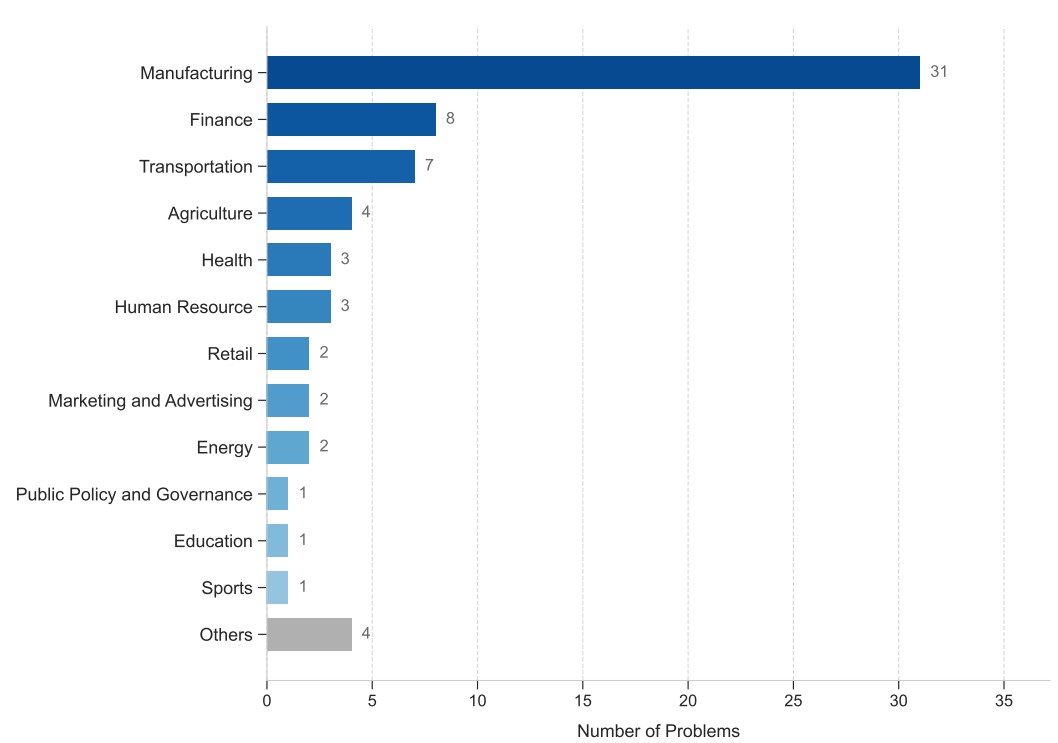

Figure 11: Scenario statistics of MIND-Bench.

### Question 1

A company is planning its production schedule over the next six months (it is currently the end of month 2). The demand (in units) for its product over that timescale is as shown below:\n\nMonth |3 |4 |5 |6| 7| 8|\n\n|:---:|:---:|:---:|:---:|:---:|:---:|\n|Demand |5000| 6000| 6500 |7000| 8000| 9500\n\nThe company currently has in stock: 1000 units which were produced in month 2; 2000 units which were produced in month 1; 500 units which were produced in month 0.\n\nThe company can only produce up to 8000 units per month and the managing director has stated that stocks must be built up to help meet demand in months 5, 6, 7 and 8. Each unit produced costs $15 and the cost of holding stock is estimated to be $0.75 per unit per month (based upon the stock held at the beginning of each month).\n\nThe company has a major problem with deterioration of stock in that the stock inspection which takes place at the end of each month regularly identifies ruined stock (costing the company $25 per unit). It is estimated that, on average, the stock inspection at the end of month t will show that 11% of the units in stock which were produced in month t are ruined; 47% of the units in stock which were produced in month t-1 are ruined; 100% of the units in stock which were produced in month t-2 are ruined. The stock inspection for month 2 is just about to take place.\n\nThe company wants a production plan for the next six months that avoids stockouts. Formulate their problem as a linear program.\n\nBecause of the stock deterioration problem the managing director is thinking of directing that customers should always be supplied with the oldest stock available. How would this affect your formulation of the problem?

### Question 2

Chip Green is the head groundskeeper at Birdie Valley Golf Club. For the mix of grass for the golf course, Chip has decided that the best fertilizer would be a 10-8-12 mixture. (Fertilizer is defined by three values: a, b and c where a is the percentage of nitrogen, b is the percentage of phosphorus, and c is the percentage of potash in the fertilizer. The remaining material is inert matter.) Chip can buy a 10-8-12 mix of fertilizer for $21.75 per 100 pounds, but there are other fertilizers on the market at a variety of prices. The chemical content of [nitrogen, phosphorus, potash] and prices are given below. Fertilizer 1: [10,8,12] for $21.75 per 100 pounds; Fertilizer 2: [8,11,15] for $23.75 per 100 pounds; Fertilizer 3: [12,7,12] for $22.00 per 100 pounds; Fertilizer 4: [10,10,10] for $19.50 per 100 pounds; Fertilizer 5: [15,10,6] for $18.50 per 100 pounds;. Chip would like to determine whether or not he could buy several fertilizers and mix them together to obtain a 10-8-12 mixture at a lower cost than $21.75 per 100 pounds. Recognizing that it might be impossible to obtain an exact 10-8-12 mix from the fertilizers, Chip is willing to accept chemical percentages of at least the target amounts, but no more than 0.5% above them (so the nitrogen level should be between 10% and 10.5%; the phosphorus level should be between 8% and 8.5%; the potash level should be between 12% and 12.5%).

### Question 3

A chocolate maker has contracted to operate a small candy counter in a fashionable store. To start with, the selection of offerings will be intentionally limited. The counter will offer a regular mix of candy made up of equal parts of cashews, raisins, caramels, and chocolates, and a deluxe mix that is one-half cashews and one-half chocolates, which will be sold in one-pound boxes. In addition, the candy counter will offer individual one-pound boxes of cashews, raisins, caramels, and chocolates. A major attraction of the candy counter is that all candies are made fresh at the counter. However, storage space for supplies and ingredients is limited. Bins are available that can hold the amounts shown in the table.\n\n| Ingredient | Capacity (pounds per day) |\n|:---:|:---|\n| Cashews | 120 |\n| Raisins | 200 |\n| Caramels | 100 |\n| Chocolates | 160 |\n\nIn order to present a good image and to encourage purchases, the counter will make at least 20 boxes of each type of product each day. Any leftover boxes at the end of the day will be removed and given to a nearby nursing home for goodwill. The profit per box for the various items has been determined as follows.\n\n| Item | Profit per Box |\n|:---:|:---:|\n| Regular | $0.80 |\n| Deluxe | $0.90 |\n| Cashews | $0.70 |\n| Raisins | $0.60 |\n| Caramels | $0.50 |\n| Chocolates | $0.75 |\n\nSolve for the optimal values of the decision variables and the maximum profit.

### Question 4

International Wool Company operates a large farm on which sheep are raised. The farm manager determined that for the sheep to grow in the desired fashion, they need at least minimum amounts of four nutrients (the nutrients are nontoxic so the sheep can consume more than the minimum without harm). The manager is considering three different grains to feed the sheep. The table below lists the number of units of each nutrient in each pound of grain, the minimum daily requirements of each nutrient for each sheep, and the cost of each grain. The manager believes that as long as a sheep receives the minimum daily amount of each nutrient, it will be healthy and produce a standard amount of wool. The manager wants to raise the sheep at minimum cost.\n\n| | Grain 1 | Grain 2 | Grain 3 | Minimum Daily Requirement (units) |\n|:---:|:---:|:---:|:---:|:---:|\n| Nutrient A | 20 | 30 | 70 | 110 |\n| Nutrient B | 10 | 10 | 0 | 18 |\n| Nutrient C | 50 | 30 | 0 | 90 |\n| Nutrient D | 6 | 2.5 | 10 | 14 |\n| Cost ($/lb) | 41 | 36 | 96 | |\n\n

### Question 5

Dorian Auto has a $20,000 advertising budget. Dorian can purchase full-page ads in two magazines: Inside Jocks (IJ) and Family Square (FS). An exposure occurs when a person reads a Dorian Auto ad for the first time. The number of exposures generated by each ad in IJ is as follows: ads 1-6, 10,000 exposures; ads 7-10, 3,000 exposures; ads 11-15, 2,500 exposures; ads 16+, 0 exposures. For example, 8 ads in IJ would generate 6(10,000) + 2(3,000) = 66,000 exposures. The number of exposures generated by each ad in FS is as follows: ads 1-4, 8,000 exposures; ads 5-12, 6,000 exposures; ads 13-15, 2,000 exposures; ads 16+, 0 exposures. Thus, 13 ads in FS would generate 4(8,000) + 8(6,000) + 1(2,000) = 82,000 exposures. Each full-page ad in either magazine costs $1,000. Assume there is no overlap in the readership of the two magazines. Formulate an IP to maximize the number of exposures that Dorian can obtain with limited advertising funds.

### Question 6

A company is considering opening warehouses in four cities: New York, Los Angeles, Chicago, and Atlanta. Each warehouse can ship 100 units per week. The weekly fixed cost of keeping each warehouse open is $400 for New York, $500 for Los Angeles, $300 for Chicago, and $150 for Atlanta. Region 1 of the country requires 80 units per week, region 2 requires 70 units per week, and region 3 requires 40 units per week. The costs (including production and shipping costs) of sending one unit from a plant to a region are shown in the table below. We want to meet weekly demands at minimum cost, subject to the preceding information and the following restrictions: 1. If the New York warehouse is opened, then the Los Angeles warehouse must be opened. 2. At most two warehouses can be opened. 3. Either the Atlanta or the Los Angeles warehouse must be opened. Formulate an IP that can be used to minimize the weekly costs of meeting demand.\n\n| To ($) From | Region 1 | Region 2 | Region 3 |\n|:---:|:---:|:---:|:---:|\n| New York | 20 | 40 | 50 |\n| Los Angeles | 48 | 15 | 26 |\n| Chicago | 26 | 35 | 18 |\n| Atlanta | 24 | 50 | 35 |\n\n

### Question 7

There are six cities (cities 1-6) in Kilroy County. The county must determine where to build fire stations. The county wants to build the minimum number of fire stations needed to ensure that at least one fire station is within 15 minutes (driving time) of each city. The times (in minutes) required to drive between the cities in Kilroy County are shown in the Table: Time Required to Travel between Cities in Kilroy County, From City 1 to ctiy 1-6 are 0 10 20 30 30 20; From City 2 to ctiy 1-6 are 10 0 25 35 20 10; From City 3 to ctiy 1-6 are 20 25 0 15 30 20; From City 4 to ctiy 1-6 are 30 35 15 0 15 25; From City 5 to ctiy 1-6 are 30 20 30 15 0 14; From City 5 to ctiy 1-6 are 20 10 20 25 14 0. Formulate an IP that will tell Kilroy how many fire stations should be built and where they should be located.

### Question 8

The transportation cost per unit in shipping a product from a factory (A or B) to a warehouse (W1 or W2) is shown below.\n\n| | W1 | W2 |\n|:---:|:---:|:---:|\n| A | 4 | 5 |\n| B | 6 | 3 |\n\nFor example sending one unit from Factory A to warehouse W2 costs $5. In the forthcoming month it is estimated that production capacity at A and B are 2500 and 3000 units respectively. Demand at W1 and W2 is estimated to be 4000 and 1500 units respectively.\nFor a variety of logistical reasons the amount shipped from factory A to warehouse W1 must be within 500 units of the amount shipped from factory B to warehouse W1. Formulate the problem of determine the optimal transportation schedule that minimises the total transportation cost as a linear program.

### Question 9

A client asks his stockbroker to invest $100,000 for maximum annual income, subject to the three conditions: Spread the investment over no more than three different stocks. Put no more than 40 percent of the money into any one stock. Put a minimum of $10,000 into an oil stock. The broker has identified three stocks for investment. Their estimated annual returns per share and price per share are shown in the following table: Stock, Price and annual returns are, (Oil, $120, $11), (Auto, $52, $4), (Pharmaceutical, $18, $2).

Figure 12: Problem examples from MIND-Bench.

# B PROMPT TEMPLATES

## B.1 PROMPT TEMPLATE FOR PRELIMINARY RESULTS

---

**Prompt template used for preliminary results**

You will be given:
- A natural language description of an optimization problem.
- A correct mathematical formulation for the optimization problem.
- PySCIPOpt code that may contain errors for the optimization problem.
```

{question}
```

is the natural language description of an optimization problem.
```

{mathematical formulation}
```

is the correct mathematical formulation for the optimization problem.
```

{python}
```

is the PySCIPOpt code that may contain errors for the optimization problem.
We define a mathematical formulation size function $S(\cdot)$ as follows:

$$S(\mathcal{MF}) = N_{\text{var}} + N_{\text{obj}} + N_{\text{cont}}, \tag{6}$$

where $N_{\text{var}}$, $N_{\text{obj}}$, and $N_{\text{cont}}$ denote the numbers of variables, objectives (always set to 1), and constraints, respectively.
Your task is to analyze the consistency between the correct formulation and its implementation in PySCIPOpt.

Step 1: Using the correct mathematical formulation $\mathcal{MF}^*$ as a reference, first compute the size of $\mathcal{MF}^*$, $S(\mathcal{MF}^*)$, by summing the sizes of all core expressions (variables, objectives, and constraints) in $\mathcal{MF}^*$.

Step 2: Identify which components of $\mathcal{MF}^*$ are incorrectly implemented in the PySCIPOpt code. When computing the size of the corresponding mathematical formulation, $S(\mathcal{MF}_{err})$, focus only on the correctness of each component's logic, ignoring other errors that do not affect the logical structure. Sum the sizes of these logically incorrect or missing components to obtain $S(\mathcal{MF}_{err})$.

Step 3: Calculate the error ratio $\mathcal{E}$ as

$$\mathcal{E} = \frac{S(\mathcal{MF}_{err})}{S(\mathcal{MF}^*)}.$$

Provide detailed, step-by-step reasoning for how $S(\mathcal{MF}^*)$ is computed from the correct formulation, how $S(\mathcal{MF}_{err})$ is determined based on missing or incorrect components, and report the final numeric value of the error ratio.

---

## B.2 PROMPT TEMPLATES FOR DATA SYNTHESIS

We use Deepseek-R1 for the error-driven reverse data synthesis pipeline.

**Prompt template used for single-error data synthesis**

You are a data synthesis expert in operations research. You will be given:
- A natural language description of an optimization problem.
- A correct mathematical formulation of the optimization problem.
- PySCIPOpt code that may contain errors for the optimization problem.
"'{question}'" is the natural language description of an optimization problem.
"'{mathematical formulation}'" is the correct mathematical formulation of the optimization problem
"'{python}'" is the PySCIPOpt code that may contain errors for the optimization problem.
Your task:
1. Carefully compare the PySCIPOpt code against both the natural language description and the correct mathematical formulation to detect all errors. These errors could include missing constraints, incorrect coefficients in the objective function or constraints, improper variable bounds or types (e.g., continuous instead of integer), a wrong objective direction (e.g., maximization instead of minimization), or other logical errors in translating the mathematical formulation into PySCIPOpt code.
2. Identify the specific portions of the PySCIPOpt code that are erroneous and label them as Error_Code_Portion. Also, identify and label the parts of the PySCIPOpt code that correctly implement the problem's requirements as Correct_Code_Portion. Then, for each Error_Code_Portion, provide the corrected PySCIPOpt code and label it as the Corrected_Code_Portion. From this corrected code, explicitly define the underlying modeling logic or pattern that was initially misapplied; this will be referred to as the Corrected_Modeling_Pattern.
3. Based on the Corrected_Modeling_Pattern, generate as many distinct additional problem instances as reasonably possible. These instances should showcase variety, covering different types of optimization problems, such as assignment and resource allocation optimization, cutting and packing optimization, domain-specific optimization (e.g., specific to a particular industry), facility location optimization, financial and revenue optimization, network flow optimization, production planning and scheduling optimization, or transportation and routing optimization. Similarly, explore diverse application scenarios, including agriculture, energy, health, retail, environment, education, financial services, transportation, public utilities, manufacturing, software, construction, legal, customer service, entertainment, and others. Each generated instance must include a natural language description (in plain English), its complete mathematical formulation, and the corresponding PySCIPOpt code.
4. You must ensure that the additional problem instances generated in the previous step adhere to a critical principle of uniqueness and focused reusability. Specifically, while each new problem instance must incorporate an implementation that is analogous in its core logic to the Corrected_Modeling_Pattern (this pattern can be adapted, for instance, by using a different number of variables, different coefficients suitable for the new problem within that pattern, or a moderately more complex variant of the same core idea), all other components of each new problem instance must be fundamentally different and more complex (more variables, more constraints, more advanced modeling strategies). This means the objective function, other constraints, overall problem structure, and variable sets not directly involved in the Corrected_Modeling_Pattern must not resemble the Correct_Code_Portion of the original PySCIPOpt code or the details of the original natural language description and correct mathematical formulation. This ensures that the additional problem instances are truly distinct from the original optimization problem in both their formulation and implementation, beyond the shared corrected modeling pattern.
5. Present the output as a JSON list of objects, each with fields "question" (problem description) and "code_solution" (PySCIPOpt code).

**Prompt template used for multi-error data synthesis**

You are a data synthesis expert in operations research. You will be given two automated optimization modeling problems (Problem A and Problem B), each composed of three components:
- A natural language description of an optimization problem,
- A correct mathematical formulation of the optimization problem,
- PySCIPOpt code that may contain errors for the optimization problem.

You need to identify the errors in the two automated optimization modeling problems and perform data synthesis to construct more challenging instances compared to the original problems. You can follow the steps below to do this:

1. Carefully compare the PySCIPOpt code for Problem A and Problem B against their corresponding natural language descriptions and mathematical formulations. Identify and document all discrepancies, including but not limited to: missing constraints, incorrect coefficients in the objective function or constraints, improper variable bounds or types (e.g., continuous instead of integer), a wrong objective direction (e.g., maximization instead of minimization), or other logical errors in translating the mathematical model into PySCIPOpt code.

2. Identify the specific portions of the PySCIPOpt code that are erroneous and label them as `Error_Code_Portion` for Problem A and Problem B. Also, identify and label the parts of the PySCIPOpt code that correctly implement the problem's requirements as `Correct_Code_Portion` for Problem A and Problem B. Then, for each `Error_Code_Portion`, provide the corrected PySCIPOpt code, labeling it as the `Corrected_Code_Portion` for Problem A and Problem B. From this corrected code, explicitly define the underlying modeling logic or pattern that was initially misapplied; this will be referred to as the `Corrected_Modeling_Pattern` for Problem A and Problem B.

3. Based on the `Corrected_Modeling_Pattern` for Problem A and Problem B, you should generate new, more complex instances that simultaneously include the `Corrected_Modeling_Pattern` of both Problem A and Problem B within a single instance. These instances should showcase variety, covering different optimization problem types such as assignment and resource allocation optimization, cutting and packing optimization, domain-specific optimization (e.g., specific to a particular industry), facility location optimization, financial and revenue optimization, network flow optimization, production planning and scheduling optimization, or transportation and routing optimization. Similarly, explore diverse application scenarios, including agriculture, energy, health, retail, environment, education, financial services, transportation, public utilities, manufacturing, software, construction, legal, customer service, entertainment, and others. Each generated instance must include a natural language description (in plain English), its complete mathematical formulation, and the corresponding PySCIPOpt code.

4. For each newly generated instance, you must simultaneously include the `Corrected_Modeling_Pattern` of both Problem A and Problem B. The rest of the mathematical formulation can be arbitrary, but it should be substantially different from the original formulations of Problem A and Problem B.

5. Present the output as a JSON list of objects, each with fields "question" (problem description) and "code_solution" (PySCIPOpt code).

Automated optimization problem A as follows:
{question1} is the natural language description of an optimization problem. {model1} is the correct mathematical formulation for the optimization problem. {python1} is PySCIPOpt code for the optimization problem.

Automated optimization problem B as follows:
{question2} is the natural language description of an optimization problem. {model2} is a correct mathematical formulation for the optimization problem. {python2} is PySCIPOpt code for the optimization problem.

Now, follow the examples to present the output as a JSON list of object...

## B.3 Prompt Template for Chain-of-Thought

Following DeepSeek-R1-Zero (Guo et al., 2025) and SIRL (Chen et al., 2025), we adopt a chain-of-thought prompt. First, we prompt the LLM to analyze the problem and extract key information to build a rationale. Second, we prompt the LLM to construct a mathematical formulation. Finally, we prompt the LLM to translate the mathematical formulation into executable PySCIPOpt Python code.

---

**Prompt template used for chain-of-thought reasoning with Qwen2.5-7B-Instruct**

**SYSTEM:** You are a helpful assistant with expertise in mathematical modeling and the PySCIPOpt solver. When the User provides an operations research problem, you will analyze it, build a detailed mathematical model, and provide the PySCIPOpt code to solve it.
Your response should follow these steps:

1. `<think>`
   Carefully analyze the problem to identify decision variables, objective, and constraints.
   `</think>`

2. `<model>`
   Develop a complete mathematical model, explicitly defining:

   - Sets
   - Parameters
   - Decision Variables (and their types)
   - Objective Function
   - Constraints

   `</model>`

3. `<python>`
   Provide the corresponding PySCIPOpt Python code to implement the model.
   `</python>`

**USER:** Answer the following mathematical modeling question:
'''question
`{question}`
'''

Let's think step by step and fill in the PySCIPOpt code into
''' python
`{python}`
'''.

---

---

**Prompt template used for chain-of-thought reasoning with Qwen3-8B**

**SYSTEM:** You are a helpful assistant with expertise in mathematical modeling and the PySCIPOpt solver. When the User provides an operations research problem, you will analyze it, build a detailed mathematical model, and provide the PySCIPOpt code to solve it.

Your response should follow these steps:

1. `<analysis>`

   Carefully analyze the problem to identify decision variables, objective, and constraints.

   `</analysis>`

2. `<model>`

   Develop a complete mathematical model, explicitly defining:

   - Sets
   - Parameters
   - Decision Variables (and their types)
   - Objective Function
   - Constraints

   `</model>`

3. `<python>`

   Provide the corresponding PySCIPOpt Python code to implement the model.

   `</python>`

**USER:** Answer the following mathematical modeling question:

```question
{question}
```

Letś think step by step and fill in the PySCIPOpt code into

``` python
{python}
```. /no_think

---

### B.4 PROMPT TEMPLATE FOR DYNAMIC SFT

First, we design a prompt to generate `correct_response`. Specifically, we use ground-truth solutions as guidance and independently solve the operations research problems through chain-of-thought reasoning, thereby generating the desired responses.

> **Prompt template used to generate correct response**
>
> **SYSTEM:** You are a helpful Assistant with expertise in mathematical modeling and the PySCIPOpt solver. When the User provides an OR question, you will analyze it, build a detailed mathematical model, and provide the PySCIPOpt code to solve it.
>
> Before answering, you may review the provided reference reasoning or code {ground_truth_formulation} for guidance only. Do not copy or rely on it directly. Your solution must be fully generated independently, using your own analysis and reasoning. Your response should follow these steps:
>
> 1. `<analysis>`
>     Explain how the reference {ground_truth_formulation} can guide your reasoning. Highlight any insights or techniques you can borrow, but do not copy any content verbatim. Be concise and structured.
>    `</analysis>`
> 2. `<response>`
>     Provide your complete independent solution, including:
>
>     1. `<think>`
>        Carefully analyze the problem to identify decision variables, objective, and constraints.
>        `</think>`
>
>     2. `<model>`
>        Develop a complete mathematical model, explicitly defining:
>
>        - Sets
>        - Parameters
>        - Decision Variables (and their types)
>        - Objective Function
>        - Constraints
>
>        `</model>`
>
>     3. `<python>`
>        Provide the corresponding PySCIPOpt Python code to implement the model.
>        `</python>`
>
>    `</response>`
> Your final output must therefore contain exactly two sections:
> `<analysis>`...`</analysis>`
> `<response>`...`</response>`
> **USER:** Answer the following mathematical modeling question:
> '''question
> {question}
> '''
> Letś think step by step.

Then, we design a prompt to correct wrong responses. Specifically, we use `correct_response` as a reference to correct wrong responses from LLM post-training rollouts, thereby obtaining the corrected responses.

---

**Prompt template used to correct wrong response**

You are a helpful assistant with expertise in mathematical modeling and the PySCIPOpt solver. The operations research question is as follows:
`{question}`.
The correct mathematical modeling response (for reference only) is as follows:
`{correct_response}`.
The wrong mathematical modeling response from another LLM is as follows:
`{wrong_response}`.
Your task:
1. Write your reasoning about how to modify the wrong response based on the correct response inside `<analysis>`...`</analysis>` tags.
  - In this section you may explain which parts of the wrong response are incorrect, why, and how they should be corrected.
  - Be concise and structured.
2. Output the \*\*entire corrected version of the wrong response\*\* inside `<corrected response>`...`</corrected response>` tags.
  - The corrected response must preserve all parts of the wrong response that are already correct.
  - Change only the portions that are actually incorrect.
  - Do not add extra explanation, justification, or commentary in this section — only the corrected content.
  - Keep the same Python coding style as in the wrong response. Do not wrap code into a function.
Your final output must therefore contain exactly two sections:
`<analysis>`...`</analysis>`
`<corrected response>`...`</corrected response>`

---

## C  MIND DETAILS

### C.1  TRAINING AND INFERENCE DETAILS

**Training Hyperparameters**  All experiments were conducted on a single computing node equipped with four NVIDIA A100 GPUs, each with 80 GB of memory. The ms-swift framework (Zhao et al., 2025) was used to implement SFT, while the VeRL framework (Sheng et al., 2025) was used to implement GRPO, DAPO and DFPO. All training hyperparameters are listed in Table 6, Table 7, Table 8 and Table 9.

**Inference Hyperparameters**  As shown in Table 10, we use a greedy decoding strategy for LLM inference to ensure reproducibility.

### C.2  PRELIMINARY RESULTS ON DEEPSEEK-V3

As a supplement to the preliminary results on Qwen2.5-7B-Instruct, we conduct the same preliminary experiments using Deepseek-V3, a model with a different architecture, on the OR-Instruct-3K. We also analyze the distribution of error ratios for the questions on which Deepseek-V3 make errors. As shown in Figure 13, when errors occur, Deepseek-V3 also introduces only a small fraction of errors rather than producing entirely incorrect formulations in most cases, further supporting the conclusions observed for Qwen2.5-7B-Instruct. Additionally, we find that Deepseek-V3 has a lower average error ratio of 29% compared with 33% for Qwen2.5-7B-Instruct, indicating that more powerful LLM may have a higher capacity to produce fewer errors per instance.

### C.3  REWARD WEIGHT SENSITIVITY ANALYSIS

For our reward function hyperparameter $\alpha$, we evaluate its influence by testing values in {0.0, 0.2, 0.4, 0.6}, with the results shown in Figure 14. For the experimental details, we use DAPO to train Qwen2.5-7B-Instruct on the training dataset (10,000 instances) for 7 epochs. We note that $\alpha = 0.0$

Table 6: List of training hyperparameters and their values used in the DFPO.

| Data | |
|---|---|
| **Parameter** | **Value** |
| Optimizer | AdamW |
| Training epochs | 26 |
| Training batch size | 1024 |
| Max prompt length | 4096 |
| Max response length | 8192 |
| Learning rates | $10^{-6}$ |
| Truncation | left |
| **Actor** | |
| **Parameter** | **Value** |
| Number of rollouts per prompt | 8 |
| PPO mini-batch size | 256 |
| Clip ratio low | 0.20 |
| Clip ratio high | 0.28 |
| Entropy loss | Disabled |
| KL loss | Disabled |
| Gradient clipping | 1.0 |
| temperature (sampling) | 1.0 |
| Top p (sampling) | 1.0 |
| Top k (sampling) | -1 |
| $\alpha$ | 0.2 |
| $\beta$ | 0.05 |
| $\gamma$ | 0.8 |
| **Reward** | |
| **Parameter** | **Value** |
| Overlong buffer length | 4096 |
| Overlong penalty factor | 1.0 |

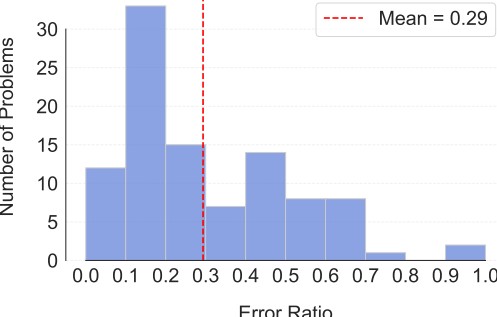

Figure 13: Distribution of error ratio across 100 incorrect generation results for Deepseek-V3.

Table 7: List of training hyperparameters and their values used in the DAPO.

| Data | |
|---|---|
| **Parameter** | **Value** |
| Optimizer | AdamW |
| Training epochs | 26 |
| Training batch size | 1024 |
| Max prompt length | 4096 |
| Max response length | 8192 |
| Learning rates | $10^{-6}$ |
| Truncation | left |

| Actor | |
|---|---|
| **Parameter** | **Value** |
| Number of rollouts per prompt | 8 |
| PPO mini-batch size | 256 |
| Clip ratio low | 0.20 |
| Clip ratio high | 0.28 |
| Entropy loss | Disabled |
| KL loss | Disabled |
| Gradient clipping | 1.0 |
| temperature (sampling) | 1.0 |
| Top p (sampling) | 1.0 |
| Top k (sampling) | -1 |

| Reward | |
|---|---|
| **Parameter** | **Value** |
| Overlong buffer length | 4096 |
| Overlong penalty factor | 1.0 |

Table 8: List of training hyperparameters and their values used in the GRPO.

| Data | |
|---|---|
| **Parameter** | **Value** |
| Optimizer | AdamW |
| Training epochs | 26 |
| Training batch size | 1024 |
| Max prompt length | 2048 |
| Max response length | 8192 |
| Learning rates | $10^{-6}$ |

| Actor | |
|---|---|
| **Parameter** | **Value** |
| Number of rollouts per prompt | 8 |
| PPO mini-batch size | 256 |
| Entropy loss | Disabled |
| KL loss coefficient | 0.001 |
| KL loss type | Low Var KL |
| Gradient clipping | 1.0 |
| temperature (sampling) | 1.0 |
| Top p (sampling) | 1.0 |
| Top k (sampling) | -1 |

Table 9: List of training hyperparameters and their values used in the SFT.

| Parameter | Value |
|---|---|
| Optimizer | AdamW |
| Training epochs | 3 |
| Training batch size | 2 |
| Gradient accumulation steps | 8 |
| Max prompt length | 4096 |
| Max response length | 8192 |
| Learning rates | $10^{-4}$ |
| Train type | LoRA Yu et al. (2023) |
| LoRA rank | 8 |
| LoRA alpha | 32 |

Table 10: List of inference hyperparameters and their values used in the DFPO.

| Decoding Settings | |
|---|---|
| **Parameter** | **Value** |
| Max tokens | 8192 |
| Temperature | 0.0 |

corresponds to a standard 0-1 reward. The results show that $\alpha = 0.2$ and $\alpha = 0.4$ achieve better performance on most benchmarks compared with $\alpha = 0.0$ and $\alpha = 0.6$, indicating that the fidelity reward, as an auxiliary signal, should not dominate the final reward value.

## C.4 ABLATION STUDY OF DATA SYNTHESIS STRATEGIES

To verify the difference between single-error and multi-error strategies, we split the MIND-3K training dataset, which is a mixture of single-error and multi-error data synthesis dataset, into MIND-Single-1.5K (1,500 instances) and MIND-Multi-1.5K (1,500 instances). We then employ DAPO to train Qwen2.5-7B-Instruct from scratch on MIND-Single-1.5K and MIND-Multi-1.5K. As shown in Figure 15, the model achieves better training performance on MIND-Mix-3K compared with MIND-Single-1.5K and MIND-Multi-1.5K. Furthermore, Table 11 presents a detailed performance comparison after seven training epochs across six benchmarks. Our results also show that training on MIND-Single-1.5K leads to better performance than training on MIND-Multi-1.5K. We hypothesize that this disparity arises because LLMs struggle to learn effectively when trained directly on highly challenging datasets. To further substantiate this hypothesis, we evaluate Qwen2.5-7B-Instruct on both datasets. The model achieves an average accuracy of 52.9% on MIND-Single-1.5K, but only 41.2% on MIND-Multi-1.5K. This pronounced accuracy gap corroborates our claim that multi-error reverse data synthesis generates datasets that are substantially more difficult than those produced by single-error synthesis.

Table 11: Ablation results for the single-error and multi-error strategies on Qwen2.5-7B-Instruct. (pass@1↑).

| Data | NL4OPT | IndustryOR | EasyLP | ComplexLP | OptMATH | OptiBench | Macro AVG |
|---|---|---|---|---|---|---|---|
| MIND-Single-1.5K | 91.4% | 29.0% | 90.4% | 40.9% | 8.4% | 53.6% | 52.3% |
| MIND-Multi-1.5K | 91.4% | 29.0% | 90.2% | 33.0% | 6.0% | 54.0% | 50.6% |
| MIND-Mix-3K | 94.3% | 30.0% | 90.8% | 39.9% | 7.8% | 55.5% | 53.1% |

## C.5 MODELING ERROR ANALYSIS

We randomly sample 300 erroneous responses each from Qwen2.5-7B-Instruct (before post-training) and MIND-Qwen2.5-7B (after DFPO-based post-training). We first defined a taxonomy of

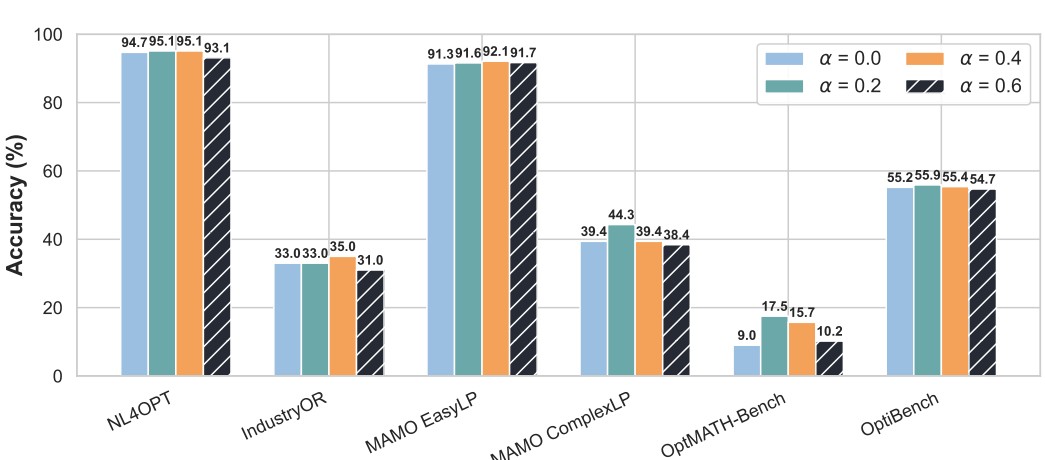

Figure 14: Performance comparison for different $\alpha$ in the reward function.

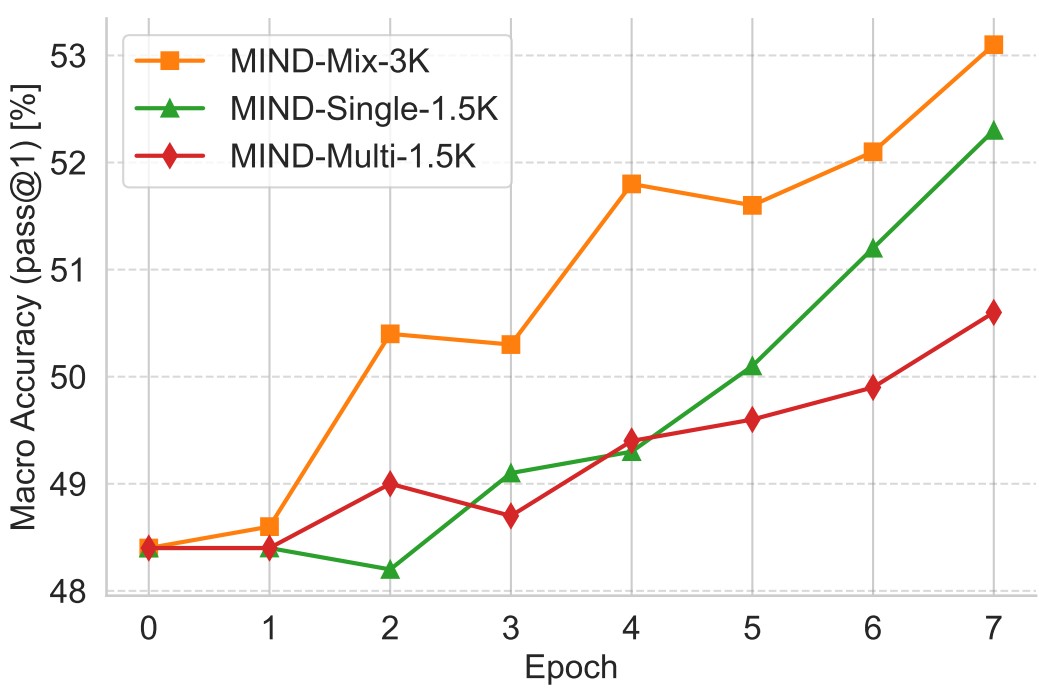

Figure 15: Ablation study for the single-error and multi-error strategies across six benchmarks.

Table 12: Analysis of modeling error types in optimization modeling.

| Error Type | Concrete Error | Qwen2.5-7B | MIND-Qwen2.5-7B |
|---|---|---|---|
| Variables | Incorrect decision variables. | 12.1% | 15.5% (↑ 3.4%) |
| | Decision variables omission. | 4.4% | 10.6% (↑ 6.2%) |
| | Superfluous decision variables. | 7.7% | 8.1% (↑ 0.4%) |
| | Incorrect variable types. | 11.8% | 7.1% (↓ 4.7%) |
| Objective | Optimization direction error. | 1.4% | 0.0% (↓ 1.4%) |
| | Incorrect objective terms. | 12.8% | 4.2% (↓ 8.6%) |
| | Objective terms omission. | 3.0% | 2.5% (↓ 0.5%) |
| | Superfluous objective terms. | 1.7% | 0.4% (↓ 1.3%) |
| | Incorrect or missing advanced modeling techniques. The incorrect application or omission of sophisticated modeling techniques, which can lead to improper handling of multi-objective problems, non-linear objectives or other advanced modeling scenarios. | 2.7% | 5.3% (↑ 2.6%) |
| Constraints | Incorrect constraint. | 11.8% | 15.5% (↑ 3.7%) |
| | Constraint omission. | 10.1% | 8.5% (↓ 1.6%) |
| | Superfluous constraints. | 3.7% | 0.0% (↓ 3.7%) |
| | Equality and inequality constraints confusion. | 4.0% | 4.2% (↑ 0.2%) |
| | Incorrect or missing advanced modeling techniques. The incorrect application or omission of sophisticated modeling techniques, which can lead to improper handling of non-linear constraints, logical constraints, or other advanced modeling scenarios. | 1.0% | 11.7% (↑ 10.7%) |
| Parameters | Incorrect parameters definition. This includes missing essential parameters, incorrectly defined parameters, parameters assigned with wrong numerical values, or other incorrect parameter definition scenarios. | 8.4% | 4.6% (↓ 3.8%) |
| | Parameters misuse. The incorrect use of defined parameters, such as value misuse, unit or scale misuse, reference errors, or other improper applications of parameters. | 3.4% | 1.8% (↓ 1.6%) |

error types relevant to optimization modeling. For each query-response pair, three domain experts independently annotated the dominant error category, achieving high inter-annotator agreement. As shown in Table 12, the top five error types for Qwen2.5-7B-Instruct are "incorrect objective terms (12.8%)", "incorrect decision variables (12.1%)", "incorrect constraint (11.8%)", "incorrect variable types (11.8%)", and "constraint omission (10.1%)". In contrast, the top five errors for MIND-Qwen2.5-7B are "incorrect decision variables (15.5%)", "incorrect constraint (15.5%)", "incorrect or missing advanced modeling techniques (11.7%)", "decision variable omission (10.6%)", and "constraint omission (8.5%)".

Notably, while basic syntactic or structural errors (e.g., wrong variable types) diminish after post-training, new dominant errors involve more sophisticated modeling challenges, such as the appropriate use of advanced techniques (e.g., piecewise linearization, or indicator constraints) and comprehensive problem scoping (e.g., omitting key variables or constraints in complex scenarios). This shift strongly suggests that DFPO effectively mitigates simpler, surface-level errors, push the model's failure modes toward deeper, semantics-rich challenges—a hallmark of improved reasoning capability.

---

**Question 1 from MAMO ComplexLP Benchmark**

Imagine you are a dietitian and you have been tasked with creating a meal plan for a bodybuilder. You have six food items to choose from: Steak, Tofu, Chicken, Broccoli, Rice, and Spinach. Each food provides certain amounts of protein, carbohydrates, and calories, and each has its own cost.\n\nHere's the nutritional value and cost of each food:\n\n- Steak: It gives you 14 grams of protein, 23 grams of carbohydrates, and 63 calories for $4.\n- Tofu: It offers 2 grams of protein, 13 grams of carbohydrates, and 162 calories for $6.\n- Chicken: It packs a punch with 17 grams of protein, 13 grams of carbohydrates, and gives you 260 calories for $6.\n- Broccoli: It provides 3 grams of protein, a mere 1 gram of carbohydrates, and 55 calories for $8.\n- Rice: It gives a hearty 15 grams of protein, 23 grams of carbohydrates, and 231 calories for $8.\n- Spinach: It provides 2 grams of protein, 8 grams of carbohydrates, and a huge 297 calories for just $5.\n\nYour goal is to ensure that the bodybuilder gets at least 83 grams of protein, 192 grams of carbohydrates, and 2089 calories from whatever combination of these foods you choose. The challenge is to keep the cost as low as possible while meeting these nutritional targets. \n\nWhat is the minimum cost to meet these nutritional requirements with the available food options?

Figure 16: Question of example 1.

---

**Question 2 from OptMATH Benchmark**

The manufacturing facility produces custom components for two jobs, Job 0 and Job 1, each consisting of a sequence of operations that must be performed in a specific order. The goal is to schedule these operations to minimize the total completion time (makespan) while satisfying all operational constraints. Job 0 has five operations with processing times: Operation 0 takes 4 units, Operation 1 takes 1 unit, Operation 2 takes 6 units, Operation 3 takes 6 units, and Operation 4 takes 8 units. Job 1 has four operations with processing times: Operation 0 takes 9 units, Operation 1 takes 1 unit, Operation 2 takes 4 units, and Operation 3 takes 2 units.\n\nPrecedence constraints ensure that operations within each job are performed in sequence with specific gaps. For Job 0, Operation 1 must start at least 4 units after Operation 0 starts, Operation 2 must start at least 1 unit after Operation 1 starts, Operation 3 must start at least 6 units after Operation 2 starts, and Operation 4 must start at least 6 units after Operation 3 starts. For Job 1, Operation 1 must start at least 9 units after Operation 0 starts, Operation 2 must start at least 1 unit after Operation 1 starts, and Operation 3 must start at least 4 units after Operation 2 starts.\n\nMachine capacity constraints ensure that operations assigned to the same machine do not overlap. Binary variables determine the order of operations on shared machines. For example, if Operation 1 of Job 0 and Operation 3 of Job 0 are on the same machine, one must complete at least 6 units before the other starts. Similarly, if Operation 1 of Job 0 and Operation 2 of Job 1 are on the same machine, one must complete at least 4 units before the other starts. These constraints apply to all operation pairs on shared machines, ensuring no overlap and maintaining required time gaps. The large constant of 100,000 is used in these constraints to enforce the sequencing logic by ensuring that the constraints are only active when the binary variable is set to 1.\n\nThe makespan must be at least as large as the completion time of every operation. Specifically, the makespan must be at least 4 units after Operation 0 of Job 0 starts, 1 unit after Operation 1 of Job 0 starts, 6 units after Operation 2 of Job 0 starts, 6 units after Operation 3 of Job 0 starts, 8 units after Operation 4 of Job 0 starts, 9 units after Operation 0 of Job 1 starts, 1 unit after Operation 1 of Job 1 starts, 4 units after Operation 2 of Job 1 starts, and 2 units after Operation 3 of Job 1 starts.\n\nThe objective is to determine the start times for all operations and the sequence of operations on shared machines to minimize the makespan while respecting all precedence constraints, machine capacity constraints, and the makespan definition. Binary variables are used to enforce the correct sequencing of operations on shared machines, with a large constant of 100,000 ensuring the constraints are properly applied. The start times for all operations and the makespan must be non-negative. The binary variables are explicitly defined for all relevant operation pairs on shared machines, and the start times and makespan are continuous variables.

Figure 17: Question of example 2.

### C.6 CASE STUDY: EXAMPLES OF CORRECTED WRONG RESPONSES FOR DYNAMIC SFT

We select the first instances from MAMO ComplexLP and OptMATH as example 1 (Figure 16) and example 2 (Figure 17), respectively. As shown in Figure 18 and Figure 19, these cases illustrate how a powerful LLM can correct the errors in the wrong response from the base model, producing a corrected response whose distribution closely matches that of the wrong response.

Example 1 is a diet problem. Since the problem does not explicitly require food to be purchased in whole units, the variables should be allowed to take continuous values. Although the variable type is incorrect, this does not affect the correctness of the parameters, objective function, or constraints.

| Question 1 | |
|---|---|
| **Wrong Response** | **Corrected Response** |

**Wrong Response:**

```
<think>
To solve this problem, we need to define decision variables, the objective
function, and the constraints based on the given nutritional requirements and
costs of the food items.
- **Decision Variables**: We need to decide how much of each food item to
include in the meal plan. Let's denote the amount of each food item by a
variable. Since we are dealing with quantities of food, these variables will
be integer (non-negative real numbers).
- **Objective Function**: The goal is to minimize the total cost of the meal
plan. The cost for each food item is given, so we will sum the costs of all the
food items chosen.
- **Constraints**: We need to ensure that the total amount of protein,
carbohydrates, and calories meet or exceed the specified requirements:
  - At least 83 grams of protein.
  - At least 192 grams of carbohydrates.
  - At least 2089 calories.
</think>
<model>
- **Sets**: None needed as we are dealing with individual food items.
- **Parameters**:
  - `protein[i]`: Protein content of food `i`.
  - `carbs[i]`: Carbohydrates content of food `i`.
  - `calories[i]`: Calories content of food `i`.
  - `cost[i]`: Cost of food `i`.
  - `min_protein`: Minimum required protein.
  - `min_carbs`: Minimum required carbohydrates.
  - `min_calories`: Minimum required calories.
- **Decision Variables**:
  - `x[i]`: Amount of food `i` to include in the meal plan. This is a integer
variable (non-negative).
- **Objective Function**:
  - Minimize the total cost: \(\text{Minimize} \sum_{i} \text{cost}[i] \cdot
x[i]\).
- **Constraints**:
  - Protein constraint: \(\sum_{i} \text{protein}[i] \cdot x[i] \geq
\text{min\_protein}\).
  - Carbohydrates constraint: \(\sum_{i} \text{carbs}[i] \cdot x[i] \geq
\text{min\_carbs}\).
  - Calories constraint: \(\sum_{i} \text{calories}[i] \cdot x[i] \geq
\text{min\_calories}\).
</model>
<python>
```python
from pyscipopt import Model
# Create model
model = Model("DietitianProblem")
# Define food items and their nutritional values and costs
foods = {
    'Steak': {'protein': 14, 'carbs': 23, 'calories': 63, 'cost': 4},
    'Tofu': {'protein': 2, 'carbs': 13, 'calories': 162, 'cost': 6},
    'Chicken': {'protein': 17, 'carbs': 13, 'calories': 260, 'cost': 6},
    'Broccoli': {'protein': 3, 'carbs': 1, 'calories': 55, 'cost': 8},
    'Rice': {'protein': 15, 'carbs': 23, 'calories': 231, 'cost': 8},
    'Spinach': {'protein': 2, 'carbs': 8, 'calories': 297, 'cost': 5}
}
# Define minimum nutritional requirements
min_protein = 83
min_carbs = 192
min_calories = 2089
# Define decision variables
x = {}
for food in foods:
    x[food] = model.addVar(vtype="I", name=food)
# Objective: Minimize the total cost
model.setObjective(sum(foods[food]['cost'] * x[food] for food in foods),
"minimize")
# Constraints
# Protein constraint
model.addCons(sum(foods[food]['protein'] * x[food] for food in foods) >=
min_protein, "ProteinRequirement")
# Carbohydrates constraint
model.addCons(sum(foods[food]['carbs'] * x[food] for food in foods) >=
min_carbs, "CarbsRequirement")
# Calories constraint
model.addCons(sum(foods[food]['calories'] * x[food] for food in foods) >=
min_calories, "CaloriesRequirement")
# Solve the model
model.optimize()
# Print the solution
if model.getStatus() == 'optimal':
    print("Objective: ", model.getObjVal())
else:
    print("No optimal solution found.")
</python>
```

**Corrected Response:**

```
<think>
To solve this problem, we need to define decision variables, the objective
function, and the constraints based on the given nutritional requirements and
costs of the food items.
- **Decision Variables**: We need to decide how much of each food item to
include in the meal plan. Let's denote the amount of each food item by a
variable. Since we are dealing with quantities of food, these variables will
be continuous (non-negative real numbers).
- **Objective Function**: The goal is to minimize the total cost of the meal
plan. The cost for each food item is given, so we will sum the costs of all the
food items chosen.
- **Constraints**: We need to ensure that the total amount of protein,
carbohydrates, and calories meet or exceed the specified requirements:
  - At least 83 grams of protein.
  - At least 192 grams of carbohydrates.
  - At least 2089 calories.
</think>
<model>
- **Sets**: None needed as we are dealing with individual food items.
- **Parameters**:
  - `protein[i]`: Protein content of food `i`.
  - `carbs[i]`: Carbohydrates content of food `i`.
  - `calories[i]`: Calories content of food `i`.
  - `cost[i]`: Cost of food `i`.
  - `min_protein`: Minimum required protein.
  - `min_carbs`: Minimum required carbohydrates.
  - `min_calories`: Minimum required calories.
- **Decision Variables**:
  - `x[i]`: Amount of food `i` to include in the meal plan. This is a continuous
variable (non-negative).
- **Objective Function**:
  - Minimize the total cost: \( \text{Minimize} \sum_{i} \text{cost}[i] \cdot
x[i] \).
- **Constraints**:
  - Protein constraint: \(\sum_{i} \text{protein}[i] \cdot x[i] \geq
\text{min\_protein}\).
  - Carbohydrates constraint: \(\sum_{i} \text{carbs}[i] \cdot x[i] \geq
\text{min\_carbs}\).
  - Calories constraint: \(\sum_{i} \text{calories}[i] \cdot x[i] \geq
\text{min\_calories}\).
### Step 3: Provide the Corresponding Pyscipopt Code
</model>
<python>
```python
from pyscipopt import Model
# Create model
model = Model("DietitianProblem")
# Define food items and their nutritional values and costs
foods = {
    'Steak': {'protein': 14, 'carbs': 23, 'calories': 63, 'cost': 4},
    'Tofu': {'protein': 2, 'carbs': 13, 'calories': 162, 'cost': 6},
    'Chicken': {'protein': 17, 'carbs': 13, 'calories': 260, 'cost': 6},
    'Broccoli': {'protein': 3, 'carbs': 1, 'calories': 55, 'cost': 8},
    'Rice': {'protein': 15, 'carbs': 23, 'calories': 231, 'cost': 8},
    'Spinach': {'protein': 2, 'carbs': 8, 'calories': 297, 'cost': 5}
}
# Define minimum nutritional requirements
min_protein = 83
min_carbs = 192
min_calories = 2089
# Define decision variables
x = {}
for food in foods:
    x[food] = model.addVar(vtype="C", name=food)
# Objective: Minimize the total cost
model.setObjective(sum(foods[food]['cost'] * x[food] for food in foods),
"minimize")
# Constraints
# Protein constraint
model.addCons(sum(foods[food]['protein'] * x[food] for food in foods) >=
min_protein, "ProteinRequirement")
# Carbohydrates constraint
model.addCons(sum(foods[food]['carbs'] * x[food] for food in foods) >=
min_carbs, "CarbsRequirement")
# Calories constraint
model.addCons(sum(foods[food]['calories'] * x[food] for food in foods) >=
min_calories, "CaloriesRequirement")
# Solve the model
model.optimize()
# Print the solution
if model.getStatus() == 'optimal':
    print("Objective: ", model.getObjVal())
else:
    print("No optimal solution found.")
</python>
```

Figure 18: Wrong resposne and corrected response corresponding to example 1.

| Question 2 | |
|---|---|
| **Wrong Response** | **Corrected Response** |

**Wrong Response:**

### Step 1: Analyze the Problem
**Decision Variables:**
- $S_i$: Start time of operation $i$ (for all operations $i$).
- $M$: Makespan (total completion time).
- **$x_{ij}$: Binary variable indicating if operation $i$ and operation $j$ are on the same machine (where $i$ and $j$ are operations from different jobs or from the same job but on shared machines).**

**Objective:**
- Minimize the makespan $M$.

**Constraints:**
1. **Precedence Constraints:**
   - For Job 0:
     - Operation 1 must start at least 4 units after Operation 0 starts: $S_1 \geq S_0 + 4$
     - Operation 2 must start at least 1 unit after Operation 1 starts: $S_2 \geq S_1 + 1$
     - Operation 3 must start at least 6 units after Operation 2 starts: $S_3 \geq S_2 + 6$
     - Operation 4 must start at least 6 units after Operation 3 starts: $S_4 \geq S_3 + 6$
   - For Job 1:
     - Operation 1 must start at least 9 units after Operation 0 starts: $S_5 \geq S_4 + 9$
     - Operation 2 must start at least 1 unit after Operation 1 starts: $S_6 \geq S_5 + 1$
     - Operation 3 must start at least 4 units after Operation 2 starts: $S_7 \geq S_6 + 4$

2. **Machine Capacity Constraints:**
   - For each pair of operations on the same machine, ensure there is no overlap. Use a large constant (100,000) to enforce sequencing logic.
   - For example, if Operation 1 of Job 0 and Operation 3 of Job 0 are on the same machine, one must complete at least 6 units before the other starts: $S_1 + p_1 \leq S_3 + 100000 \cdot (1 - x_{13})$ and **$S_3 + p_3 \leq S_1 + 100000 \cdot x_{13}$**
   - Similarly, if Operation 1 of Job 0 and Operation 2 of Job 1 are on the same machine, one must complete at least 4 units before the other starts: $S_1 + p_1 \leq S_6 + 100000 \cdot (1 - x_{16})$ and **$S_6 + p_6 \leq S_1 + 100000 \cdot x_{16}$**

3. **Makespan Constraints:**
   - The makespan must be at least as large as the completion time of every operation:
     - Operation 0 of Job 0: $M \geq S_0 + 4$ - Operation 1 of Job 0: $M \geq S_1 + 1$
     - Operation 2 of Job 0: $M \geq S_2 + 6$ - Operation 3 of Job 0: $M \geq S_3 + 6$
     - Operation 4 of Job 0: $M \geq S_4 + 8$ - **Operation 0 of Job 1: $M \geq S_4 + 9$**
     - **Operation 1 of Job 1: $M \geq S_5 + 1$ - Operation 2 of Job 1: $M \geq S_6 + 4$**
     - **Operation 3 of Job 1: $M \geq S_7 + 2$**

4. **Non-negativity and Continuity:**
   - All start times and the makespan must be non-negative and continuous variables.

### Step 2: Develop the Mathematical Model
**Sets:**
- No explicit sets are required here as the operations and their dependencies are explicitly defined.

**Parameters:**
- $p_i$: Processing time of operation $i$.

**Decision Variables:**
- $S_i$: Start time of operation $i$.
- $M$: Makespan.
- **$x_{ij}$: Binary variable indicating if operation $i$ and operation $j$ are on the same machine.**

**Objective:**
Minimize $M$.

**Constraints:**
1. **Precedence Constraints:**
   - For Job 0:
     $[S_1 \geq S_0 + 4]\[S_2 \geq S_1 + 1]\[S_3 \geq S_2 + 6]\ [S_4 \geq S_3 + 6]$
   - For Job 1:
     $[S_5 \geq S_4 + 9]\[S_6 \geq S_5 + 1]\[S_7 \geq S_6 + 4]$

2. **Machine Capacity Constraints:**
   - Example for Operation 1 of Job 0 and Operation 3 of Job 0 on the same machine:
     $[S_1 + p_1 \leq S_3 + 100000 \cdot (1 - x_{13})]$
     **$[S_3 + p_3 \leq S_1 + 100000 \cdot x_{13}]$**
   - Example for Operation 1 of Job 0 and Operation 2 of Job 1 on the same machine:
     $[S_1 + p_1 \leq S_6 + 100000 \cdot (1 - x_{16})]$
     **$[S_6 + p_6 \leq S_1 + 100000 \cdot x_{16}]$**

3. **Makespan Constraints:**
   - For each operation:
     $[M \geq S_0 + 4]$  $[M \geq S_1 + 1]$  $[M \geq S_2 + 6]$
     $[M \geq S_3 + 6]$  $[M \geq S_4 + 8]$  **$[M \geq S_4 + 9]$**
     **$[M \geq S_5 + 1]$  $[M \geq S_6 + 4]$  $[M \geq S_7 + 2]$**

### Step 3: Provide the Corresponding Pyscipopt Python Code
```python
from pyscipopt import Model
model = Model("Job_Scheduling")
# Define operations and their processing times
operations = {
    '0': 4,  # Operation 0 of Job 0  '1': 1,  # Operation 1 of Job 0  '2': 6,  # Operation 2 of Job 0
    '3': 6,  # Operation 3 of Job 0  '4': 8,  # Operation 4 of Job 0  '5': 9,  # Operation 0 of Job 1
    '6': 1,  # Operation 1 of Job 1  '7': 4,  # Operation 2 of Job 1  '8': 2  # Operation 3 of Job 1
}
# Define start times for each operation
S = {op: model.addVar(vtype="C", name=f"S_{op}") for op in operations}
# Define the makespan
M = model.addVar(vtype="C", name="M")
# Precedence constraints for Job 0
model.addCons(S['1'] >= S['0'] + 4, name="precedence_0_1")
model.addCons(S['2'] >= S['1'] + 1, name="precedence_1_2")
model.addCons(S['3'] >= S['2'] + 6, name="precedence_2_3")
model.addCons(S['4'] >= S['3'] + 6, name="precedence_3_4")
# Precedence constraints for Job 1
model.addCons(S['5'] >= S['4'] + 9, name="precedence_4_5")
model.addCons(S['6'] >= S['5'] + 1, name="precedence_5_6")
model.addCons(S['7'] >= S['6'] + 4, name="precedence_6_7")
# Makespan constraints
model.addCons(M >= S['0'] + operations['0'], name="makespan_0")
model.addCons(M >= S['1'] + operations['1'], name="makespan_1")
model.addCons(M >= S['2'] + operations['2'], name="makespan_2")
model.addCons(M >= S['3'] + operations['3'], name="makespan_3")
model.addCons(M >= S['4'] + operations['4'], name="makespan_4")
model.addCons(M >= S['4'] + operations['5'], name="makespan_5")
model.addCons(M >= S['5'] + operations['6'], name="makespan_6")
model.addCons(M >= S['6'] + operations['7'], name="makespan_7")
model.addCons(M >= S['7'] + operations['8'], name="makespan_8")
# Minimize the makespan
model.setObjective(M, "minimize")
# Solve the model
model.optimize()
# Print the solution
if model.getStatus() == 'optimal':
    print("Objective: ", model.getObjVal())
else:
    print("No optimal solution found.")
```

**Corrected Response:**

### Step 1: Analyze the Problem
**Decision Variables:**
- $S_i$: Start time of operation $i$ (for all operations $i$).
- $M$: Makespan (total completion time).
- **$y_{ij}$: Binary variable indicating the sequencing of operations $i$ and $j$ on shared machines (1 if $i$ comes before $j$, 0 otherwise).**

**Objective:**
- Minimize the makespan $M$.

**Constraints:**
1. **Precedence Constraints:**
   - For Job 0:
     - Operation 1 must start at least 4 units after Operation 0 starts: $S_1 \geq S_0 + 4$
     - Operation 2 must start at least 1 unit after Operation 1 starts: $S_2 \geq S_1 + 1$
     - Operation 3 must start at least 6 units after Operation 2 starts: $S_3 \geq S_2 + 6$
     - Operation 4 must start at least 6 units after Operation 3 starts: $S_4 \geq S_3 + 6$
   - For Job 1:
     - Operation 1 must start at least 9 units after Operation 0 starts: $S_5 \geq S_4 + 9$
     - Operation 2 must start at least 1 unit after Operation 1 starts: $S_6 \geq S_5 + 1$
     - Operation 3 must start at least 4 units after Operation 2 starts: $S_7 \geq S_6 + 4$

2. **Machine Capacity Constraints:**
   - For each pair of operations on the same machine, ensure there is no overlap. Use a large constant (100000) to enforce sequencing logic.
   - For Operation 1 of Job 0 and Operation 3 of Job 0 on the same machine:
     $S_1 + 1 \leq S_3 + 100000 \cdot (1 - y_{13})$
     **$S_3 + 6 \leq S_1 + 100000 \cdot y_{13}$**
   - For Operation 1 of Job 0 and Operation 2 of Job 1 on the same machine:
     $S_1 + 1 \leq S_6 + 100000 \cdot (1 - y_{16})$
     **$S_6 + 4 \leq S_1 + 100000 \cdot y_{16}$**

3. **Makespan Constraints:**
   - The makespan must be at least as large as the completion time of every operation:
     - Operation 0 of Job 0: $M \geq S_0 + 4$) - Operation 1 of Job 0: $M \geq S_1 + 1$
     - Operation 2 of Job 0: $M \geq S_2 + 6$) - Operation 3 of Job 0: $M \geq S_3 + 6$
     - Operation 4 of Job 0: $M \geq S_4 + 8$) - **Operation 0 of Job 1: $M \geq S_5 + 9$**
     - **Operation 1 of Job 1: $M \geq S_6 + 1$ - Operation 2 of Job 1: $M \geq S_7 + 4$**
     - **Operation 3 of Job 1: $M \geq S_8 + 2$**

4. **Non-negativity and Continuity:**
   - All start times and the makespan must be non-negative and continuous variables.

### Step 2: Develop the Mathematical Model
**Sets:**
- No explicit sets are required here as the operations and their dependencies are explicitly defined.

**Parameters:**
- $p_i$: Processing time of operation $i$.

**Decision Variables:**
- $S_i$: Start time of operation $i$.
- $M$: Makespan.
- **$y_{ij}$: Binary variable for sequencing operations $i$ and $j$ on shared machines.**

**Objective:**
Minimize $M$.

**Constraints:**
1. **Precedence Constraints:**
   - For Job 0:
     $[S_1 \geq S_0 + 4]\[S_2 \geq S_1 + 1]\[S_3 \geq S_2 + 6]\[S_4 \geq S_3 + 6]$
   - For Job 1:
     $[S_5 \geq S_4 + 9]\[S_6 \geq S_5 + 1]\[S_7 \geq S_6 + 4]$

2. **Machine Capacity Constraints:**
   - For Operation 1 of Job 0 and Operation 3 of Job 0:
     $[S_1 + 1 \leq S_3 + 100000 \cdot (1 - y_{13})]$
     **$[S_3 + 6 \leq S_1 + 100000 \cdot y_{13}]$**
   - For Operation 1 of Job 0 and Operation 2 of Job 1:
     $[S_1 + 1 \leq S_6 + 100000 \cdot (1 - y_{16})]$
     **$[S_6 + 4 \leq S_1 + 100000 \cdot y_{16}]$**

3. **Makespan Constraints:**
   - For each operation:
     $[M \geq S_0 + 4]$  $[M \geq S_1 + 1]$  $[M \geq S_2 + 6]$
     $[M \geq S_3 + 6]$  $[M \geq S_4 + 8]$  **$[M \geq S_5 + 9]$**
     **$[M \geq S_6 + 1]$  $[M \geq S_7 + 4]$  $[M \geq S_8 + 2]$**

### Step 3: Provide the Corresponding Pyscipopt Python Code
```python
from pyscipopt import Model
model = Model("Job_Scheduling")
# Define operations and their processing times
operations = {
    '0': 4,  # Operation 0 of Job 0  '1': 1,  # Operation 1 of Job 0  '2': 6,  # Operation 2 of Job 0
    '3': 6,  # Operation 3 of Job 0  '4': 8,  # Operation 4 of Job 0  '5': 9,  # Operation 0 of Job 1
    '6': 1,  # Operation 1 of Job 1  '7': 4,  # Operation 2 of Job 1  '8': 2  # Operation 3 of Job 1
}
# Define start times for each operation
S = {op: model.addVar(vtype="C", name=f"S_{op}") for op in operations}
# Define the makespan
M = model.addVar(vtype="C", name="M")
# Binary variables for machine sequencing
y13 = model.addVar(vtype="B", name="y13")
y16 = model.addVar(vtype="B", name="y16")
# Precedence constraints for Job 0
model.addCons(S['1'] >= S['0'] + 4, name="precedence_0_1")
model.addCons(S['2'] >= S['1'] + 1, name="precedence_1_2")
model.addCons(S['3'] >= S['2'] + 6, name="precedence_2_3")
model.addCons(S['4'] >= S['3'] + 6, name="precedence_3_4")
# Precedence constraints for Job 1
model.addCons(S['6'] >= S['5'] + 9, name="precedence_5_6")
model.addCons(S['7'] >= S['6'] + 1, name="precedence_6_7")
model.addCons(S['8'] >= S['7'] + 4, name="precedence_7_8")
# Machine constraints for shared machines
model.addCons(S['1'] + 1 <= S['3'] + 100000 * (1 - y13), name="machine_1_before_3")
model.addCons(S['3'] + 6 <= S['1'] + 100000 * y13, name="machine_3_before_1")
model.addCons(S['1'] + 1 <= S['7'] + 100000 * (1 - y16), name="machine_1_before_6")
model.addCons(S['7'] + 4 <= S['1'] + 100000 * y16, name="machine_6_before_1")
# Makespan constraints
model.addCons(M >= S['0'] + operations['0'], name="makespan_0")
model.addCons(M >= S['1'] + operations['1'], name="makespan_1")
model.addCons(M >= S['2'] + operations['2'], name="makespan_2")
model.addCons(M >= S['3'] + operations['3'], name="makespan_3")
model.addCons(M >= S['4'] + operations['4'], name="makespan_4")
model.addCons(M >= S['5'] + operations['5'], name="makespan_5")
model.addCons(M >= S['6'] + operations['6'], name="makespan_6")
model.addCons(M >= S['7'] + operations['7'], name="makespan_7")
model.addCons(M >= S['8'] + operations['8'], name="makespan_8")
# Minimize the makespan
model.setObjective(M, "minimize")
# Solve the model
model.optimize()
# Print the solution
if model.getStatus() == 'optimal':
    print("Objective: ", model.getObjVal())
else:
    print("No optimal solution found.")
```

Figure 19: Wrong response and corrected response corresponding to example 2.

Therefore, the errors in the wrong response are localizable. If we fix only these localizable errors, the overall answer will be corrected.

Example 2 is a scheduling problem. The wrong response includes precedence constraints, machine capacity constraints, makespan constraints, and non-nagativity constraints. We observe minor errors in the machine capacity and makespan constraints. However, the errors in the makespan constraints do not affect the correctness of the precedence constraints, machine capacity constraints, or non-negativity constraints. Similarly, the errors in the machine capacity constraints do not affect the correctness of the precedence constraints, makespan constraints, or non-negativity constraints. Therefore, the errors in the wrong response are localizable. By fixing only these localizable errors, the overall solution can be corrected.

## D    REPRODUCIBILITY STATEMENT

Upon acceptance of this paper, we will publicly release the code, MIND-Train, MIND-Bench, and MIND-Qwen2.5-7B on GitHub and Hugging Face under the MIT License. All assets used in this research are properly credited.

## E    THE USE OF LARGE LANGUAGE MODELS (LLMS)

In this work, we used large language models (LLMs) solely as an auxiliary tool for checking grammar and improving the clarity of our writing. The research ideas, experiments, analyzes, and all scientific contributions were conducted entirely by the authors.

