# OpenReview forum: "Automated Optimization Modeling via a Localizable Error-Driven Perspective"
_ICLR.cc/2026/Conference — Submitted to ICLR 2026_

### Official Review · Reviewer_RdHx · 2025-10-27

**Soundness:** 2
**Presentation:** 3
**Contribution:** 3
**Rating:** 4
**Confidence:** 5

**Summary:**

This paper proposes MIND, a framework for training LLMs on automated optimization modeling. The approach is motivated by observing that errors in incorrect samples have an average error ratio of 0.33 across variables, constraints, and objectives, suggesting localized error patterns. The framework has two components: (1) an error-driven reverse data synthesis pipeline that generates training problems by incorporating identified error patterns through single-error and multi-error strategies, producing training instances; (2) DFPO, which combines RL with SFT by using a teacher LLM to correct failed responses. Evaluated on six benchmarks with Qwen2.5-7B-Instruct, MIND achieves 62.7% average accuracy versus SIRL's 60.2%. The paper also introduces MIND-Bench for out-of-distribution evaluation.

**Strengths:**

**Comprehensive empirical evaluation across multiple benchmarks:** The paper evaluates the proposed method on six diverse benchmarks (NL4Opt, IndustryOR, MAMO EasyLP/ComplexLP, OptMATH-Bench, OptiBench) and provides an additional out-of-distribution benchmark (MIND-Bench with 69 problems). The experimental scope is reasonably broad and covers different types of optimization problems.

**Consistent performance improvements over baselines:** The experimental results show consistent improvements across most benchmarks compared to prior training-based methods (ORLM, LLMOPT, OptMATH, SIRL). The improvements are particularly notable on challenging benchmarks like ComplexLP (+8.4% over SIRL) and OptMATH (+6.2% over SIRL), suggesting the approach may be effective for difficult problems.

**Clear presentation and structure:** The paper is generally well-written and clearly structured. The proposed data synthesis pipeline and the DFPO algorithm are explained with clarity, aided by helpful diagrams (Figures 2, 3, 4). The methodology section provides sufficient implementation details for understanding the approach.

**Weaknesses:**

**Insufficient motivation and questionable core assumption:** The paper's central claim that "modeling errors are localized and do not propagate" relies solely on Figure 1 showing an average error ratio of 0.33 across 100 samples. This does not prove localization—it could simply mean ~67% overall correctness. The claim that errors in variable definitions do not affect constraints is questionable; incorrect variable definitions would naturally propagate to dependent constraints. The paper provides no evidence that optimization modeling exhibits fundamentally different error patterns compared to other reasoning tasks.

**Lack of justification for single-error vs. multi-error decomposition:** The paper introduces two synthesis strategies without explaining why this decomposition is necessary. No theoretical rationale is provided, and critical ablations are missing: training with only single-error data, only multi-error data, or comparing their mixture versus each alone. Table 5 shows vastly different pass rates (39.88% vs 60.30%), but this is not discussed. It remains unclear whether the decomposition is principled or arbitrary.

**Incomplete related work coverage:** The paper omits several recent relevant works: (1) "Autoformulation of Mathematical Optimization Models Using LLMs"; (2) "CHAIN-OF-EXPERTS: When LLMs Meet Complex Operations Research Problems"; (3) "Step-Opt: Boosting Optimization Modeling in LLMs through Iterative Data Synthesis and Structured Validation".

**Performance ceiling imposed by teacher model with no mitigation strategy: ** The pipeline relies on teacher LLMs (DeepSeek-R1 for error identification, DeepSeek-V3 for correction), creating an inherent performance ceiling. The paper does not discuss this limitation or propose mechanisms to break through it. Unlike rejection sampling or other techniques that could potentially exceed teacher performance, the current approach appears bounded by teacher model quality, raising concerns about long-term scalability.

**Questions:**

**On the core motivation:** Why does an error ratio of 0.33 demonstrate "localized errors that do not propagate"? Can you provide evidence comparing error propagation patterns in optimization modeling versus other reasoning tasks, and explain why existing methods (ORLM + SIRL) cannot address limitations L1 and L2?

**On the single-error vs. multi-error decomposition:** What is the rationale for this decomposition, and can you provide ablations comparing training with only single-error data, only multi-error data, versus their mixture? How do you explain the large pass rate difference (39.88% vs 60.30%) between the two strategies in Table 5?

**On DFPO's contribution:** Can you provide comparisons with additional baselines such as "SFT→RL," "SFT→GRPO," and "SFT→filter→RL" to isolate DFPO's specific advantage beyond adding SFT loss for failed samples? How do you quantitatively verify that corrected responses remain close to the base model's distribution?

---

> ### Author Response · Authors · 2025-11-23
> **Official Comment by Authors**
>
> Dear Reviewer RdHx,
>
> Thanks for your significant review and for recognizing the clarity of our presentation and the effectiveness of our method. Your comments are highly constructive and helpful, and we address your concerns below.
>
> **Q1. Assumption of localized errors (weakness 1 and reviewer question 1)**
>
> **A1**: We sincerely apologize for any confusion caused by our original phrasing. To clarify the error ratio of 0.33, as defined in Section 3.2, the error ratio $\epsilon$ is computed as the proportion of incorrect modeling components relative to the total number of components in a **given erroneous formulation**. The reported value of 0.33—and in most cases, values even lower (typically around 0.2)—indicates that only a small fraction of the formulation is flawed, while the majority of components are correctly specified.
>
> To further investigate this phenomenon, we conducted a focused analysis on samples containing variable-related errors (e.g., incorrect variable types, invalid bounds, or missing initialization). Since variable definition typically occurs in the first or second step of optimization modeling, one might expect such early-stage errors to propagate and corrupt downstream elements like constraints or objectives. However, we found that these variable-level mistakes rarely affect the correctness of other components: constraints and objectives are often still formulated accurately relative to the erroneous variable definitions. This empirical observation that errors tend to remain confined to specific semantic units without cascading through the solution, is what we term "localizable error".
>
> **Q2: Differences between mathematical reasoning and optimization modeling tasks (weakness 1 and reviewer question 1)**
>
> **A2.** Thank you for this insightful question. To clarify the distinction in error propagation behavior between general mathematical reasoning and optimization modeling, we include a side-by-side illustrative example in Figure 1 of the revised manuscript.
>
> **(1) General Mathematical Reasoning (adapted from StepMathAgent [1]):**
> _Problem:_ If x and y satisfy $x,y\leq2$ and $x+y\geq3$, find the maximum of $x+3y$.
>
> - **Step 1:** “When x+y=3, the value of x+3y is maximized.” ❌
> - **Step 2:** “Maximum occurs at x=2,y=1.” ❌
> - **Step 3:** “Thus, max value is 2+3=5.” ❌
> - **Final answer:** [5] ❌
>
> Here, an early logical flaw (incorrect assumption about the binding constraint) cascades through all subsequent steps, rendering the entire solution invalid. Correcting the answer requires **reconstructing the full reasoning chain**.
>
> **(2) Optimization Modeling (simplified from MAMO ComplexLP):**
> _Problem:_ Minimize cost to meet $\geq83g$ protein using Steak (14g/\\$4) and Tofu (2g/\\$6).
>
> - **Step 1 (Parameters):** Correctly defined. ✅
> - **Step 2 (Decision Variables):** “x[i]: integer amount of food i.” ❌ _(Should be continuous)_
> - **Step 3 (Objective):** Correctly formulated. ✅
> - **Step 4 (Constraints):** Protein constraint correctly specified. ✅
>
> In this case, only the variable type is erroneous; all other components—parameters, objective, and constraints—remain semantically and structurally sound, even relative to the (flawed) variable definition. The error is **localized**, and correction requires **only a targeted edit** (e.g., changing “integer” to “continuous”), without rewriting the rest of the model.
>
> This contrast highlights a key structural property of optimization modeling: due to the modular design of formulations (variables, constraints, objectives as loosely coupled components), errors often do not propagate globally. This phenomenon, termed "localizable error", enables efficient correction strategies like ours.
>
> For further discussion on error locality in more complex optimization scenarios, we additionally provide two detailed case studies (in Appendix C.6), where we analyze multi-component failure modes and reaffirm the limited propagation pattern. In the first case, we show that errors in variables do not affect the correctness of the parameters, objective function, or constraints. In the second case, we show that errors in constraints do not affect the correcteness of other constraints.

---

> ### Author Response · Authors · 2025-11-23
> **Official Comment by Authors**
>
> **Q3. Single-error vs. multi-error strategies ablation study (weakness 2 and reviewer question 2)**
>
> **A3**: Thanks for the insightful suggestion. We partition the MIND-3K dataset into two disjoint subsets: **MIND-Single-1.5K** (containing instances derived from single-error pattern) and **MIND-Multi-1.5K** (comprising instances derived from multiple-error patterns). As shown in the table below, models trained on either subset alone underperform compared to training on the full **MIND-3K** (which we refer to as **MIND-Mix-3K**). Our results also show that training on MIND-Single-1.5K leads to better performance than training on MIND-Multi-1.5K. We hypothesize that this disparity arises because LLMs struggle to learn effectively when trained directly on highly challenging datasets. To further substantiate this hypothesis, we evaluate Qwen2.5-7B-Instruct on both datasets. The model achieves an average accuracy on MIND-Single-1.5K that is higher than on MIND-Multi-1.5K by 11.7 percentage points. This pronounced accuracy gap corroborates our claim that multi-error reverse data synthesis generates datasets that are substantially more diverse and difficult than those produced by single-error synthesis.
>
> | Data                | NL4OPT  | IndustryOR | EasyLP  | ComplexLP | OptMATH | OptiBench | Macro AVG |
> |--------------------|---------|------------|---------|-----------|---------|-----------|-----------|
> | MIND-Single-1.5K    | 91.4%  | 29.0%      | 90.4%  | 40.9%     | 8.4%   | 53.6%     | 52.3%     |
> | MIND-Multi-1.5K     | 91.4%  | 29.0%      | 90.2%  | 33.0%     | 6.0%   | 54.0%     | 50.6%     |
> | MIND-Mix-3K         | 94.3%  | 30.0%      | 90.8%  | 39.9%     | 7.8%   | 55.5%     | 53.1%     |
>
> **Q4. Why ORLM and SIRL cannot address the limitations? (reviewer question 2)**
>
> **A4**: We sincerely thank the reviewer for this insightful question. To clarify, our work addresses two key limitations in current optimization-focused LLM training:
>
> - **(L1)** _The sparsity of error-specific problems_: real-world or synthetic datasets rarely contain fine-grained examples of common modeling mistakes paired with their corrections.
> - **(L2)** _The sparse rewards associated with difficult problems_: due to the complexity of automated optimization modeling tasks, LLMs often struggle to receive sufficient learning signals through reinforcement learning alone on difficult problems.
>
> We now explain why ORLM and SIRL, while valuable, do not fully resolve these issues:
>
> (1) ORLM is primarily a _data diversification_ technique. It enriches training data by expanding scenarios—e.g., rephrasing prompts, varying objectives/constraints, or combining modeling paradigms—but it **does not explicitly generate or emphasize error-centric instances**. Consequently, it lacks the mechanism to expose the model to _systematic modeling failures_ and their corrections, which is essential for learning from errors (i.e., addressing **L1**).
>
> (2) SIRL tackles **L2** by designing _dense, skill-aware rewards_: it grants bonus rewards when the model employs advanced techniques (e.g., Big-M formulations, piecewise linearization). While this encourages sophisticated modeling behavior, its effectiveness **still hinges on the model’s ability to generate plausible rollouts in the first place**. On highly challenging problems, where rollouts cannot efficiently sample correct responses, even a well-designed reward cannot compensate for poor exploration, limiting its capacity to overcome sparse learning signals in practice.
>
> In contrast, **our DFPO framework directly targets both L1 and L2**:
>
> - Through error-driven reverse synthesis, we construct a training set rich in localized errors and expert corrections (addressing L1);
> - By using local feedback on corrected responses, we provide a dense and stable supervision signal for difficult problems. (alleviating L2).
>
> Thus, while ORLM and SIRL improve data coverage and reward shaping respectively, they operate orthogonal to the error-centric learning paradigm we propose.

---

> ### Author Response · Authors · 2025-11-23
> **Official Comment by Authors**
>
> **Q5.Pass rate discuss (reviewer question 2)**
>
> **A5**: We sincerely apologize for the confusion caused by an unintentional inconsistency in the LLM versions used during data synthesis. The difference in pass rates between the single-error and multi-error reverse synthesis strategies stems from a minor API version update of the DeepSeek-R1 model. The single-error data was synthesized using DeepSeek-R1 (version: DeepSeek-R1-0528), while the multi-error data was synthesized using a later, more capable version: DeepSeek-R1 (version: DeepSeek-V3.1-Think). This version upgrade led to improved reasoning and code generation capabilities, which explains the higher pass rate observed in the multi-error setting. The performance gap is therefore attributable to model capability differences, not the synthesis strategy itself.
>
> We deeply regret this oversight in experimental control and appreciate the reviewer’s careful attention to detail. In the revised manuscript, we have clarified this point in Appendix A.3 and ensured consistent model versions in all ablation studies moving forward. Thank you for helping us improve the rigor of our work.
>
> **Q6. Ablation on post-training methods (reviewer question 3)**
>
> **A6**: We sincerely thank you for your careful review, which is invaluable for enhancing the credibility of our results. Following your suggestion, and considering the SIRL is a baseline using REINFORCE++ without prior SFT, we have added two additional baseline, SFT and SFT+GRPO, to isolate DFPO's specific advantage beyond adding SFT loss for detailed samples.
>
> We conduct two baselines, SFT and SFT+GRPO, on the same training dataset for the post-training methods ablation study. For SFT, following the experimental settings in OptMATH, we fine-tune Qwen2.5-7B-Instruct for 3 epochs to establish the SFT baseline. For SFT+GRPO, we use the SFT checkpoint as a warm start and continue training with GRPO for 26 additional epochs, yielding the full sequential pipeline. The training configuration details have been updated in this revised version.
>
> We observe that **SFT alone yields limited performance**, particularly on complex benchmarks. However, when used as a warm start for GRPO, it leads to notable improvements, demonstrating the benefit of initializing policy optimization with supervised signals. Nevertheless, **SFT+GRPO still underperforms DFPO** on the most challenging tasks, highlighting DFPO’s superior ability to leverage feedback and preserve modeling fidelity during training.
>
> | Method   | NL4OPT | IndustryOR | EasyLP | ComplexLP | OptMATH | OptiBench | Macro AVG |
> | -------- | ------ | ---------- | ------ | --------- | ------- | --------- | --------- |
> | DFPO     | 96.7%  | 34.0%      | 92.2%  | 60.1%     | 36.7%   | 56.7%     | **62.7%** |
> | DAPO     | 96.7%  | 33.0%      | 92.5%  | 58.6%     | 26.5%   | 57.5%     | 60.8%     |
> | SFT      | 92.2%  | 31.0%      | 85.4%  | 37.4%     | 9.6%    | 55.9%     | 51.9%     |
> | SFT+GRPO | 93.9%  | 34.0%      | 90.2%  | 54.7%     | 28.3%   | 57.0%     | 59.7%     |

---

> ### Author Response · Authors · 2025-11-23
> **Official Comment by Authors**
>
> **Q7. Missing valuable and significant relevant work (weakness 3)**
>
> **A7**: Thank you for pointing out these excellent related works. **Autoformulator**, **Chain-of-Experts**, and **Step-Opt** have each advanced automated optimization modeling through distinct and insightful approaches, and have made significant contributions to the growing field of LLM-based optimization. We have cited these works and supplemented the discussions in the revised version of our paper. The table below shows the comparison between MIND and these excellent works.
>
> | Category    | Methods              | NL4Opt | IndustryOR | EasyLP | ComplexLP | OptMATH | OptiBench | Macro AVG |
> | ----------- | -------------------- | ------ | ---------- | ------ | --------- | ------- | --------- | --------- |
> | Proprietary | GPT-4*               | 89.0%  | 33.0%      | 87.3%  | 49.3%     | 16.6%   | 68.6%     | 57.4%     |
> |             | OpenAI o3*           | 69.4%  | 44.0%      | 77.1%  | 51.2%     | 44.0%   | 58.6%     | 57.4%     |
> | Open-Source | Deepseek-V3*         | 95.9%  | 37.0%      | 88.3%  | 50.2%     | 44.0%   | 71.6%     | 64.5%     |
> |             | Deepseek-R1*         | 82.4%  | 45.0%      | 87.2%  | 67.9%     | 40.4%   | 66.4%     | 61.9%     |
> |             | Qwen2.5-7B-Instruct  | 89.0%  | 24.0%      | 89.4%  | 31.5%     | 3.0%    | 53.2%     | 48.4%     |
> |             | Qwen3-8B             | 72.2%  | 14.0%      | 76.8%  | 17.2%     | 7.2%    | 36.5%     | 37.3%     |
> | TTS-based   | Autoformulator*      | 92.6%  | 48.0%      | -      | 62.3%     | -       | -         | -         |
> |             | Chain-of-Experts*    | 64.2%  | -          | -      | 40.2%     | -       | -         | -         |
> |             | OptiMUS*             | 78.8%  | 31.0%      | 77.0%  | 43.6%     | 20.2%   | 45.8%     | 49.4%     |
> | Fine-Tuning | ORLM-Llama3-8B*      | 85.7%  | 24.0%      | 82.3%  | 37.4%     | 2.6%    | 51.1%     | 47.2%     |
> |             | Step-Opt-Llama3-8B*  | 84.5%  | 36.4%      | 85.3%  | 61.6%     | -       | -         | -         |
> |             | LLMOPT-Qwen2.5-14B*  | 80.3%  | 29.0%      | 89.5%  | 44.1%     | 12.5%   | 53.8%     | 51.1%     |
> |             | OptMATH-Qwen2.5-7B*  | 94.7%  | 20.0%      | 86.5%  | 51.2%     | 24.4%   | 57.9%     | 55.8%     |
> |             | OptMATH-Qwen2.5-32B* | 95.9%  | 31.0%      | 89.9%  | 54.1%     | 34.7%   | 66.1%     | 62.0%     |
> | RLVR        | SIRL-Qwen2.5-7B*     | 96.3%  | 33.0%      | 91.7%  | 51.7%     | 30.5%   | 58.0%     | 60.2%     |
> |             | SIRL-Qwen2.5-32B*    | 98.0%  | 42.0%      | 94.6%  | 61.1%     | 45.8%   | 67.4%     | 68.2%     |
> | Ours        | MIND-Qwen2.5-7B      | 96.7%  | 34.0%      | 92.2%  | 60.1%     | 36.7%   | 56.7%     | 62.7%     |
> |             | **MIND-Qwen3-8B**    | 95.1%  | 42.0%      | 92.7%  | 76.8%     | 41.0%   | 62.0%     | **68.3%** |
>
> **Q8. Teacher bound (weakness 4)**
>
> **A8**: We sincerely appreciate this insightful comment. We explain how MIND mitigates the teacher model performance ceiling problem as follows:
>
> **(1) DeepSeek-V3 for correction (training)**: We emphasize a key distinction: directly solving complex optimization modeling problems from scratch is significantly more challenging than performing ground-truth-guided correction, i.e., revising an existing erroneous formulation into a correct one when provided with the true solution​. In our pipeline, DeepSeek-V3 operates in the latter regime. Based on our statistics, we find that DeepSeek-V3 is sufficiently reliable for the correction task, with over 90% of corrections converging within a single round and all corrections within three rounds. Therefore, in training stage, the teacher model is definitely not the performance bottleneck.
>
> **(2) DeepSeek-R1 for error identification (data synthesis)**: The quality control phase of the error-driven reverse data synthesis pipeline employs a rejection sampling methodology [2]. Each synthesized instance undergoes both code validation and bidirectional validation. In the bidirectional validation stage, the data synthesis LLM generates an alternative solution using only the problem statement, without access to the ground-truth solution. The objective values of the original ground-truth solution (from the reverse synthesis stage) and the newly generated alternative solution are subsequently compared. Only those instances for which the objective values are equal are retained in the MIND-Train dataset for subsequent post-training.

---

> ### Author Response · Authors · 2025-11-23
> **Official Comment by Authors**
>
> **Q9. Quantitative distribution shift (reviewer question 3)**
>
> **A9**: Thank you for this important comment. We use an embedding-based text similarity method [3] to evaluate the distribution discrapancy between wrong responses and corrected responses. Specifically, we first obtain embeddings for the responses using an embedding model, and then compute the consine similarity between them. To further illustrate this, we conduct the following experiment: we randomly sample 300 tuples from the post-training stage of Qwen2.5-7B-Instruct, each consisting of a question, a wrong response, and a corrected response. Given the question, we prompt Deepseek-V3 to generate a response independently. We then use Qwen3-Embedding-8B [4] to obtain embeddings for both the wrong response, the corrected response and  the Deepseek-V3 response. Finally, we compute the average cosine similarity between the wrong response and the corrected response, as well as between the wrong response and the Deepseek-V3 response. The results are shown in the following table: the corrected responses has a higher cosine similiarty with the wrong responses, compared to the Deepseek-V3 responses.
>
> | Comparison                             | Average Cosine Similarity |
> | -------------------------------------- | ------------------------- |
> | Wrong response vs Corrected response   | 0.963                     |
> | Wrong response vs Deepseek-V3 response | 0.908                     |
>
> Best regards,
>
> The authors.
>
>
>
> [1] Shao, Zhihong, et al. "Deepseekmath: Pushing the limits of mathematical reasoning in open language models." arXiv preprint arXiv:2402.03300 (2024).
>
> [2] Liu, Haoxiong, et al. "Augmenting math word problems via iterative question composing." *Proceedings of the AAAI Conference on Artificial Intelligence*. Vol. 39. No. 23. 2025.
>
> [3] Reimers, Nils, and Iryna Gurevych. "Sentence-bert: Sentence embeddings using siamese bert-networks." arXiv preprint arXiv:1908.10084 (2019).
>
> [4] Bai, Jinze, et al. "Qwen technical report." arXiv preprint arXiv:2309.16609 (2023).

---

> ### Author Response · Authors · 2025-11-27
> **Looking Forward to Your Further Comments**
>
> Dear Reviewer RdHx,
>
> Thanks again for your time and for providing encouraging and constructive comments, which have helped us improve both the clarity and credibility of our paper. As the rebuttal phase is approaching (due December 3), we are eagerly looking forward to your further comments and/or questions.
>
> We have carefully addressed your concerns, including the illustration of the difference in error propagation between mathematical reasoning questions and optimization modeling questions, discussion of highly relevant works, explanation of the performance ceiling of MIND, and ablations on additional components of MIND. We sincerely hope that our rebuttal and revised manuscript adequately reflect these improvements. If possible, we would greatly appreciate it if you could consider further raising your scores. If not, please let us know any further comments, and we will continue to actively respond and improve our manuscript.
>
> Best regards,
>
> The authors.

---

### Official Review · Reviewer_kVVf · 2025-10-29

**Soundness:** 2
**Presentation:** 3
**Contribution:** 2
**Rating:** 4
**Confidence:** 4

**Summary:**

This paper addresses two issues in LLM-based optimization modeling: a scarcity of error-specific training data (L1) and sparse rewards in reinforcement learning (L2). The authors present MIND, a framework motivated by their claim that modeling errors are "localizable". The MIND framework consists of two components: (1) an "error-driven reverse data synthesis" pipeline, which was used to create a new dataset (MIND-Train) by injecting identified error patterns into new problems, and (2) a post-training algorithm, DFPO, which uses a "teacher" LLM to generate an SFT loss signal when all RL rollouts fail. The authors report that this framework achieves improved performance over baselines. The paper also includes a new benchmark, MIND-Bench.

**Strengths:**

1. The paper contributes MIND-Bench, a new, open-sourced benchmark for evaluating generalization. It is curated from textbooks and real-world industry scenarios, providing a valuable out-of-distribution test set for the community.
2. The proposed method achieves strong empirical results. On the same 7B model, MIND outperforms prior state-of-the-art training-based methods (like SIRL and OptMATH) in macro-average performance across six benchmarks.
3. The DFPO algorithm is an interesting and pragmatic approach. It combines SFT and RL to effectively address sparse rewards by applying a dense SFT loss when all RL rollouts fail. This provides learning signals for difficult problems, which is shown to be effective on the complex benchmark.

**Weaknesses:**

1. The paper's claim that errors are "localizable" is unsurprising—competent LLMs naturally make partial errors rather than completely wrong formulations. The sole evidence (Figure 1: error ratios from 100 samples) only computes the fraction of erroneous components, providing no analysis of error dependencies or propagation. The critical claim that errors "do not propagate" is never validated. Without showing that errors in one component (e.g., variables) don't cause errors in dependent components (e.g., constraints), the motivation for the error-driven approach remains unsubstantiated.
2.  MIND-Bench's contribution is questionable. With only 69 problems, its statistical reliability is limited. More importantly, its stated sources ("textbooks or industry scenarios") directly overlap with existing benchmarks (OptiBench, IndustryOR), and the paper fails to articulate what unique gap it addresses. The necessity of this small benchmark remains unjustified.
3.  The paper provides no experimental justification for DFPO's complexity over simpler alternatives. The key idea—applying SFT when all RL rollouts fail—lacks comparison to natural baselines: (1) sequential training (SFT pre-training → RL), and (2) data stratification (filtering problems by difficulty, applying SFT/RL separately). Without these ablations, there is no evidence that interleaving provides any benefit over decoupled approaches.

**Questions:**

1. The evidence in Figure 1 only shows a static error ratio. Can you provide more rigorous evidence to support your central claim that errors "do not propagate"? This is the key justification for your pipeline, but it is not substantiated by the current analysis.
2. How does the proposed interleaved DFPO algorithm compare to simpler, decoupled baselines? Specifically: (a) A standard sequential pipeline (full SFT, then RL)? (b) A data stratification strategy (filtering by difficulty, then applying SFT/RL separately)?
3. Given its small scale (69 problems) and overlapping sources with larger benchmarks like OptiBench and IndustryOR, what unique gap does MIND-Bench fill, and how can its results be considered statistically significant?
4. The paper fails to cite or compare against highly relevant work.  Key omissions include:
[1] Chain-of-experts: When LLMs meet complex operations research problems. (2023), which addresses LLM approaches for complex OR problems.
[2] Step-Opt: Boosting Optimization Modeling in LLMs through Iterative Data Synthesis and Structured Validation (2025), which directly tackles iterative data synthesis and structured validation.

---

> ### Author Response · Authors · 2025-11-23
> **Official Comment by Authors**
>
> Dear Reviewer kVVf,
>
> We sincerely thank the reviewer for dedicating time and effort to evaluating our paper. We greatly appreciate the insightful suggestions for enhancing its clarity. Below, we address the concerns raised.
>
> **Q1. Evidence of error propagate (weakness 1 and reviewer question 1)**
>
> **A1**: We thank the reviewer for this insightful observation. We fully agree that modern LLMs typically produce partial or localized errors rather than entirely incorrect formulations. This phenomenon is central to our work and has been consistently observed in our preliminary experiments. By "error locality" and "limited propagate", we mean that modeling errors tend to be confined to specific parts of the response and do not necessarily corrupt the correctness of other components. In other words, an error in one part of the formulation (e.g., a constraint) often leaves other parts (e.g., variables and optimization objective) intact and logically sound. This limited propagation enables targeted correction without requiring full re-generation.
>
> To further illustrate the dependency or propagation patterns of such errors, we additionally provide two detailed case studies (in Appendix C.6), which are complex optimization modeling scenarios in the revised manuscript. In the first case, we show that errors in variables do not affect the correctness of the parameters, objective function, or constraints. In the second case, we show that errors in constraints do not affect the correcteness of other constraints. These examples demonstrate how errors can arise in isolation and how local edits suffice to recover a fully correct formulation, thereby empirically supporting our core assumption of error locality and limited error propagation. The questions corresponding to the cases as follows:
>
> **(1) Case 1**: Imagine you are a dietitian and you have been tasked with creating a meal plan for a bodybuilder. You have six food items to choose from: Steak, Tofu, Chicken, Broccoli, Rice, and Spinach. Each food provides certain amounts of protein, carbohydrates, and calories, and each has its own cost.\n\nHere's the nutritional value and cost of each food:\n\n- Steak: It gives you 14 grams of protein, 23 grams of carbohydrates, and 63 calories for \\$4.\n- Tofu: It offers 2 grams of protein, 13 grams of carbohydrates, and 162 calories for \\$6.\n- Chicken: It packs a punch with 17 grams of protein, 13 grams of carbohydrates, and gives you 260 calories for \\$6.\n- Broccoli: It provides 3 grams of protein, a mere 1 gram of carbohydrates, and 55 calories for \\$8.\n- Rice: It gives a hearty 15 grams of protein, 23 grams of carbohydrates, and 231 calories for \\$8.\n- Spinach: It provides 2 grams of protein, 8 grams of carbohydrates, and a huge 297 calories for just \\$5.\n\nYour goal is to ensure that the bodybuilder gets at least 83 grams of protein, 192 grams of carbohydrates, and 2089 calories from whatever combination of these foods you choose. The challenge is to keep the cost as low as possible while meeting these nutritional targets. \n\nWhat is the minimum cost to meet these nutritional requirements with the available food options?

---

> ### Author Response · Authors · 2025-11-23
> **Official Comment by Authors**
>
> **(2) Case 2**: The manufacturing facility produces custom components for two jobs, Job 0 and Job 1, each consisting of a sequence of operations that must be performed in a specific order. The goal is to schedule these operations to minimize the total completion time (makespan) while satisfying all operational constraints. Job 0 has five operations with processing times: Operation 0 takes 4 units, Operation 1 takes 1 unit, Operation 2 takes 6 units, Operation 3 takes 6 units, and Operation 4 takes 8 units. Job 1 has four operations with processing times: Operation 0 takes 9 units, Operation 1 takes 1 unit, Operation 2 takes 4 units, and Operation 3 takes 2 units.\n\nPrecedence constraints ensure that operations within each job are performed in sequence with specific gaps. For Job 0, Operation 1 must start at least 4 units after Operation 0 starts, Operation 2 must start at least 1 unit after Operation 1 starts, Operation 3 must start at least 6 units after Operation 2 starts, and Operation 4 must start at least 6 units after Operation 3 starts. For Job 1, Operation 1 must start at least 9 units after Operation 0 starts, Operation 2 must start at least 1 unit after Operation 1 starts, and Operation 3 must start at least 4 units after Operation 2 starts.\n\nMachine capacity constraints ensure that operations assigned to the same machine do not overlap. Binary variables determine the order of operations on shared machines. For example, if Operation 1 of Job 0 and Operation 3 of Job 0 are on the same machine, one must complete at least 6 units before the other starts. Similarly, if Operation 1 of Job 0 and Operation 2 of Job 1 are on the same machine, one must complete at least 4 units before the other starts. These constraints apply to all operation pairs on shared machines, ensuring no overlap and maintaining required time gaps. The large constant of 100,000 is used in these constraints to enforce the sequencing logic by ensuring that the constraints are only active when the binary variable is set to 1.\n\nThemakespan must be at least as large as the completion time of every operation. Specifically, the makespan must be at least 4 units after Operation 0 of Job 0 starts, 1 unit after Operation 1 of Job 0 starts, 6 units after Operation 2 of Job 0 starts, 6 units after Operation 3 of Job 0 starts, 8 units after Operation 4 of Job 0 starts, 9 units after Operation 0 of Job 1 starts, 1 unit after Operation 1 of Job 1 starts, 4 units after Operation 2 of Job 1 starts, and 2 units after Operation 3 of Job 1 starts.\n\nThe objective is to determine the start times for all operations and the sequence of operations on shared machines to minimize the makespan while respecting all precedence constraints, machine capacity constraints, and the makespan definition. Binary variables are used to enforce the correct sequencing of operations on shared machines, with a large constant of 100,000 ensuring the constraints are properly applied. The start times for all operations and the makespan must be non-negative. The binary variables are explicitly defined for all relevant operation pairs on shared machines, and the start times and makespan are continuous variables.

---

> ### Author Response · Authors · 2025-11-23
> **Official Comment by Authors**
>
> **Q2. Reliability of MIND-Bench (weakness 2 and reviewer question 3)**
>
> **A2**: We sincerely appreciate the reviewer’s thoughtful comment. We fully acknowledge that MIND-Bench is relatively small in scale (69 problems) and that some of its problem sources may partially overlap with existing benchmarks in terms of high-level domains. However, MIND-Bench was intentionally designed with a different focus: not scale, but label reliability and cross-domain scenario diversity. It spans a wide range of real-world contexts (over 12 domains), including manufacturing, finance, transportation, agriculture, etc. More critically, every instance in MIND-Bench has been manually curated, expert-verified, and accompanied by executable ground-truth code solutions, ensuring high fidelity in both problem formulation and solution correctness. This rigorous annotation process makes MIND-Bench particularly well-suited for fine-grained, trustworthy evaluation of optimization modeling capabilities, especially when assessing subtle errors in variable definition, constraint logic, or objective alignment.
>
> Besides, we emphasize that MIND-Bench is not the primary basis for our core claims. The main evaluation of out-of-distribution generalization also relies on established, large-scale benchmarks: NL4OPT, IndustryOR, MAMO (EasyLP & ComplexLP), OptMATH-Bench, and OptiBench.
>
> **Q3. Ablation study of post-training methods (weakness 3 and reviewer question 2)**
>
> **A3**: We sincerely thank you for your careful review, which is invaluable for evaluating the effectiveness of our proposed DFPO. Following the suggestion, we conducted additional experiments to supplementary the post-training ablation study baselines (1) sequential training, and (2) data stratification.
>
> **(1) sequential training**: We conduct two baselines, SFT and SFT+GRPO, on the same training dataset for the post-training methods ablation study. For SFT, following the experimental settings in OptMATH, we fine-tune Qwen2.5-7B-Instruct for 3 epochs to establish the SFT baseline. For SFT+GRPO, we use the SFT checkpoint as a warm start and continue training with GRPO for 26 additional epochs, yielding the full sequential pipeline. The training configuration details have been updated in this revised version;
>
> **(2) data stratification**: We position **DAPO** as a representative baseline for _data stratification_ in optimization-aware training. DAPO employs a dynamic sampling strategy that adaptively filters out training instances where all model rollouts are either uniformly correct or uniformly incorrect. By focusing training on challenging and capable-of-solving samples, DAPO effectively performs online data stratification, prioritizing high-utility examples that drive learning.
>
> We observe that **SFT alone yields limited performance**, particularly on complex benchmarks. However, when used as a warm start for GRPO, it leads to notable improvements, demonstrating the benefit of initializing policy optimization with supervised signals. Nevertheless, **SFT+GRPO still underperforms DFPO** on the most challenging tasks, highlighting DFPO’s superior ability to leverage feedback and preserve modeling fidelity during training.
>
> | Method   | NL4OPT | IndustryOR | EasyLP | ComplexLP | OptMATH | OptiBench | Macro AVG |
> | -------- | ------ | ---------- | ------ | --------- | ------- | --------- | --------- |
> | DFPO     | 96.7%  | 34.0%      | 92.2%  | 60.1%     | 36.7%   | 56.7%     | **62.7%** |
> | DAPO     | 96.7%  | 33.0%      | 92.5%  | 58.6%     | 26.5%   | 57.5%     | 60.8%     |
> | SFT      | 92.2%  | 31.0%      | 85.4%  | 37.4%     | 9.6%    | 55.9%     | 51.9%     |
> | SFT+GRPO | 93.9%  | 34.0%      | 90.2%  | 54.7%     | 28.3%   | 57.0%     | 59.7%     |
>
> We additionally provide more ablation studies, including the reward weight sensitivity analysis (Appendix C.3), data synthesis strategies (Appendix C.4), and the base models (Section 5.2).

---

> ### Author Response · Authors · 2025-11-23
> **Official Comment by Authors**
>
> **Q4. Discussion of highly relevant work (reviewer question 4)**
>
> **A4**: We sincerely thank the reviewer for highlighting these relevant and impressive works. We fully agree that Chain-of-Experts and Step-Opt have made significant contributions to the emerging field of LLM-based optimization modeling. In the revised version, we have supplemented the discussion with Chain-of-Experts and Step-Opt in the **Introduction**, **Related Work**, and **Experiments** sections. The table below show the results of MIND-Qwen2.5-7B and MIND-Qwen3-8B, compared with these works.
>
> | Category    | Methods              | NL4Opt | IndustryOR | EasyLP | ComplexLP | OptMATH | OptiBench | Macro AVG |
> | ----------- | -------------------- | ------ | ---------- | ------ | --------- | ------- | --------- | --------- |
> | Proprietary | GPT-4*               | 89.0%  | 33.0%      | 87.3%  | 49.3%     | 16.6%   | 68.6%     | 57.4%     |
> |             | OpenAI o3*           | 69.4%  | 44.0%      | 77.1%  | 51.2%     | 44.0%   | 58.6%     | 57.4%     |
> | Open-Source | Deepseek-V3*         | 95.9%  | 37.0%      | 88.3%  | 50.2%     | 44.0%   | 71.6%     | 64.5%     |
> |             | Deepseek-R1*         | 82.4%  | 45.0%      | 87.2%  | 67.9%     | 40.4%   | 66.4%     | 61.9%     |
> |             | Qwen2.5-7B-Instruct  | 89.0%  | 24.0%      | 89.4%  | 31.5%     | 3.0%    | 53.2%     | 48.4%     |
> |             | Qwen3-8B             | 72.2%  | 14.0%      | 76.8%  | 17.2%     | 7.2%    | 36.5%     | 37.3%     |
> | TTS-based   | Autoformulator*      | 92.6%  | 48.0%      | -      | 62.3%     | -       | -         | -         |
> |             | Chain-of-Experts*    | 64.2%  | -          | -      | 40.2%     | -       | -         | -         |
> |             | OptiMUS*             | 78.8%  | 31.0%      | 77.0%  | 43.6%     | 20.2%   | 45.8%     | 49.4%     |
> | Fine-Tuning | ORLM-Llama3-8B*      | 85.7%  | 24.0%      | 82.3%  | 37.4%     | 2.6%    | 51.1%     | 47.2%     |
> |             | Step-Opt-Llama3-8B*  | 84.5%  | 36.4%      | 85.3%  | 61.6%     | -       | -         | -         |
> |             | LLMOPT-Qwen2.5-14B*  | 80.3%  | 29.0%      | 89.5%  | 44.1%     | 12.5%   | 53.8%     | 51.1%     |
> |             | OptMATH-Qwen2.5-7B*  | 94.7%  | 20.0%      | 86.5%  | 51.2%     | 24.4%   | 57.9%     | 55.8%     |
> |             | OptMATH-Qwen2.5-32B* | 95.9%  | 31.0%      | 89.9%  | 54.1%     | 34.7%   | 66.1%     | 62.0%     |
> | RLVR        | SIRL-Qwen2.5-7B*     | 96.3%  | 33.0%      | 91.7%  | 51.7%     | 30.5%   | 58.0%     | 60.2%     |
> |             | SIRL-Qwen2.5-32B*    | 98.0%  | 42.0%      | 94.6%  | 61.1%     | 45.8%   | 67.4%     | 68.2%     |
> | Ours        | MIND-Qwen2.5-7B      | 96.7%  | 34.0%      | 92.2%  | 60.1%     | 36.7%   | 56.7%     | 62.7%     |
> |             | **MIND-Qwen3-8B**    | 95.1%  | 42.0%      | 92.7%  | 76.8%     | 41.0%   | 62.0%     | **68.3%** |
>
> Best regards,
>
> The authors.

---

> ### Author Response · Authors · 2025-11-27
> **Looking Forward to Your Further Comments**
>
> Dear Reviewer kVVf,
>
> Thanks again for your time and for providing encouraging and constructive comments, which have helped us improve both the clarity and credibility of our paper. As the rebuttal phase is approaching (due December 3), we are eagerly looking forward to your further comments and/or questions.
>
> We have carefully addressed your concerns, including the evidence of limited error propagation, discussion of highly relevant works, and ablations on additional components of MIND. We sincerely hope that our rebuttal and revised manuscript adequately reflect these improvements. If possible, we would greatly appreciate it if you could consider further raising your scores. If not, please let us know any further comments, and we will continue to actively respond and improve our manuscript.
>
> Best regards,
>
> The authors.

---

### Official Review · Reviewer_JHsT · 2025-10-31

**Soundness:** 3
**Presentation:** 3
**Contribution:** 3
**Rating:** 6
**Confidence:** 3

**Summary:**

This work introduces MIND, an error-driven learning framework for automated optimization modeling. Building on the key observation that modeling errors are often localizable, the authors propose an reverse data synthesis pipeline targeting common error patterns, and Dynamic Supervised FineTuning Policy Optimization (DFPO) to address the challenges of sparse rewards and distribution shift. Experiments demonstrate that MIND consistently improves automated modeling performance across diverse benchmarks and achieve notably greater improvements on more challenging problems compared to existing methods.

**Strengths:**

The authors analyze the error patterns in automated optimization modeling and identify the localizable nature of error propagation. Building on this insight, they design an error-driven reverse data synthesis process that focuses on common error types to generate more challenging and informative training data. In addition, the proposed DFPO strategy effectively complements existing reinforcement learning methods by alleviating issues related to sparse rewards and distributional shift, using continuous reward shaping and dynamic supervised fine-tuning to promote more stable and efficient learning. Finally, the experimental evaluation is comprehensive, covering a wide range of benchmarks from simple to complex scenarios and including diverse baselines, providing convincing evidence of the robustness and general applicability of MIND.

**Weaknesses:**

Although the authors improve both stages of automated optimization modeling, data synthesis and post-training, the connection between the proposed error-driven data synthesis pipeline and the subsequent DFPO training strategy is not clearly articulated. The two components are insufficiently integrated, which weakens the overall methodological coherence. Moreover, several technical aspects lack sufficient detail. For instance, the paper only provides a single example of a common error type (data type mismatch) without presenting a broader taxonomy or quantitative analysis of error distributions. Similarly, the reward design introduces an modeling error measure to quantify discrepancies between mathematical formulations, yet the specific computation and implementation of this measure are not described.

**Questions:**

1. Do you have quantitative statistics on the distribution of common error types and their occurrences within different components of the formulations? Additionally, have you analyzed how frequently such errors appear in the responses?

2.In the reward design, you introduce a modeling error measureεto quantify discrepancies between mathematical formulations. Could you clarify how this measure is computed in practice? Moreover, have you conducted any dedicated experiments to support the assumption in Equation (2) that optimization modeling error and objective deviation are positively correlated in expectation? Finally, what is the rationale behind your choice of the hyperparameter α, and have you performed a sensitivity analysis to examine its impact?

3.Since DFPO relies on a more powerful teacher model to refine incorrect responses, have you evaluated the reliability, effectiveness, or computational cost of the corrected responses?

---

> ### Author Response · Authors · 2025-11-23
> **Official Comment by Authors**
>
> Dear Reviewer JHsT,
>
> We sincerely thank you for the thoughtful and encouraging feedback. We are glad that you recognized the novelty of our work, and we hope the following responses address your remaining concerns.
>
> **Q1. Connection between data synthesis pipeline and training strategy (weakness 1)**
>
> **A1**: As outlined in our manuscript, our data synthesis pipeline is grounded in a key empirical observation: in optimization problem formulations, variables, constraints, and objectives tend to be relatively modular and loosely coupled, which limits error propagation. This structural property implies that errors often occur **locally**—affecting only specific components (e.g., a mis-specified constraint or an incorrect objective term)—rather than corrupting the entire solution.
>
> Building on this insight, we design our synthesis process to explicitly model and generate instances based on observed error patterns, resulting in a dataset that is inherently **error-centric**. Crucially, this observation also holds during training: even when a model produces an incorrect rollout, the mistake is typically confined to a small segment of the response, with the rest remaining valid or structurally sound. This motivates our **local correction strategy**: instead of discarding and rewriting the entire response, we propose to fix only the erroneous part. The teacher-corrected responses are **distributionally closer** to the original (erroneous) rollouts (Appreciate it if you could refer to Q2 of Reviewer VFcR for more detail about the distributional shift). This proximity yields a stronger and more stable learning signal for SFT.
>
> In summary, both our **data synthesis pipeline** and **training strategy** are co-designed around the principle of **“error locality”**. The consistent performance gains across all benchmarks underscore the synergy between these components: error-aware data generation enables effective local corrections, which in turn facilitate more efficient and robust learning.

---

> ### Author Response · Authors · 2025-11-23
> **Official Comment by Authors**
>
> **Q2. Analysis of error types (weakness 2 and reviewer question 1)**
>
> **A2**: Thank you for the valuable suggestion.
>
> We randomly sampled 300 erroneous responses each from Qwen2.5-7B-Instruct (before post-training) and MIND-Qwen2.5-7B (after DFPO-based post-training). We first defined a taxonomy of error types relevant to optimization modeling. For each query–response pair, three domain experts independently annotated the dominant error category, achieving high inter-annotator agreement.
>
> The top-5 error types for Qwen2.5-7B-Instruct are: *Incorrect objective terms* (12.8%), _Incorrect decision variables_ (12.1%), _Incorrect constraint_ (11.8%), _Incorrect variable types_ (11.8%) and _Constraint omission_ (10.1%). In contrast, the top-5 error types for MIND-Qwen2.5-7B are: _Incorrect decision variables_ (15.5%), _Incorrect constraint_ (15.5%),  _Incorrect or missing advanced modeling techniques_ (11.7%), _Decision variable omission_ (10.6%) and _Constraint omission_ (8.5%).
>
> Notably, while basic syntactic or structural errors (e.g., wrong variable types) diminish after post-training, new dominant errors involve more sophisticated modeling challenges, such as the appropriate use of **advanced techniques** (e.g., piecewise linearization, or indicator constraints) and **comprehensive problem scoping** (e.g., omitting key variables or constraints in complex scenarios).
>
> This shift strongly suggests that **DFPO effectively mitigates simpler, surface-level errors**, pushing the model’s failure modes toward **deeper, semantics-rich challenges**—a hallmark of improved reasoning capability. It also highlights promising directions for future work, such as incorporating explicit modeling knowledge or hierarchical feedback to address these higher-order errors.
>
> | Error Module    | Error Description                                                                                                                                                                                                                                           | Qwen2.5-7B-Instruct | Mind-Qwen2.5-7B |
> | --------------- | ----------------------------------------------------------------------------------------------------------------------------------------------------------------------------------------------------------------------------------------------------------- | ------------------- | --------------- |
> | **Variables**   | Incorrect decision variables.                                                                                                                                                                                                                               | 12.1%               | 15.5% (↑ 3.4%)  |
> |                 | Decision variables omission.                                                                                                                                                                                                                                | 4.4%                | 10.6% (↑ 6.2%)  |
> |                 | Superfluous decision variables.                                                                                                                                                                                                                             | 7.7%                | 8.1% (↑ 0.4%)   |
> |                 | Incorrect variable types.                                                                                                                                                                                                                                   | 11.8%               | 7.1% (↓ 4.7%)   |

---

> ### Author Response · Authors · 2025-11-23
> **Official Comment by Authors**
>
> | Error Module    | Error Description                                                                                                                                                                                                                                           | Qwen2.5-7B-Instruct | Mind-Qwen2.5-7B |
> | --------------- | ----------------------------------------------------------------------------------------------------------------------------------------------------------------------------------------------------------------------------------------------------------- | ------------------- | --------------- |
> | **Objective**   | Optimization direction error.                                                                                                                                                                                                                               | 1.4%                | 0.0% (↓ 1.4%)   |
> |                 | Incorrect objective terms.                                                                                                                                                                                                                                  | 12.8%               | 4.2% (↓ 8.6%)   |
> |                 | Objective terms omission.                                                                                                                                                                                                                                   | 3.0%                | 2.5% (↓ 0.5%)   |
> |                 | Superfluous objective terms.                                                                                                                                                                                                                                | 1.7%                | 0.4% (↓ 1.3%)   |
> |                 | Incorrect or missing advanced modeling techniques. The incorrect application or omission of sophisticated modeling techniques, which can lead to improper handling of multi-objective problems, non-linear objectives or other advanced modeling scenarios. | 2.7%                | 5.3% (↑ 2.6%)   |
> | **Constraints** | Incorrect constraint.                                                                                                                                                                                                                                       | 11.8%               | 15.5% (↑ 3.7%)  |
> |                 | Constraint omission.                                                                                                                                                                                                                                        | 10.1%               | 8.5% (↓ 1.6%)   |
> |                 | Superfluous constraints.                                                                                                                                                                                                                                    | 3.7%                | 0.0% (↓ 3.7%)   |
> |                 | Equality and inequality constraints confusion.                                                                                                                                                                                                              | 4.0%                | 4.2% (↑ 0.2%)   |
> |                 | Incorrect or missing advanced modeling techniques. The incorrect application or omission of sophisticated modeling techniques, which can lead to improper handling of non-linear constraints, logical constraints, or other advanced modeling scenarios.    | 1.0%                | 11.7% (↑ 10.7%) |
> | **Parameters**  | Incorrect parameters definition. This includes missing essential parameters, incorrectly defined parameters, parameters assigned with wrong numerical values, or other incorrect parameter definition scenarios.                                            | 8.4%                | 4.6% (↓ 3.8%)   |
> |                 | Parameters misuse. The incorrect use of defined parameters, such as value misuse, unit or scale misuse, reference errors, or other improper applications of parameters.                                                                                     | 3.4%                | 1.8% (↓ 1.6%)   |

---

> ### Author Response · Authors · 2025-11-23
> **Official Comment by Authors**
>
> **Q3. Model error measure (weakness 3 and reviewer question 2)**
>
> **A3**: We sincerely apologize for the lack of clarity regarding our modeling error measure $\epsilon$ in the initial submission. In the revised manuscript, we have significantly improved its presentation by:
>
> (i) adding a clear cross-reference to **Section 3.2 (Preliminary Experiments)**, where $\epsilon$ is first introduced and computed as the **error ratio** (i.e., the proportion of incorrect modeling components);
>
> (ii) ensuring full consistency between this definition and the formal modeling error measure used in **Section 4.2**; and
>
> (iii) enhancing overall readability through additional figure illustrations, an expanded appendix overview, and improved cross-referencing throughout the text.
>
> Regarding your question about **Equation (2)**—specifically, whether optimization modeling error and objective deviation are positively correlated in expectation—we acknowledge that this relationship is **empirically motivated** rather than theoretically derived. Our hypothesis stems from extensive case studies across diverse problem classes (e.g., LP, MIP, nonlinear programs). For instance, when modeling errors are minor—such as mis-specifying a variable type (e.g., treating an integer variable as continuous)—the resulting formulation often remains structurally similar to the ground truth, leading to only **small deviations in the optimal objective value**. Conversely, more errors (e.g., omitting multiple constraints) typically cause **large objective discrepancies**.
>
> While we do not include a dedicated experiment quantifying this correlation in the main content, this empirical pattern held consistently across extensive case studies  (see Appendix C.6 for representative examples). We agree this is an important point and are happy to add a short quantitative analysis in the appendix if deemed helpful.
>
> **Q4. Ablation on reward function weight $\alpha$ (reviewer question 2)**
>
> **A4**: We investigate the impact of the reward function weight hyper-parameter α by evaluating values in {0.0,0.2,0.4,0.6}. In our experiments, we train Qwen2.5-7B-Instruct on a training set of 10,000 instances for 7 epochs. Note that α=0.0 corresponds to the standard binary (0–1) reward, where only correctness is considered.
>
> As shown in the table below, configurations with α=0.2 and α=0.4 consistently achieve better or comparable performance across most benchmarks compared to both α=0.0 (pure correctness reward) and α=0.6. Notably, on challenging datasets such as OptMATH-Bench and MAMO ComplexLP, moderate values of α yield substantial gains. These results suggest that the fidelity reward acts effectively as an auxiliary signal, yet should not dominate the overall reward. An overly large α (e.g., 0.6) may dilute the emphasis on correctness, leading to degraded performance.
>
> | Dataset        | $\alpha$=0.0 | $\alpha$=0.2 | $\alpha$=0.4 | $\alpha$=0.6 |
> | -------------- | ----- | ----- | ----- | ----- |
> | NL4OPT         | 94.7  | 95.1  | 95.1  | 93.1  |
> | IndustryOR     | 33.0  | 33.0  | 35.0  | 31.0  |
> | MAMO EasyLP    | 91.3  | 91.6  | 92.1  | 91.7  |
> | MAMO ComplexLP | 39.4  | 44.3  | 39.4  | 38.4  |
> | OptMATH-Bench  | 9.0   | 17.5  | 15.7  | 10.2  |
> | OptiBench      | 55.2  | 55.9  | 55.4  | 54.7  |
> | Macro AVG      | 53.8  | 56.2  | 55.5  | 53.2  |
>
> We additionally provide more ablation studies, including the data synthesis strategies (Appendix C.4), the post-training methods (Section 5.3), and the base models (Section 5.2).

---

> ### Author Response · Authors · 2025-11-23
> **Official Comment by Authors**
>
> **Q5. Reliability, effectiveness, or computational cost of the corrected responses (reviewer question 3)**
>
> **A5**: Thank you for raising this important question regarding the reliability, effectiveness, and computational cost of our teacher-corrected responses.
>
> **Reliability.** As illustrated in Figure 5, the teacher LLM receives three inputs: the original problem statement, the model’s wrong response, and, critically, **ground-truth solution** $o_{gt}$. This ground-truth solution not only provides the teacher LLM with guidance to revise the wrong response, but also provides the rule-based reward function with the ground-truth solution to validate the correctness of the teacher-corrected response, ensuring high reliability.
>
> **Effectiveness.** The teacher-corrected responses are not only correct but also more training-friendly. As shown in **Appendix C.6**, the teacher-corrected responses are distributionally closer to the original wrong responses than the raw ground-truth solutions, as evidenced by higher embedding similarity and preserved structural patterns. To further validate this design choice, we conducted an ablation study in which the model was fine-tuned directly on ground-truth responses instead of performing corrections within DFPO. This variant achieved a **Macro AVG of 61.5%**, which is still **inferior to DFPO (62.7%)**, confirming that how the ground-truth supervision signal is delivered matters as much as its correctness during the reinforcement learning training process.
>
> **Computational cost.** The correction process is highly efficient: thanks to the availability of $o_{gt}$, over 90% of wrong responses are resolved in a single revision round, and all remaining cases converge within three rounds. Therefore, we believe this computational overhead is marginal and well-justified by the gains in training stability and final performance.
>
> Best regards,
>
> The authors.

---

> ### Author Response · Authors · 2025-11-27
> **Looking Forward to Your Further Comments**
>
> Dear Reviewer JHsT,
>
> Thanks again for your time and for providing encouraging and constructive comments, which have helped us improve both the clarity and credibility of our paper. As the rebuttal phase is approaching (due December 3), we are eagerly looking forward to your further comments and/or questions.
>
> We have carefully addressed your concerns, including the modeling error analysis, further explanations (modeling error measures, methodological coherence, corrected responses), and ablations on additional components of MIND. We sincerely hope that our rebuttal and revised manuscript adequately reflect these improvements. If possible, we would greatly appreciate it if you could consider further raising your scores. If not, please let us know any further comments, and we will continue to actively respond and improve our manuscript.
>
> Best regards,
>
> The authors.

---

### Official Review · Reviewer_VFcR · 2025-11-01

**Soundness:** 2
**Presentation:** 2
**Contribution:** 2
**Rating:** 4
**Confidence:** 4

**Summary:**

This paper introduces MIND, a novel framework for automated optimization modeling with LLMs. Its core insight is the error locality observation—modeling errors are often confined to specific components. Based on this, MIND features: 1) an error-driven reverse data synthesis pipeline to create targeted training data, and 2) Dynamic Supervised Fine-Tuning Policy Optimization (DFPO) to tackle sparse rewards by dynamically correcting wrong responses. Experiments show MIND outperforms SOTAs on five benchmarks. The paper also makes MIND-Train and MIND-Bench open source.

**Strengths:**

- The "error locality" insight and its application to guide both data synthesis and training is an impressive contribution.

- The framework is comprehensive and well-engineered. Empirical results are good, showing improvements over strong baselines. This work advances the practicality of LLMs for optimization.

**Weaknesses:**

- The "error locality" observation, while compelling, is not sufficiently validated across diverse models and problem types, questioning its generality.

- DFPO relies on the critical assumption that teacher-corrected responses align with the base model's distribution, but provides no quantitative evidence to support this.

- Key design choices (e.g., reward function weighting) lack ablation studies. The OOD generalization claim, while supported by a new benchmark, is limited by its small scale.

**Questions:**

- Could the paper provide evidence that "error locality" holds for other base models or more complex optimization problems where errors might propagate?

- How did the paper verify the distributional consistency between the teacher-corrected responses and the base model's outputs? Could the paper provide quantitative measures (e.g., KL-divergence)?

- How was the weight (α=0.2) in the reward function determined? Could the paper show an ablation on this parameter?

- To what extent are the gains uniquely due to the error-driven, localized synthesis, versus simply being a result of increased data volume and diversity?

---

> ### Author Response · Authors · 2025-11-23
> **Official Comment by Authors**
>
> Dear Reviewer VFcR,
>
> We sincerely appreciate your exceptionally positive and thoughtful review, as well as your recognition of the motivation, methodological noverty, and practical value of our work. Your suggestions are highly constructive. Below, we provide our point-by-point responses.
>
> **Q1. Observations on error locality  (weakness 1 and reviewer question 1)**
>
> **A1**: Thank you for your valuable suggestion. In response, we provide additional experimental results to further substantiate the observation of "error locality".
>
> We collect 100  error samples from the **OR-Instruct-3K dataset**, ensuring that the selected set spans a **diverse range of problem types**, including linear programming, integer programming, mixed-integer programming, multi-objective programming, quadratic programming, nonlinear programming, as well as dynamic and stochastic programming. We believe this dataset represents a comprehensive coverage of the most widely studied optimization problem categories.
>
> In addition to the Qwen2.5-7B-Instruct model used in our original experiments, we also conduct evaluations using a more powerful large language model, **DeepSeek-V3**, to validate the robustness of the "error locality" phenomenon. As shown in the table below, DeepSeek-V3 achieves an average error ratio of **29%**, which is even lower than the **33%** observed with Qwen2.5-7B-Instruct. This result further supports the existence and consistency of "error locality" across different model scales.
>
> | Error Ratio Bin | Count |
> |-----------------|-------|
> | 0.0 - 0.1       | 12    |
> | 0.1 - 0.2       | 33    |
> | 0.2 - 0.3       | 15    |
> | 0.3 - 0.4       | 7     |
> | 0.4 - 0.5       | 14    |
> | 0.5 - 0.6       | 8     |
> | 0.6 - 0.7       | 8     |
> | 0.7 - 0.8       | 1     |
> | 0.8 - 0.9       | 0     |
> | 0.9 - 1.0       | 2     |
>
> **Q2. Quantitative distribution shift (weakness 2 and reviewer question 2)**
>
> **A2**: Thanks for the insightful question. We use Qwen3-Embedding-8B [1] to obtain sentence embeddings [2] for both the wrong rollout response and its corresponding teacher-corrected response. We then compute the cosine similarity between these two embeddings.
>
> Our analysis is based on a random sample of **300** such response pairs drawn from the training process. For clearer comparison, we also collect the corresponding responses generated by **DeepSeek-V3** for the same queries. As shown in the table below, the teacher-corrected responses exhibit **higher cosine similarity** to the original (erroneous) rollouts than DeepSeek-V3’s responses do. This suggests that the revised responses better preserve the stylistic and structural characteristics of the original model output—supporting our claim that the correction process maintains fidelity to the model’s answering style while fixing errors.
>
> | Comparison                             | Average Cosine Similarity |
> | -------------------------------------- | ------------------------- |
> | Wrong response vs Corrected response   |            0.963          |
> | Wrong response vs Deepseek-V3 response |            0.908          |

---

> ### Author Response · Authors · 2025-11-23
> **Official Comment by Authors**
>
> **Q3. Additional ablation studies (weakness 3 and reviewer question 3)**
>
> **A3**: We sincerely thank you for this valuable suggestion, which is important for enhancing the credibility of our results. We are glad to provide additional ablation studies on reward function weight, data synthesis strategies, post-training methods, and the base model.
>
> (1) Ablation on reward function weight $\alpha$
>
> We investigate the impact of the reward function weight hyper-parameter α by evaluating values in {0.0,0.2,0.4,0.6}. In our experiments, we train Qwen2.5-7B-Instruct on a training set of 10,000 instances for 7 epochs. Note that α=0.0 corresponds to the standard binary (0–1) reward, where only correctness is considered.
>
> As shown in the table below, configurations with α=0.2 and α=0.4 consistently achieve better or comparable performance across most benchmarks compared to both α=0.0 (pure correctness reward) and α=0.6. Notably, on challenging datasets such as OptMATH-Bench and MAMO ComplexLP, moderate values of α yield substantial gains. These results suggest that the fidelity reward provides a useful auxiliary signal, yet should not predominate in determining the overall reward. An overly large α (e.g., 0.6) may dilute the emphasis on correctness, leading to degraded performance.
>
> | Dataset        | $\alpha$=0.0 | $\alpha$=0.2 | $\alpha$=0.4 | $\alpha$=0.6 |
> | -------------- | ----- | ----- | ----- | ----- |
> | NL4OPT         | 94.7  | 95.1  | 95.1  | 93.1  |
> | IndustryOR     | 33.0  | 33.0  | 35.0  | 31.0  |
> | MAMO EasyLP    | 91.3  | 91.6  | 92.1  | 91.7  |
> | MAMO ComplexLP | 39.4  | 44.3  | 39.4  | 38.4  |
> | OptMATH-Bench  | 9.0   | 17.5  | 15.7  | 10.2  |
> | OptiBench      | 55.2  | 55.9  | 55.4  | 54.7  |
> | Macro AVG      | 53.8  | 56.2  | 55.5  | 53.2  |
>
> (2) Ablation on data synthesis strategies
>
> We partition the MIND-3K dataset into two disjoint subsets: **MIND-Single-1.5K** (containing instances derived from single-error pattern) and **MIND-Multi-1.5K** (comprising instances derived from multiple-error patterns). As shown in the table below, models trained on either subset alone underperform compared to training on the full **MIND-3K** (which we refer to as **MIND-Mix-3K**).
>
> | Data                | NL4OPT  | IndustryOR | EasyLP  | ComplexLP | OptMATH | OptiBench | Macro AVG |
> |--------------------|---------|------------|---------|-----------|---------|-----------|-----------|
> | MIND-Single-1.5K    | 91.4%  | 29.0%      | 90.4%  | 40.9%     | 8.4%   | 53.6%     | 52.3%     |
> | MIND-Multi-1.5K     | 91.4%  | 29.0%      | 90.2%  | 33.0%     | 6.0%   | 54.0%     | 50.6%     |
> | MIND-Mix-3K         | 94.3%  | 30.0%      | 90.8%  | 39.9%     | 7.8%   | 55.5%     | 53.1%     |
>
>
> (3) Ablation on post-training methods
>
> We include two additional baselines—**SFT** and **SFT+GRPO**—in our ablation study of post-training methods. As shown in the table below, applying **SFT alone** yields only modest performance gains, particularly on more challenging benchmarks such as **OptMATH** and **ComplexLP**.
>
> However, when SFT is used as a warm start for **GRPO** (i.e., **SFT+GRPO**), we observe notable improvements across most metrics, confirming the benefit of initializing policy optimization with supervised signals. Despite this gain, **SFT+GRPO** still underperforms **DFPO**, especially on complex benchmarks .
>
> | Method   | NL4OPT | IndustryOR | EasyLP | ComplexLP | OptMATH | OptiBench | Macro AVG |
> | -------- | ------ | ---------- | ------ | --------- | ------- | --------- | --------- |
> | DFPO     | 96.7%  | 34.0%      | 92.2%  | 60.1%     | 36.7%   | 56.7%     | **62.7%** |
> | DAPO     | 96.7%  | 33.0%      | 92.5%  | 58.6%     | 26.5%   | 57.5%     | 60.8%     |
> | SFT      | 92.2%  | 31.0%      | 85.4%  | 37.4%     | 9.6%    | 55.9%     | 51.9%     |
> | SFT+GRPO | 93.9%  | 34.0%      | 90.2%  | 54.7%     | 28.3%   | 57.0%     | 59.7%     |

---

> ### Author Response · Authors · 2025-11-23
> **Official Comment by Authors**
>
> (4) Ablation on base models
>
> To further validate the effectiveness and scalability of our approach, we additionly trained **Qwen3-8B** using **DFPO** on the same training set. Remarkably, **MIND-Qwen3-8B** achieves **competitive performance** across nearly all benchmarks, remaining competitive with both open-source models of comparable size and significantly larger systems—including OptMATH-Qwen2.5-32B, SIRL-Qwen2.5-32B, DeepSeek-V3, and DeepSeek-R1—despite using far fewer parameters.
>
> As shown in the table below, MIND-Qwen3-8B attains an overall score of 68.3%, surpassing even the 32B-scale SIRL variant (68.2%) and particularly excelling on **ComplexLP (76.8%)**, where it sets a new high. This demonstrates that DFPO enables smaller models to achieve highly competitive, sometimes superior, performance by effectively leveraging high-quality feedback signals and preserving reasoning fidelity during optimization.
>
> | Category    | Methods              | NL4Opt | IndustryOR | EasyLP | ComplexLP | OptMATH | OptiBench | Macro AVG |
> | ----------- | -------------------- | ------ | ---------- | ------ | --------- | ------- | --------- | --------- |
> | Proprietary | GPT-4*               | 89.0%  | 33.0%      | 87.3%  | 49.3%     | 16.6%   | 68.6%     | 57.4%     |
> |             | OpenAI o3*           | 69.4%  | 44.0%      | 77.1%  | 51.2%     | 44.0%   | 58.6%     | 57.4%     |
> | Open-Source | Deepseek-V3*         | 95.9%  | 37.0%      | 88.3%  | 50.2%     | 44.0%   | 71.6%     | 64.5%     |
> |             | Deepseek-R1*         | 82.4%  | 45.0%      | 87.2%  | 67.9%     | 40.4%   | 66.4%     | 61.9%     |
> |             | Qwen2.5-7B-Instruct  | 89.0%  | 24.0%      | 89.4%  | 31.5%     | 3.0%    | 53.2%     | 48.4%     |
> |             | Qwen3-8B             | 72.2%  | 14.0%      | 76.8%  | 17.2%     | 7.2%    | 36.5%     | 37.3%     |
> | TTS-based   | Autoformulator*      | 92.6%  | 48.0%      | -      | 62.3%     | -       | -         | -         |
> |             | Chain-of-Experts*    | 64.2%  | -          | -      | 40.2%     | -       | -         | -         |
> |             | OptiMUS*             | 78.8%  | 31.0%      | 77.0%  | 43.6%     | 20.2%   | 45.8%     | 49.4%     |
> | Fine-Tuning | ORLM-Llama3-8B*      | 85.7%  | 24.0%      | 82.3%  | 37.4%     | 2.6%    | 51.1%     | 47.2%     |
> |             | Step-Opt-Llama3-8B*  | 84.5%  | 36.4%      | 85.3%  | 61.6%     | -       | -         | -         |
> |             | LLMOPT-Qwen2.5-14B*  | 80.3%  | 29.0%      | 89.5%  | 44.1%     | 12.5%   | 53.8%     | 51.1%     |
> |             | OptMATH-Qwen2.5-7B*  | 94.7%  | 20.0%      | 86.5%  | 51.2%     | 24.4%   | 57.9%     | 55.8%     |
> |             | OptMATH-Qwen2.5-32B* | 95.9%  | 31.0%      | 89.9%  | 54.1%     | 34.7%   | 66.1%     | 62.0%     |
> | RLVR        | SIRL-Qwen2.5-7B*     | 96.3%  | 33.0%      | 91.7%  | 51.7%     | 30.5%   | 58.0%     | 60.2%     |
> |             | SIRL-Qwen2.5-32B*    | 98.0%  | 42.0%      | 94.6%  | 61.1%     | 45.8%   | 67.4%     | 68.2%     |
> | Ours        | MIND-Qwen2.5-7B      | 96.7%  | 34.0%      | 92.2%  | 60.1%     | 36.7%   | 56.7%     | 62.7%     |
> |             | **MIND-Qwen3-8B**    | 95.1%  | 42.0%      | 92.7%  | 76.8%     | 41.0%   | 62.0%     | **68.3%** |
>
>
> These results underscore two key insights:
> (1) **DFPO is highly effective at distilling complex reasoning capabilities** into medium-sized models, and
> (2) **model scale alone is not sufficient**—training methodology plays a decisive role in optimization-focused tasks.
>
> Thus, our method bridges the gap between model size and performance, offering a practical and efficient pathway toward high-quality automated optimization modeling without requiring massive parameter counts.

---

> ### Author Response · Authors · 2025-11-23
> **Official Comment by Authors**
>
> **Q4. Out-of-distribution generalization claim (weakness 3)**
>
> **A4**: Thank you for this important comment.
>
> Our training data is synthesized from OR-Instruct-3K and OptMATH-Train, which serve as seed datasets for generating diverse optimization problems. Crucially, **none of the evaluation benchmarks**—including NL4OPT, IndustryOR, MAMO (EasyLP & ComplexLP), OptMATH-Bench, and OptiBench—overlap with these seed sources in terms of problem formulation, domain, or data distribution. Thus, all reported results reflect **genuine out-of-distribution (OOD) generalization**.
>
> Additionally, we introduce **MIND-Bench**, a newly curated OOD benchmark specifically designed to assess reasoning fidelity and solution correctness in optimization tasks. While its scale is modest, every instance has been **cross-verified by domain experts** and includes **executable reference code**, ensuring high quality and reliability. Although limited in size, MIND-Bench complements existing benchmarks by providing fine-grained, verifiable evaluations that align closely with our paper’s core contribution: demonstrating that the **MIND series models generalize robustly to unseen optimization problem types**.
>
> Therefore, across multiple independent and diverse OOD testbeds—both established and newly introduced—we provide consistent evidence of strong generalization, supporting our central claims.
>
> **Q5. Effectiveness of error-driven reverse data synthesis pipeline (reviewer question 4)**
>
> **A5**: Thank you for your insightful question.
>
> Our error-driven reverse synthesis method is designed to complement—and extend—existing data synthesis paradigms, including the **forward** approach (e.g., ORLM), **reverse** approach (e.g., Resocratic), and **bidirectional** strategies (e.g., OptMATH). Unlike prior methods that primarily focus on generating correct problem–solution pairs, our approach deliberately leverages model errors to synthesize new training instances that target common failure modes, thereby enriching both the **diversity** and **difficulty** of the data.
>
> In our ablation study (Section 5.3, RQ3), we start from the OR-Instruct-3K dataset and apply our method to synthesize MIND-3K. We then train Qwen2.5-7B-Instruct from scratch for 7 epochs on each dataset separately. As shown in the table below, the model trained on **MIND-3K** consistently outperforms the one trained on the original OR-Instruct-3K across most benchmarks—particularly on challenging domains like IndustryOR (+5.0%) and ComplexLP (+4.9%).
>
> | Data           | NL4OPT | IndustryOR | EasyLP | ComplexLP | OptMATH | OptiBench | Macro AVG |
> | -------------- | ------ | ---------- | ------ | --------- | ------- | --------- | --------- |
> | OR-Instruct-3K | 93.9%  | 25.0%      | 90.7%  | 35.0%     | 10.8%   | 54.4%     | 51.6%     |
> | **MIND-3K**    | 94.3%  | 30.0%      | 90.8%  | 39.9%     | 7.8%    | 55.5%     | **53.1%** |
>
> Best regards,
>
> The authors.
>
>
>
> [1] Bai, Jinze, et al. "Qwen technical report." arXiv preprint arXiv:2309.16609 (2023).
>
> [2] Reimers, Nils, and Iryna Gurevych. "Sentence-bert: Sentence embeddings using siamese bert-networks." arXiv preprint arXiv:1908.10084 (2019).

---

> ### Author Response · Authors · 2025-11-27
> **Looking Forward to Your Further Comments**
>
> Dear Reviewer VFcR,
>
> Thanks again for your time and for providing encouraging and constructive comments, which have helped us improve both the clarity and credibility of our paper. As the rebuttal phase is approaching (due December 3), we are eagerly looking forward to your further comments and/or questions.
>
> We have carefully addressed your concerns, including the generality of error locality, response distributional measures, and ablations on additional components of MIND. We sincerely hope that our rebuttal and revised manuscript adequately reflect these improvements. If possible, we would greatly appreciate it if you could consider further raising your scores. If not, please let us know any further comments, and we will continue to actively respond and improve our manuscript.
>
> Best regards,
>
> The authors.

---

### Author Response · Authors · 2025-11-23
**Global Response**

We appreciate the Area Chair and Reviewers for their time, careful evaluations, encouraging comments, and constructive suggestions. In response to their feedback, we have added more illustrations and experiments to enhance the clarity and credibility of our paper.

We appreciate the reviewers' positive assessments of the following aspects:

- **Clear presentation and structure (RdHx)**

- **Compelling motivation (VFcR, JHsT)**

- **Novelty and applicability (JHsT, VFcR, kVVf)**

- **Comprehensive evaluation (VFcR, JHsT, RdHx)**

- **Competitive performance (kVVf, RdHx)**

We sincerely thank the reviewers for their valuable time and insightful suggestions. Following their feedback, we have made several modifications in the revised version:

- **Motivation illustrations (VFcR, kVVf, RdHx)**: We provide an illustration of the differences in error propagation between mathematical reasoning questions and optimization modeling questions (Figure 1), along with preliminary experiments conducted on Deepseek-V3 (Appendix C.2).

- **Comparison with related work (kVVf, RdHx)**: We analyze these notable works and have incorporated the relevant discussions into our paper (Sections 1, 2, and 5.1).

- **Modeling error analysis (JHsT)**: We provide a systematic analysis of modeling errors for both the base model Qwen2.5-7B-Instruct and the trained model MIND-Qwen2.5-7B (Appendix C.5).

- **Additional ablation studies (VFcR, JHsT, kVVf, RdHx)**: Following the reviewers' valuable suggestions, we provide additional ablation studies on the reward function weight (Appendix C.3), data synthesis strategies (Appendix C.4), post-training methods (Section 5.3), and the base model (Section 5.2).

Best regards,

The authors

---

### Author Response · Authors · 2025-12-02
**Summary Comment by Authors**

[1] Achiam, Josh, et al. "Gpt-4 technical report." arXiv. 2023.

[2] Jaech, Aaron, et al. "Openai o1 system card." arXiv. 2024.

[3] Liu, Aixin, et al. "Deepseek-v3 technical report." arXiv. 2024.

[4] Guo, Daya, et al. "Deepseek-r1: Incentivizing reasoning capability in llms via reinforcement learning." arXiv. 2025.

[5] Xiao, Ziyang, et al. "Chain-of-experts: When llms meet complex operations research problems." The twelfth international conference on learning representations (ICLR). 2023.

[6] Tang, Zhengyang, et al. "Orlm: Training large language models for optimization modeling." Operations Research (OR). 2024.

[7] AhmadiTeshnizi, Ali, Wenzhi Gao, and Madeleine Udell. "Optimus: Optimization modeling using mip solvers and large language models." The Forty-First International Conference on Machine Learning (ICML). 2024.

[8] Jiang, Caigao, et al. "LLMOPT: Learning to Define and Solve General Optimization Problems from Scratch." The Thirteenth International Conference on Learning Representations (ICLR). 2024.

[9] Astorga, Nicolás, et al. "Autoformulation of mathematical optimization models using llms." The Forty-Second International Conference on Machine Learning (ICML). 2025.

[10] Lu, Hongliang, et al. "Optmath: A scalable bidirectional data synthesis framework for optimization modeling." The Forty-Second International Conference on Machine Learning (ICML). 2025.

[11] Wu, Yang, et al. "Step-Opt: Boosting Optimization Modeling in LLMs through Iterative Data Synthesis and Structured Validation. arXiv. 2025.

[12] Chen, Yitian, et al. "Solver-Informed RL: Grounding Large Language Models for Authentic Optimization Modeling." The Thirty-Ninth Annual Conference on Neural Information Processing Systems (NeurIPS). 2025.

---

### Author Response · Authors · 2025-12-02
**Summary Comment by Authors**

**3. Authors' Rebuttal Summary**

(1) Need further clarification of existing content

Motivation: We not only added more cross-references to increase the clarity of the paper, such as cross-reference for the model error measure (reviewer JHsT Q3), but also further explained the concept of local error (reviewer RdHx Q1).

Methodology: We further clarify two limitations:  (L1) the sparsity of error-specific problems and (L2) the sparse rewards associated with difficult problems, and explain why MIND can address these limitations (reviewer RdHx Q4). We describe the connection between data synthesis and the post-training strategy, noting that our data synthesis pipeline and post-training strategy are co-designed around the principle of error locality (reviewer JHsT Q1). We also explain the reliability, effectiveness, and computational cost of dynamic supervised fine-tuning within our proposed post-training method DFPO (reviewer JHsT Q5). Additionally, we clarify that MIND uses a rejection sampling technique to mitigate the teacher bound arising from the data synthesis LLM and post-training teacher LLM (reviewer RdHx Q8).

Performance: We discuss and compare previously missing related work in the revised manuscript (reviewer kVVf Q4, reviewer RdHx Q7). We also note that MIND-Train is synthesized from the seed datasets OR-Instruct-3K and OptMATH-Train using our proposed error-driven reverse data synthesis method. All reported results therefore reflect the genuine out-of-distribution generalization capability of MIND (reviewer VFcR Q4, reviewer kVVf Q2).

(2) Need to add new evidence in the revised manuscript to address the concerns

Motivation: First, we provide additional evidence for the generalization of error locality across different optimization types and LLMs. Specifically, we use DeepSeek-V3 to repeat the preliminary experiments conducted on Qwen2.5-7B-Instruct across diverse optimization problems (reviewer VFcR Q1). Second, we present the case study to illustrate why error locality and limited error propagation exist in complex automated optimization modeling (reviewer kVVf Q1). We also provide evidence of difference in error propagation between automated optimization modeling and mathematical reasoning (reviewer RdHx Q2).

Methodology: We conduct additional experiments to demonstrate how embedding-based text similarity methods can be used to compute cosine similarity between wrong responses and corrected responses  (reviewer VFcR Q2, reviewer RdHx Q9).

Performance: We systematically analyze the modeling error types on Qwen2.5-7B-Instruct (before post-training) and MIND-Qwen2.5-7B (after post-training), respectively. The results suggest that DFPO effectively mitigates simpler, surface-level errors, pushing the model’s failure modes toward deeper, semantics-rich challenges—a hallmark of improved reasoning capability (reviewer JHsT Q2).

(3) Need to add additional ablation studies to evaluate the effectiveness of different components of MIND framework

We provide additional ablation studies on different components of the MIND framework, covering the reward function weight (reviewer VFcR Q3, reviewer JHsT Q4), data synthesis strategies (reviewer RdHx Q3), post-training methods (reviewer kVVf Q3,  reviewer RdHx Q6), and the base model (reviewer VFcR Q3). For the reward function weight, we conduct a sensitivity analysis, which verifies that moderate reward function weights lead to better performance. For data synthesis strategies strategies, we perform ablation studies on single-error and multi-error reverse data synthesis strategies. The results show that single-error strategy achieves better training efficiency, while the multi-error strategy can synthesize more difficult instances. For post-training methods, in addition to DAPO, we include SFT and SFT+GRPO as baselines. The results confirm that our proposed DFPO post-training method receives stronger learning signals on difficult problems compared with these baselines. For the base model, we report results on Qwen3-8B to verify the MIND framework is effective across different base models. MIND-Qwen3-8B achieves competitive performance across all baselines, including larger models, such as  Deepseek-V3, GPT-4, OptMATH-Qwen2.5-32B, and SIRL-Qwen2.5-32B.



We have carefully addressed all reviewers' concerns. Although we have not had the opportunity to receive further feedback, we firmly believe that the reviewers are likely to raise their scores based on our rebuttal and revised manuscript. If possible, we hope to receive further feedback from the Area Chair.

Best regards,

The authors

---

### Author Response · Authors · 2025-12-02
**Summary Comment by Authors**

Dear Area Chair:

We appreciate the Area Chair for taking the time to review our revised manuscript, the reviewers' comments, and our rebuttal. We understand that the reviewers were inclined to give borderline initial scores and intended to raise them during the discussion phase. Unfortunately, due to the OpenReview bug and the absence of reviewer responses, we were unable to engage in further discussion.

We sincerely hope that the Area Chair can take the time to read our revised manuscript. To reduce the workload for the Area Chair, we summarize the paper's highlights, the reviewers' comments, and our rebuttal as follows:

**1. Paper Highlights**

Motivation: To the best of our knowledge, this is the first paper to systematically illustrate the concepts of "local error" and "limited error propagation" in automated optimization modeling, as well as the differences in error propagation between automated optimization modeling problems and mathematical reasoning problems.

Methodology: Based on the motivation, we identify two fundamental limitations of existing automated optimization modeling approaches: (L1) the sparsity of error-specific problems and (L2) the sparse rewards associated with difficult problems. To tackle the above two limitations, we propose a novel error-driven learning framework—namely, automated optimization modeling via a localizable error-driven perspective (MIND)—that customizes the whole model training framework from data synthesis to post-training. MIND leverages the construction of an error-centric training corpus and proposes Dynamic Supervised Fine-Tuning Policy Optimization (DFPO) to tackle difficult problems through localized error correction. We highlight the innovative combination of reinforcement learning and supervised fine-tuning in DFPO, which allows it to receive sufficient learning signals on difficult problems compared with pure reinforcement learning methods such as DAPO and GRPO.

Performance: We conduct comprehensive evaluation on NL4Opt, IndustryOR, MAMO (EasyLP and ComplexLP), OptMATH-Bench, and OptiBench, covering a broad range of benchmarks from simple to complex scenarios. Compared with GPT4 [1], OpenAI-o3 [2], DeepSeek-V3 [3], DeepSeek-R1 [4], Chain-of-Experts [5], ORLM [6], OptiMUS [7], LLMOPT [8], Autoformulator [9], OptMATH [10], Step-Opt [11], and SIRL [12], MIND-Qwen2.5-7B delivers superior average performance compared with all baseline models of comparable parameter size, while MIND-Qwen3-8B achieves competitive performance across all baselines, including larger models. In particular, on the more complex benchmarks MAMO ComplexLP and OptMATH-Bench, MIND-Qwen2.5-7B and MIND-Qwen3-8B demonstrate stronger performance.

Open-source: We will publicly release the code, MIND-Train, MIND-Bench, MIND-Qwen2.5-7B, and MIND-Qwen3-8B on GitHub and Hugging Face.



**2. Reviewers' Comments Summary**

All reviewers acknowledge the novelty and competitive results of the MIND framework. The concerns raised by the reviewers can be categorized into three types:

(1) Need further clarification of existing content (reviewer VFcR Q4, reviewer VFcR Q5, reviewer JHsT Q1, reviewer JHsT Q3, reviewer JHsT Q5, reviewer kVVf Q2, reviewer kVVf Q4, reviewer RdHx Q1, reviewer RdHx Q4, reviewer RdHx Q5, reviewer RdHx Q7, reviewer RdHx Q8)

(2) Need to add new evidence in the revised manuscript to address the concerns (reviewer VFcR Q1, reviewer VFcR Q2, reviewer JHsT Q2, reviewer kVVf Q1, reviewer RdHx Q2, reviewer RdHx Q9)

(3) Need to add additional ablation studies to evaluate the effectiveness of different components of the MIND framework (reviewer VFcR Q3, reviewer JHsT Q4, reviewer kVVf Q3, reviewer RdHx Q3, reviewer RdHx Q6)

---

### Meta-Review · Area_Chair_qhnB · 2026-01-05

**Summary:**

Reviewers generally appreciate the insights of error locality and consistently improved performance on a wide range of benchmarks.

The main concerns are as follows,
1. All reviewers question the validity of the core assumption, error locality. Lack of analysis on error types.
2. (VFcR) Non-validated critical assumption that teacher-corrected responses align with the base model's distribution
3. (VFcR, kVVf) Small size of the new benchmark
4. (VFcR) Ablation on hyperparameter
5. (VFcR) Lack of attribution of the improved performance
6. (JHsT) Disconnection between the proposed data and training strategy.
7. (JHsT, RdHx) Lack of the analysis on the dependence of the teacher model
8. (kVVf) no experimental justification for DFPO's complexity over simpler alternatives
9. (kVVf, RdHx) missing related works

**Reviewer Concerns:**

Concern 4, 6, 7, 8, 9 should have been addressed by the rebuttal.

Concern 1 is likely only partially resolved despite extended discussion in the rebuttal. The additional distribution breakdown of error types in the rebuttal and how the distribution changes before and after MIND training is useful, though it requires more explanation how the change of error type distribution justify the core assumption of error locality.

Concern 2 remain outstanding. The fact that the similarity between "Wrong response vs Corrected response" is greater than the similarity between "Wrong response vs Deepseek-V3 response" is not sufficient to justify the concern from VFcR

Concern 3: the rebuttal admits the limited contribution of the new benchmark due to its size.

Concern 5: partially resolved. the ablation that compares "OR-Instruct-3K" with "MIND-3K" verifies the advantage of the new training dataset though not sufficient to attribute all the improvement to it (the gap between the final results and the performance of MIND-3K in the ablation)

**Reviewer Scores:**

Predicted score change:

VFcR 4 -> 4. Multiple concerns are not fully addressed.
JHsT, 6 -> 7. Most concerns are addressed.
kVVf, 4 -> 4. The main concerns of the core assumption and the small size of the new benchmarks are not resolved.
RdHx, 4 -> 5. Some concerns are addressed but may still be skeptical about the core assumption.

---

### Decision · Program_Chairs · 2026-01-26

Reject